# The underappreciated role of nonvolatile cations on aerosol ammonium-sulfate molar ratios

Hongyu Guo[1], Athanasios Nenes[1,2,3,4], Rodney J. Weber[1]

[1] School of Earth and Atmospheric Sciences, Georgia Institute of Technology, Atlanta, GA 30332, USA
[2] School of Chemical and Biomolecular Engineering, Georgia Institute of Technology, Atlanta, GA 30332, USA
[3] Institute for Chemical Engineering Sciences, Foundation for Research and Technology – Hellas, Patras, GR-26504, Greece
[4] Institute for Environmental Research and Sustainable Development, National Observatory of Athens, P. Penteli, Athens, GR-15236, Greece

*Correspondence to*: Rodney J. Weber, (rweber@eas.gatech.edu), Athanasios Nenes (athanasios.nenes@gatech.edu)

**Abstract.** Overprediction of fine particle ammonium-sulfate molar ratios ($R$) by thermodynamic models is suggested as evidence for interactions with organic constituents that inhibit the equilibration of gas phase ammonia with aerosol sulfate and questions the equilibrium assumption long thought to apply for submicron aerosol. This hypothesis is tested through thermodynamic analysis of ambient observations. We find that the deviation between $R$ from a molar ratio of 2 is strongly correlated with the concentration of sodium ($Na^+$), a nonvolatile cation (NVC), but exhibits no correlation to organic aerosol mass concentration or mass fraction. Thermodynamic predictions of both $R$ and ammonia gas-particle partitioning can accurately reproduce observations when small amounts of nonvolatile cations (NVC) are included in the calculations, whereas exclusion of NVCs results in predicted $R$ consistently near 2. The sensitivity of $R$ to small amounts of NVCs arises because when the latter are present but not included in the thermodynamic calculations, the missing cations are replaced with ammonium in the model ($NH_3$-$NH_4^+$ equilibrium shifts to the particle), resulting in $R$ that is biased high. Results and conclusions based on bulk aerosol considerations that assume all species are internally mixed are not changed even if NVCs and sulfate are largely externally mixed; fine particle pH is found to be much less sensitive to mixing state assumptions than molar ratios. We also show that the data used to support the "organic inhibition" of $NH_3$ from equilibrium, when compared against other network and field campaign data sets, displays a systematically and significantly lower $NH_4^+$ (thought to be from an evaporation bias), that is of the order of the effect postulated to be caused by organics. Altogether, these results question the postulated ability of organic compounds to considerably perturb aerosol acidity and prevent ammonia from achieving gas-particle equilibrium, at least for the locations considered. Furthermore, the results demonstrate the limitations of using molar ratios to infer aerosol properties or processes that depend on particle pH.

## 1. Introduction

pH is a fundamental aerosol property that affects aerosol formation and composition through reactions that involve the hydronium ion (e.g., Jang et al., 2002; Eddingsaas et al., 2010; Surratt et al., 2010) and gas-particle partitioning of semivolatile acids and bases (e.g., Fridlind and Jacobson, 2000; Young et al., 2013; Guo et al., 2016; Guo et al., 2017). Acidity also

modulates aerosol toxicity and atmospheric nutrient supply to the oceans by augmenting the solubility of transition metals and other nutrient species (Meskhidze et al., 2003; Nenes et al., 2011; Longo et al., 2016; Stockdale et al., 2016; Fang et al., 2017). Despite its importance, challenges in measuring fine mode particle pH have led to the adoption of measurable aerosol properties as acidity proxies, such as aerosol ammonium-sulfate ratio or ion balances with a priori assumed dissociation states (e.g., Paulot and Jacob, 2014; Wang et al., 2016; Silvern et al., 2017). Recent work has shown that such proxies are not uniquely related to pH

because they do not capture variability in particle water content, ion activity coefficients, or dissociation state of polyprotic acids and bases (Guo et al., 2015; Hennigan et al., 2015; Guo et al., 2016; Song et al., 2018). An alternative approach that better constrains aerosol pH is a thermodynamic analysis of semivolatile acid (or base) measurements, whose partitioning is observably sensitive to shifts in aerosol acidity (pH is optimally constrained when gas-particle concentration ratios approach 1:1), and with the aerosol water content or phase state constrained as well (Guo et al., 2015; Hennigan et al., 2015). $NH_3$-$NH_4^+$, $HNO_3$-$NO_3^-$,

and HCl-Cl$^-$ pairs often meet this condition for a wide range of atmospherically-relevant pH. The method has been utilized for a range of meteorological conditions (RH, T) and gas/aerosol concentrations demonstrating that model predictions are often in agreement with observations (Bougiatioti et al., 2016; Guo et al., 2016; Guo et al., 2017; Liu et al., 2017; Murphy et al., 2017; Song et al., 2018).

Despite their skill and widespread use in regional and global models, aerosol thermodynamic models can predict ammonium-sulfate molar ratios (Kim et al., 2015; Weber et al., 2016; Silvern et al., 2017) that departs from observations in seemingly counterintuitive ways. In the southeastern US, where total ammonium ($NH_x = NH_3 + NH_4^+$) is in large excess of particle sulfate, observed $NH_4^+/SO_4^{2-}$ molar ratios are in the range of 1-2 (Hidy et al., 2014; Guo et al., 2015; Kim et al., 2015). Thermodynamic models predict very low pH (0.5 to 2) (Guo et al., 2015) and molar ratios always close to 2 (Kim et al., 2015; Weber et al., 2016;

Silvern et al., 2017). This predicted-observed molar ratio discrepancy has led to the hypothesis that thermodynamic predictions are incorrect because they do not consider interactions with organic species, either in the form of films that inhibit gas-to-particle mass transfer of $NH_3$ or other mechanisms that are not accounted for (Silvern et al., 2017). Such limitations, if prevalent, are suggested to oppose the validity of aerosol thermodynamic equilibrium with significant impacts on aerosol chemistry and acidity-mediated processes worldwide (Silvern et al., 2017), especially given the expected increasing organic mass fractions in

the future due to reduced anthropogenic emissions, as seen with $SO_2$ emission reductions in the eastern US (Hand et al., 2012; Attwood et al., 2014; Hidy et al., 2014).

The effect of organic species on gas-particle equilibrium of inorganic species has been the subject of many past studies. Organic "films" are often hypothesized to act as barriers for gas-particle mass transfer, which given their ubiquity, means they require

special attention in studies. For example, Anttila et al. (2007) reports the formation of ~10 nm thick organic films in regions with monoterpene emissions, which is the largest source of summertime organic aerosol in the southeastern US (Zhang et al., 2018). Lab studies have shown that organic films may significantly slow down mass transfer of $NH_3$ from gas to particle at low relative humidity (less effect at higher RH, such as the southeastern US) (Daumer et al., 1992; Liggio et al., 2011) but have little effect on water vapor uptake for a large RH range (Garland et al., 2005). Such films, as noted by Silvern et al. (2017), would have

important implications for partitioning of $NH_3$ and other larger semivolatile molecules, such as $H_2O$, $HNO_3$ and organic acids.

However, in contrast, numerous studies show that $NH_3$, water vapor, and $HNO_3$ equilibrate with organic-rich atmospheric aerosols (Ansari and Pandis, 2000; Moya et al., 2001; Morino et al., 2006; Fountoukis et al., 2009; Guo et al., 2015; Guo et al., 2016; Guo et al., 2017; Liu et al., 2017; Murphy et al., 2017; Paulot et al., 2017), which suggest organic films, if present, do not impose considerable delays in mass transfer and gas-particle equilibration.

At low temperature and low relative humidity, particles may be in a semi-liquid or glassy state characterized by a very low molecular diffusivity throughout its volume (e.g., Zobrist et al., 2008; Bertram et al., 2011; Tong et al., 2011; Zobrist et al., 2011; Bones et al., 2012; Reid et al., 2018). When in this state, gas-particle mass transfer of all semivolatile components may be severely limited and require much longer time scales to equilibrate than the ~20 minutes typically thought to apply for $PM_1$

(Dassios and Pandis, 1999; Cruz et al., 2000; Fountoukis et al., 2009). However, such an effect has not been observed for the conditions in the eastern US, as there is good agreement between observed and predicted particle water, and partitioning of $NH_3$-$NH_4^+$ and $HNO_3$-$NO_3^-$, especially in cases where RH is sufficiently high (greater than 40%) to maintain the aerosol in a deliquesced (completely liquid) state (Guo et al., 2015; Guo et al., 2016).

Other reasons, unrelated to the presence of organic aerosol, may drive the observed molar ratio discrepancy. Analyses of aerosol acidity, molar ratios, and partitioning of semivolatile species, often neglect the variations of composition with size, especially in the $PM_1$ to $PM_{2.5}$ range (Keene et al., 1998; Fridlind and Jacobson, 2000; Nenes et al., 2011; Young et al., 2013; Bougiatioti et al., 2016; Fang et al., 2017). If acidity across size changes sufficiently, average equilibrium composition (including molar ratios) may deviate considerably against observations owing to the nonlinear dependence of partitioning with acidity (e.g., Guo et al.,

2015). Soluble nonvolatile cations (NVCs, such as $Na^+$, $K^+$, $Ca^{2+}$, $Mg^{2+}$), potentially present in large quantities in $PM_{2.5}$ and to a lesser extent in $PM_1$, can strongly modulate acidity and molar ratios. NVCs are often omitted from thermodynamic calculations because of their relatively minor contribution to aerosol mass and ion charge balance; for similar reasons, NVCs are not routinely included in aerosol composition measurements; when they are, proximity to level of detection (LOD) often increases their concentration uncertainty. Here we show, based on analysis of observational aerosol and gas data sets, that excluding even small

amounts of NVC in thermodynamic analyses results in predicted $NH_4^+/SO_4^{2-}$ molar ratios close to 2, whereas including them brings model-predicted molar ratios into agreement with observed levels. We also assess the implications of using specific datasets on molar ratios, and the impact of adopting a size-averaged ("bulk") thermodynamic analysis against one that considers an incomplete mixing (size-dependent composition) of ambient aerosols.

## 2. Methods

**Molar ratios definition:** Two ammonium-sulfate aerosol molar ratios (mol mol$^{-1}$) are used in the following analysis,

$$R = \frac{NH_4^+}{SO_4^{2-}} \tag{1}$$

$$R_{SO_4} = \frac{NH_4^+ - NO_3^-}{SO_4^{2-}} \tag{2}$$

Both are based on mole concentrations in units of mol m$^{-3}$. $R_{SO4}$ is a more narrowly defined molar ratio that excludes $NH_4^+$ associated with $NO_3^-$, because some fractions of ammonium sulfate and ammonium nitrate can be associated with different sized particles (Zhuang et al., 1999) and molar ratios are calculated based on bulk composition data ($PM_{2.5}$ or $PM_1$). This issue is discussed in more detail below. The upper limit for $R$ and $R_{SO4}$ is 2 for a particle composition of pure $(NH_4)_2SO_4$, and a lower

limit of 0 for $R$ when $SO_4^{2-}$ is associated with other cations instead of $NH_4^+$ (e.g. $Na_2SO_4$) or if there is free $H_2SO_4$ in the aerosol. A negative $R_{SO4}$ can occur for conditions of high $NO_3^-$ and low $NH_4^+$, $SO_4^{2-}$ concentrations (e.g., $NaNO_3$), but are rare for ambient fine particles (at least not seen in the three data sets studied in this paper). $R$ or $R_{SO4}$ is typically observed in the range of 1 and 2 in the southeastern US (Hidy et al., 2014; Guo et al., 2015; Weber et al., 2016). In cases where $NO_3^-$ levels are low

relative to $SO_4^{2-}$, the two ratios, $R_{SO4}$ and $R$, are equivalent, as is observed in the summertime southeastern US, where $NO_3^-$ is typically ~0.2 µg m$^{-3}$, $NH_4^+$ ~1 µg m$^{-3}$, and $SO_4^{2-}$ ~3 µg m$^{-3}$ (Blanchard et al., 2013).

**Data:** Two datasets are mainly used for analysis; the Southern Oxidant and Aerosol Study (SOAS) and the Wintertime Investigation of Transport, Emissions, and Reactivity (WINTER). The SOAS study was conducted from 1 June to 15 July in the

summer of 2013 at a rural ground site in Centreville (CTR), AL, representative of the southeastern US background atmosphere in summer. $PM_{2.5}$ ions were determined with a Particle-Into-Liquid-Sampler coupled with an Ion Chromatograph (PILS-IC). The PILS-IC detects aerosol water-soluble anions and cations collected and diluted by deionized water to the extent of complete deprotonation of $H_2SO_4$ in the aqueous sample (Orsini et al., 2003). $NH_3$ was obtained from chemical ionization mass spectrometer measurements (You et al., 2014). In the following, we only use $PM_{2.5}$ ion data from a 12-day period (11-23 June) of

the SOAS campaign. (PILS $PM_1$ data were collected in the second half of the study and are not used here). Periods of rainfall are not included in the analysis, as equilibrium does not apply. The same data set was used to study pH sensitivity to sulfate and ammonia (Weber et al., 2016). $PM_{2.5}$ anion and cation data along with $NH_3$ and $HNO_3$ were also collected with a Monitor for AeRosols and GAses (MARGA) during SOAS (Allen et al., 2015). The WINTER data was collected during 13 research aircraft flights from 1 February to 15 March 2015 mainly sampling over the northeastern US. We use $PM_1$ aerosol data collected with a

High-Resolution Time-of-Flight Aerosol Mass Spectrometer (hereafter referred to as AMS), which have been extensively compared to the PILS anion measurements also made in that study (Guo et al., 2016). Details of the these two campaigns and instruments, and calculations and verification of pH based on the observation datasets, have been described in Guo et al. (2015) and Guo et al. (2016), respectively.

In the following analysis, we focus on $R$ for summertime data sets since $NO_3^-$ was generally low, and $R_{SO4}$ for wintertime data sets where higher $NO_3^-$ concentrations were observed. Thermodynamic analysis of both datasets indicate highly acidic aerosols with average pH~1 (Guo et al., 2015; Guo et al., 2016). At these pH levels, aerosol sulfate can be in the partially deprotonated form of $HSO_4^-$ instead of $SO_4^{2-}$. For example, 10% of the total sulfate is predicted to be $HSO_4^-$ for the SOAS condition (see Fig. S1 in the supplement). Free form $H_2SO_4$, which requires even lower pH, is rare. To avoid any confusion, we note that in this

study $SO_4^{2-}$ refers to the sum of total aqueous aerosol sulfate ($SO_4^{2-}$, $HSO_4^-$, and $H_2SO_4$), i.e., S(VI). Similarly, $NH_4^+$ refers to the sum of total aqueous ammonium ($NH_4^+$, $NH_3$) and $NO_3^-$ refers to the sum of total nitrate ($NO_3^-$, $HNO_3$) in aqueous aerosols. $SO_4^{2-}$, $NH_4^+$, and $NO_3^-$ are reported by PILS-IC. However, PILS-IC cannot distinguish the in-situ aerosol ion forms for collecting aerosols in diluted deionized water (i.e., the ionic strength is altered) (Orsini et al., 2003). The AMS vaporizes aerosols and ionizes non-refractory species with a 70 eV electron impact ionization and also cannot distinguish the dissociation states of

inorganic ions (DeCarlo et al., 2006).

In addition to the SOAS and WINTER data sets, the Southeastern Aerosol Research and Characterization (SEARCH) CTR sampling site (the same as SOAS) historical data from year 1998 to 2013 is re-analyzed to show that the thermodynamic model can reproduce the observed decreasing trend of $R_{SO4}$ when NVCs are considered. Molar ratios determined from the Chemical

Speciation Network (CSN), which were utilized and discussed by Silvern et al. (2017) and Pye et al. (2018), are not used in this

work because of a significant low bias when compared to the SEARCH and SOAS data (see Table S1 and S2 in the supplement). The discrepancy is likely due to the loss of semivolatile $NH_4^+$ collected on the CSN nylon filters (Yu et al., 2006; Silvern et al., 2017), and can result in an under-measurement in $R$, compared to online measurements, by as much as 1 unit (Table S1).

*Thermodynamic analysis of observations:* We have used the thermodynamic model ISORROPIA-II (Fountoukis and Nenes, 2007) to determine the liquid water content and composition (including $H^+$) of an $NH_4^+$-$SO_4^{2-}$-$NO_3^-$-$Cl^-$-$Na^+$-$Ca^{2+}$-$K^+$-$Mg^{2+}$-water inorganic aerosol (or a subset therein) and its partitioning with corresponding gases. A molality-based definition of pH is used:

$$\text{pH} = -\log_{10} \gamma_{H^+} H_{aq}^+ = -\log_{10} \frac{1000 \gamma_{H^+} H_{air}^+}{W_i + W_o} \cong -\log_{10} \frac{1000 \gamma_{H^+} H_{air}^+}{W_i} \tag{3}$$

where $\gamma_{H^+}$ is the hydronium ion activity coefficient (assumed = 1; note that the binary activity coefficients of ionic pairs, including $H^+$, is calculated in the model), $H_{aq}^+$ (mol kg$^{-1}$) and $H_{air}^+$ ($\mu$g m$^{-3}$) are the hydronium ion concentration in particle liquid
water and volume of air, respectively. $W_i$ and $W_o$ ($\mu$g m$^{-3}$) are particle water concentrations associated with inorganic and organic species, respectively. pH predicted solely with $W_i$ is systematically lower by 0.15-0.23 units but highly correlated (r$^2$ = 0.97) to pH predicted with measured total particle water ($W_i + W_o$) for the southeast US (which includes the SOAS study), where $W_o$ accounted for 35% of total particle water (Guo et al., 2015). For simplicity, we therefore use only $W_i$ for the following pH calculations. ISORROPIA-II was run in "forward" mode to calculate gas-particle equilibrium concentrations based on the input
of total concentration of various inorganic species (e.g., $NH_3$ + $NH_4^+$). In all cases we also chose a "metastable" (not "stable") solution, which assumes inorganic ions are associated with the aerosol components that are completely aqueous and contain no solid precipitate forms, other than $CaSO_4$ ($H_{aq}^+$ is meaningless in a completely effloresced aerosol). Given this phase state requirement, we restrict the analysis to conditions where RH > 40%.

*Mixing State:* Because the aerosol composition data is bulk $PM_1$ or $PM_{2.5}$, and used as input to ISORROPIA-II, the thermodynamic analysis implicitly assumes that all particle species were internally mixed, so that one value of pH represents the aerosols and governs the gas-particle partitioning. The existence of externally mixed particles may quantitatively and qualitatively affect the bulk thermodynamic analysis. To address this, we begin from the bulk analysis, then repeat the same calculation, augmenting each time the degree of external mixture of NVCs and sulfate. Direct measurements of aerosol mixing
state during SOAS suggests that ambient particles indeed exhibit a range of mixing states (Bondy et al., 2018). In the external mixing analysis, the bulk aerosol is split into two subgroups: (1) species largely found in $PM_1$ (e.g., $NH_4^+$ and $SO_4^{2-}$) and (2) species found in $PM_{1-2.5}$, which contains mostly the NVCs, $NO_3^-$ and some $SO_4^{2-}$ and $NH_4^+$. These two external mixtures are in equilibrium with gaseous $NH_3$ and $HNO_3$, and so interact through these species (i.e., $NH_4^+$ and $NO_3^-$ can move between the two). Nonvolatile species, such as $SO_4^{2-}$ and NVCs ($Na^+$) remain in the original size class assumed at the start of the analysis. To
determine the composition of the two subgroups, we iteratively solve for the equilibrium conditions, by sequentially calling ISORROPIA for each subgroup. The solution is found when the composition of each group no longer changes with iteration and both are in equilibrium with the gas phase species (in this case, $NH_3$, $HNO_3$, and $H_2O$ (water vapor)). Mass of each species (gas plus particle) is conserved at all times and constrained by the observations. Given that pH is size dependent and generally higher at larger sizes (Fridlind and Jacobson, 2000; Young et al., 2013; Bougiatioti et al., 2016; Fang et al., 2017), bulk pH is compared
against an aerosol liquid water-weighted pH:

$$\text{pH} = -\log_{10} \frac{1000 \left( H_{air,subgroup\ 1}^+ + H_{air,subgroup\ 2}^+ \right)}{W_{i,subgroup\ 1} + W_{i,subgroup\ 2}} \tag{4}$$

# 3. Results

## 3.1 NVCs cause discrepancy in molar ratio ($R$) predictions

*SOAS data set:* We first investigate the issue of $R$ discrepancy using PILS-IC PM$_{2.5}$ data from a 12-day period of the SOAS campaign. To test the sensitivity of ISORROPIA-II predictions to the level of NVCs, we ran the model with three different Na$^+$

concentration inputs, with all other inputs remaining the same, including total ammonium (NH$_x$ = NH$_4^+$ + NH$_3$), SO$_4^{2-}$, NO$_3^-$, and Cl$^-$. Ca$^{2+}$, Mg$^{2+}$, K$^+$ inputs were set to zero as they were mostly below detection limits. Three sets of Na$^+$ input concentrations were tested: (1) Measured PM$_{2.5}$ Na$^+$ from PILS-IC, including data below the LOD (data below LOD are clearly identified in the plots); (2) Na$^+$ determined from an ion charge balance, Na$^+$ = 2SO$_4^{2-}$ + NO$_3^-$ + Cl$^-$ − NH$_4^+$ (unit: nmol m$^{-3}$), hereafter as "inferred NVCs", and, (3) Na$^+$ = 0, which corresponds to ignoring NVCs all together.

The LOD of PILS-IC Na$^+$ was 0.07 μg m$^{-3}$, close to the average Na$^+$ concentration for the whole observation time-series. In the following, Na$^+$ data below the LOD is used in the analysis. Although in most studies data below LOD are excluded, here we include it to allow a continuum in the analysis down to zero NVCs. No obvious discontinuity in the results for data above and below the Na$^+$ LOD (e.g., see Figs. 1 and 4) and that Na$^+$ below the PILS LOD still roughly agrees with MARGA measurements

of Na$^+$, which has a lower LOD (see Fig. S2a in the supplement).

The inferred NVCs, determined from the charge balance, provide an upper limit of the NVC equivalents that can affect aerosol pH and satisfy solution electroneutrality. Overall, Na$^+$ is chosen as a proxy NVC in our dataset because in this case it constitutes most of the NVC mass and does not precipitate out of solution. The choice of Na$^+$ as a NVC proxy, although appropriate here,

may not be generally applicable, such as in regions with considerable dust contributions, as treating NVC as "equivalent Na$^+$" in the thermodynamic calculations can result in large prediction errors (e.g., Fountoukis et al., 2009). Inferred NVCs have an expected high uncertainty due to error propagation of NH$_4^+$, SO$_4^{2-}$, NO$_3^-$, and Cl$^-$ measurements (see Fig. S2b), and uncertainties in the dissociation state of sulfate (see Fig. S1 for the pH-dependence). The concentration of H$^+$ is ignored in the ion charge balance calculation for inferred NVCs, since H$^+$ is at least an order of magnitude less than the NVC ion equivalents, even for

these very low pH data points (between 0 and 2). To demonstrate this, the average ion molar concentrations in PM$_{2.5}$ were NH$_4^+$ = 35.4, SO$_4^{2-}$ = 21.1, NO$_3^-$ = 3.7, Na$^+$ = 2.9, and Cl$^-$ = 0.82 nmol m$^{-3}$ by PILS-IC, compared to ISORROPIA-predicted H$^+$ = 0.31 nmol m$^{-3}$. For the three data sets used in this study, the difference in Na$^+$ predicted from an ion balance without considering H$^+$ compared to including H$^+$ is less than 1% for SOAS and SEARCH CTR, and 6% for the WINTER study (see Fig. S3 in the supplement). In the following, we have not included H$^+$ in the ion balance. In SOAS, inferred NVCs are generally above zero

indicating a cation deficiency, but 8 out of 229 points (3% of the data) were slightly below zero. In these cases, a small positive value of 0.005 μg m$^{-3}$ was assigned to inferred NVCs. Including these data has no effect on the results because the observed $R$ was ~2. Fig. 1a shows that the inferred NVCs were always higher than measured Na$^+$. The inferred NVCs from PILS and MARGA generally agree with each other and also agree with the total NVCs from MARGA measurements before June 18, suggesting that the magnitude of inferred NVCs is reasonable (see Fig. S2). The larger differences after June 18 are likely from

difficulties in detecting NVCs in low concentrations.

The SOAS study period investigated here includes an episode of high Na$^+$ associated with a sea-salt (NaCl) aerosol event (Fig. 1a). This provided an opportunity to assess the role of NVCs on pH when concentrations were substantially above LOD. The observed Na$^+$ is mainly associated with NO$_3^-$ (Fig. 1a), and to a lesser degree with Cl$^-$. These ions are highly correlated (Na$^+$-

NO$_3^-$ r$^2$ = 0.82 and Na$^+$-Cl$^-$ r$^2$ = 0.64) and indicate some level of "chloride depletion" as the observed Cl$^-$/Na$^+$ ratio was 0.24 ±

0.16 (mol mol$^{-1}$) (mean ± SD), whereas fresh sea salts would have a molar ratio close to 1 (Tang et al., 1997). Chloride depletion occurs when an acid, such as HNO$_3$, is mixed with NaCl producing HCl that evaporates owing to its higher volatility relative to HNO$_3$ (e.g., Katoshevski et al., 1999; Fountoukis and Nenes, 2007), resulting in a loss of aerosol Cl$^-$. The chloride depletion in sea-salt aerosols during the SOAS study was discussed in detail by Bondy et al. (2017). Cl$^-$ concentrations were sufficiently

small (0.03 ± 0.04 µg m$^{-3}$, LOD = 0.01 µg m$^{-3}$) compared to the dominant and nonvolatile anion SO$_4^{2-}$, and HCl was not included in the model input, so Cl$^-$ had negligible effect on ISORROPIA predictions of pH and molar ratios. Periods where Na$^+$ was closer to typical background levels and near or below the LOD lead to similar conclusions in the following analysis.

Fig. 1 shows the effect of Na$^+$ on ISORROPIA-predicted SO$_4^{2-}$, NH$_4^+$, NH$_3$, $R$, and pH. Fig. 1b and Fig. 1e show that measured

and predicted SO$_4^{2-}$ and NH$_x$ are always identical. SO$_4^{2-}$ completely resides in the aerosol phase in all calculations. The model predicts the gas-particle partitioning by conserving NH$_x$, so the discrepancy between modeled and measured $R$ must result from variation in the model prediction of NH$_x$ partitioning. It is noteworthy that NH$_x$/SO$_4^{2-}$ is practically always above 2, indicating excess NH$_x$ compared to SO$_4^{2-}$. Under such conditions, conventional thought suggests that NH$_3$ must completely neutralize sulfate so that it can be in the form of SO$_4^{2-}$ (Kim et al., 2015; Silvern et al., 2017); this view however neglects the large

difference in volatility between SO$_4^{2-}$ and NH$_x$, which thermodynamic models consider. Because of this, PM$_{2.5}$ can remain highly acidic, with a pH between 0 and 2 (Fig. 1h), even if there is a large amount of excess NH$_x$ (Weber et al., 2016).

Comparing measured to ISORROPIA-predicted NH$_3$-NH$_4^+$ partitioning (particle phase fraction of total ammonium, ε(NH$_4^+$) = NH$_4^+$/NH$_x$) can be used to test the sensitivity to NVC input concentrations. Fig. 1g and Fig. 2a shows very good agreement

between measured and observed NH$_3$-NH$_4^+$ partitioning when measured Na$^+$ is used in the model. Using inferred NVCs generally results in an underestimation of ε(NH$_4^+$). This is consistent with using overestimated NVC levels – as the resulting pH is overestimated (Fig. 1h), which in turn shifts a fraction of the NH$_4^+$ to gas phase NH$_3$ and biases ε(NH$_4^+$) low. Zero Na$^+$ shows the opposite behavior (Fig. 1g and Fig. 2c); ε(NH$_4^+$) is over-predicted because neglecting NVCs biases pH low, driving more NH$_3$ to the particle phase and biasing ε(NH$_4^+$) high.


From the above it is clear that $R$ strongly depends on how NVCs are considered in the thermodynamic analysis. Fig. 1f shows the time series comparison between $R$ for various Na$^+$ levels included in the ISORROPIA input. Fig. 3 shows the summary statistics for various comparisons of $R$. For the SOAS analyzed time period, the predicted $R$ using measured Na$^+$ was on average 1.85 ± 0.17. As expected, predicted $R$ was significantly lower when inferred NVCs were used (mean $R$ = 1.43 ± 0.32), and highest for

zero NVC (average $R$ = 1.97 ± 0.02) in the thermodynamic analysis (see Figs. 1f and 3). The average measured $R$ was 1.70 ± 0.23 for all PILS data and 1.61 ± 0.19 excluding the points with Na$^+$ below LOD. The MARGA-derived $R$ is very similar, with measured $R$ = 1.78 ± 0.18 for all data and 1.65 ± 0.15 for periods when PILS Na$^+$ was above LOD (see Figs. S4 and S5 in the supplement). Note that CSN data used by other investigators (Silvern et al., 2017; Pye et al., 2018) have much lower $R$ (Table S1 and S2) due to a known ammonium sampling artifact (Yu et al., 2006) that cannot be accounted for, and so the dataset cannot be

used in this analysis. Together, the analysis shows that (1) when NVCs are well constrained by measurements, predicted $R$ is in close agreement with measured $R$ ($t$-test at α = 0.05 confirms no statistical difference); (2) using inferred NVCs overestimates NVC and biases $R$ low; however, the trend in predicted $R$ generally follows measured $R$ (see Fig. 1f), which argues that inferred NVCs can be a useful upper limit in NVC concentrations, when not constrained by measurements; (3) when NVC levels are zero, ISORROPIA predicts $R \sim 2$, a consequence of having the maximum possible condensation of NH$_3$ to the aerosol. Even if $R$

$\sim 2$, however, the aerosol continues to remain strongly acidic.

***Sensitivity of R and pH to NVCs and organic species:*** The results until now have clearly shown that the difference between predicted and observed $R$ for this dataset is affected by the levels of NVC. However, it is important to assess whether organic species are associated with changes in the partitioning of semivolatile inorganics and aerosol acidity (Pye et al., 2018) or other

unaccounted effects that drive the discrepancy between observed and predicted $R$. To avoid any cross correlations between organics and NVC variations, we examine how the discrepancy between observed $R$ and its theoretical limit of 2 (corresponding to when NVC=0) correlates with organic aerosol. The results in Fig. 4 clearly suggest that $\Delta R = R_{measured} - 2$ increases with measured $Na^+$ but does not depend on OA mass fraction (gray points) or OA concentration (see Fig. S6 in the supplement). This suggests that $\Delta R$ is not driven by organic aerosol effects, but instead a poor representation of NVCs in the thermodynamic model.

Fig. 4 also shows that ISORROPIA-predicted $R$ also depends on $Na^+$. Predicted $R$ with $Na^+$ in the model input minus predicted $R$ without $Na^+$ decreases with increasing measured $Na^+$ and is remarkably correlated with $Na^+$ concentration (orthogonal linear regression, $\Delta R = (-1.74 \pm 0.03) Na^+ + (0.001 \pm 0.003)$, $r^2 = 0.93$). The decreasing trend in $R$ with increasing $Na^+$ can be explained simply by the pH increasing with $Na^+$, as shown in Fig. 4c. With increasing pH, some $NH_4^+$ shifts to the gas phase $NH_3$, resulting in lower $NH_4^+$ and lower $R$.

From the regression slope, for the SOAS measurement period analyzed, an average measured $Na^+$ level of 0.07 µg m$^{-3}$ (very small NVC concentrations) decreases $R$ by 0.12 units. For a $Na^+$ level of 0.3 µg m$^{-3}$, $R$ decreases by 0.5 units, from $R = 2$ (i.e., no NVC) to $R = 1.5$ (i.e., with NVC). Thus, $\Delta R$ is highly correlated and sensitive to $Na^+$, both of which is not seen for the organic aerosol mass fraction. Mass fraction can be used as a proxy for organic film thickness too, given that the maximum possible

thickness (and delay) associated for an organic film scales with (organic volume)$^{1/3}$ or (organic mass)$^{1/3}$.

In comparison to $R$, pH is less sensitive to inclusion of $Na^+$, or other NVCs in general. $\Delta$pH is only 0.09 for the average $Na^+$ level of 0.07 µg m$^{-3}$ and increases to 0.38 at 0.3 µg m$^{-3}$ $Na^+$ (Fig. 4b). The magnitude of $\Delta$pH is relatively small and consistent with our previous studies where we investigated the effects of sea-salt on pH (Guo et al., 2016; Weber et al., 2016). $\Delta$pH would be higher

in regions with more abundant NVC. For instance, a $\Delta$pH of 0.8 unit was found in Pasadena, CA, where the average $PM_{2.5}$ $Na^+$ mass was 0.77 µg m$^{-3}$ (Guo et al., 2017). Differences in sensitivity of $R$ and pH to $Na^+$ from the slope of the linear regressions (Fig. 4c) are 1.74 ($\Delta R$-$Na^+$ slope), and 1.2 ($\Delta$pH-$Na^+$), respectively. NVC effects on $R$ and pH are studied next for a very different aerosol data set.

***WINTER data set:*** The $R$ discrepancy is investigated for a different season and a larger and different region by repeating the analysis using the WINTER study data set collected from the NSF C-130 research aircraft during wintertime. The aerosol inorganic composition data used in the analysis is from an AMS and is $PM_1$. In this study, NVCs were generally higher than those measured during SOAS, especially when the aircraft sampled near coastlines (e.g., $PM_1$ $Na^+ = 0.23$ µg m$^{-3}$). Also, $PM_1$ nitrate was comparable to sulfate, largely owing to lower temperatures ($NO_3^-$ 13 nmol m$^{-3}$ vs. $SO_4^{2-}$ 11 nmol m$^{-3}$) (Guo et al.,

2016). Therefore, $R_{SO4}$ was calculated instead of $R$.

The base case input to ISORROPIA-II in this analysis included $NH_4^+$, $SO_4^{2-}$, and total nitrate ($NO_3^-$ + $HNO_3$). ($NH_3$ should be included to determine $NH_x$ for input, but was not measured. It was found to have a small effect on predicted pH; e.g., ~0.2 higher pH when including an $NH_3$ concentration of 0.10 µg m$^{-3}$ typical of the Eastern US levels, and estimated from an order-of-

magnitude iteration method (Guo et al., 2016)). Fig. 5a shows that ISORROPIA over-predicted $R_{SO4}$ for the base case (i.e., when

cations are not included) and that this deviation increases as molar ratios approach 2 when inferred NVCs are smaller. Again, NVC concentrations were determined as NVCs = $Na^+$ = $2SO_4^{2-}$ + $NO_3^-$ − $NH_4^+$ (unit: nmol $m^{-3}$) where all NVC are assumed to be $Na^+$. (Note that the predicted $R_{SO4}$ should be biased low since $NH_4^+$ was under-predicted due to lack of $NH_3$ data, resulting in some fraction of input particle phase $NH_4^+$ repartitioned in the model to the gas phase, thus the deviation is even worse than

shown). Fig. 5a shows that $R_{SO4}$ is highly sensitive to lack of inclusion of NVCs when their concentrations are very low. However, when concentrations of NVC reach zero, predicted and measured $R_{SO4}$ converge to the expected value of 2 (dark blue symbols in Fig. 5a). Interestingly, as predicted NVCs increase, predicted and measured $R_{SO4}$ converge to zero, because NVC progressively dominate the cations, and force $NH_4^+$ to evaporate. On average, predicted $R_{SO4}$ was $1.68 \pm 0.51$ versus the measured value of $1.47 \pm 0.43$.

In contrast to ISORROPIA-predicted $R_{SO4}$ without NVCs, including NVCs (inferred NVCs) brings predicted and measured ammonium-sulfate molar ratios into agreement (Fig. 5b). Including or excluding $H^+$ in the $Na^+$ calculation produces similar results (Fig. S7). Findings based on other NVCs are shown in supplemental Fig. S8. $K^+$ and $Mg^{2+}$ work similarly to $Na^+$, while $Ca^{2+}$ can precipitate sulfate in the form of $CaSO_4$ and so cannot be used. For $Na^+$, the linear regression result is $R_{SO4,predicted}$ =

$(1.089 \pm 0.001)$ $R_{SO4,measured}$ − $(0.166 \pm 0.002)$, $r^2 = 0.996$. As found for the SOAS data set, again, the molar ratio bias from the thermodynamic model appears to result from not including small amounts of NVC (e.g., in this case on average 0.15 µg $m^{-3}$ $Na^+$ or 0.26 µg $m^{-3}$ $K^+$). The average amount of inferred $PM_1$ $Na^+$ from the ion charge balance was 0.15 µg $m^{-3}$, in this case smaller than what was measured offline 0.23 µg $m^{-3}$ (Guo et al., 2016) (In comparison, inferred NVCs are higher than measured $Na^+$ in the SOAS case). The analysis using measured $PM_1$ $Na^+$ results in highly scattered data due to the high sensitivities of $R_{SO4}$ to

NVC and the significant $Na^+$ measurement uncertainty at these low levels given the analytical sampling method used in this study (i.e., offline analysis).

### 3.2 Implications of not including NVC on predicting gas-particle partitioning and historical trends in molar ratios

*Sensitivity of semi-volatile species partitioning to NVCs:* In our datasets, inferred NVCs group all NVCs, including $K^+$ and $Mg^{2+}$, into one species and are upper limits of the NVCs because it assumes complete dissociation of all dissolved ionic species. For example, 10% of the total sulfate is predicted to be $HSO_4^-$ and the rest as $SO_4^{2-}$ for the SOAS average pH ~ 1 (see Fig. S1). Additional errors can occur if other ions are also missing, but this approach satisfies electroneutrality. Comparing ISORROPIA predictions that includes the other major species, an inferred NVCs input versus $Na^+$ = 0 input results in an average increase in

pH by 0.32 for SOAS and 0.49 for WINTER, respectively. Even though the effect of NVC on pH may appear relatively small, the impact on predicted partitioning of a semivolatile species can be significant due to the highly non-linear response of $NH_3$-$NH_4^+$ or $HNO_3$-$NO_3^-$ partitioning to pH (i.e., S curve). For example, as shown in supplemental Fig. S9, a 0.3 unit pH bias in SOAS campaign could cause ~ 20% bias in $\varepsilon(NH_4^+)$ or $\varepsilon(NO_3^-)$ prediction when $\varepsilon(NH_4^+)$ or $\varepsilon(NO_3^-)$ = 50%, or no bias at all when the species are completely in one phase, $\varepsilon(NH_4^+)$ or $\varepsilon(NO_3^-)$ = 0% or 100%. For the WINTER study, a 0.5 pH bias causes

up to 30% bias in $\varepsilon(NH_4^+)$ or $\varepsilon(NO_3^-)$. These partitioning biases may constitute a significant source of bias for aerosol nitrate formation, especially if the total nitrate present in the gas-aerosol system is significant. In fact, the bias from the NVC may completely change the predicted response of nitrate to aerosol emissions and lead to errors in the predicted vs. observed trends in pH, such as was seen in the southeastern US (Vasilakos et al., 2018).

***Effect of NVCs in trends in pH and R in the southeastern US:*** The organic aerosol impacts on $NH_3$ equilibration (Silvern et al., 2017) was postulated to address the decreasing trend in $R$ in the southeastern US despite the substantial drop in sulfate. Weber et al., (2016) also noted this and proposed that it could be explained by $NH_4^+$ volatility. However, the thermodynamic model predictions of $R_{SO4}$ in that study did not find a comparable decreasing $R_{SO4}$ rate with time (see Fig. 6a), since the SOAS study

mean PILS-IC $Na^+$ concentration of 0.03 µg m$^{-3}$ was applied to all historical data. With this constant input of 0.03 µg m$^{-3}$, predicted $R_{SO4}$ was nearly constant at ~2 for the input $SO_4^{2-}$ range (Fig. 6a) and would only rapidly decrease below 1 µg m$^{-3}$ $SO_4^{2-}$ (See Fig. 2b in Weber et al., (2016)). Repeating the calculations using $Na^+$ inferred from the ion charge balance of $Na^+$-$NH_4^+$-$SO_4^{2-}$-$NO_3^-$, determined for each daily data point in the historical data set, results in good agreement between observed and ISORROPIA-predicted $R_{SO4}$ (Fig. 6 & Fig. S10). It also predicts a decreasing $R_{SO4}$ rate of –0.017 yr$^{-1}$, fairly close to the

measured rate at the SOAS site (Centerville, AL) of –0.021 yr$^{-1}$ (see Fig. 6a), and in the range of the $R_{SO4}$ trend of –0.01 to –0.03 yr$^{-1}$ reported by Hidy et al. (2014) for the SEARCH sites throughout the southeast. In contrast, using these different $Na^+$ input concentrations did not change the trends in ISORROPIA-predicted pH; in both cases, it remained relatively constant (Fig. 6b), but as expected the pH was slightly higher with higher input $Na^+$ concentrations. Thus, including daily estimates of NVC in ISORROPIA, the conclusion that $PM_{2.5}$ pH has remained largely constant over the last 15 years remains, but the unexpected

decreasing $R_{SO4}$ trend can be accounted for only with including NVC effects and $NH_4^+$ volatility. These observations can all be explained by volatility of $NH_4^+$ (Weber et al., 2016), without need to invoke organic effects on the ammonia partitioning.

## 4. Discussion

***Internal vs External Mixtures:*** Our thermodynamic analysis up to this point has been based on the assumption that all ions were internally mixed (e.g., bulk $PM_{2.5}$ or $PM_1$). Although over time, gas-particle and particle-particle interactions will lead to

complete internally mixed systems (Seinfeld and Pandis, 2016), aerosol near their source regions tend to be externally mixed. Typical ambient conditions can be expected to exist somewhere between these two extreme cases (Bondy et al., 2018) owing to chemistry, coagulation, cloud processing, dilution, and gas-to-particle mass transfer (Zaveri et al., 2010). We address this here by studying how the conclusions described above are affected by the degree of mixing of NVCs with ammonium and sulfate – as the other species, being semivolatile, quickly equilibrate.


$PM_{2.5}$ $Na^+$, $K^+$, $Ca^{2+}$, and $Mg^{2+}$ from sea-salt (or dust) are often not well mixed with ammonium and sulfate because of their different sources and sizes. NVC from sea-salt and dust are largely produced by mechanical means and so are mainly in the coarse mode, with a tail extending into the fine mode (Whitby, 1978). Biomass burning and biogenic $K^+$ is emitted into the fine mode (Bougiatioti et al., In review), however, ammonium and sulfate are formed through gas-phase processes and mostly reside

in the accumulation mode (e.g., Whitby, 1978; Seinfeld and Pandis, 2016). For the SOAS PILS-IC data set, $NH_4^+$ and $SO_4^{2-}$ were highly correlated ($r^2 = 0.88$), but $NH_4^+$ and $Na^+$ ($r^2 = 0.07$) or $SO_4^{2-}$ and $Na^+$ ($r^2 = 0.17$) were not. In contrast, $PM_{2.5}$ $Na^+$ and $NO_3^-$ ($r^2 = 0.82$) or $Na^+$ and $Cl^-$ ($r^2 = 0.64$) were highly correlated, consistent with internal mixing of most $Na^+$, $NO_3^-$, and $Cl^-$ ions, leading to depletion of some $Cl^-$ through evaporation of HCl (e.g., Katoshevski et al., 1997; Seinfeld and Pandis, 2016). Rapid scavenging of $HNO_3$ by sea salt aerosols is well established (Hanisch and Crowley, 2001; Meskhidze et al., 2005), with

equilibrating time scales 3-10 hours for $HNO_3$ uptake by 1-3 µm sea spray aerosols (Meng and Seinfeld, 1996; Fridlind and Jacobson, 2000), and subsequent evaporation of HCl.

NVCs can also be associated with small amounts of sulfate. For example, sea salt aerosols are largely composed of NaCl but also include sulfate, approximately 8% (g g$^{-1}$) of all ions (~25% $SO_4^{2-}$/$Na^+$ mass ratio) (DOE, 1994). In addition, sulfur enrichment and chloride depletion in aged sea salt aerosols are possible by uptake of $H_2SO_4$ or oxidation of dissolved $SO_2$ by $O_3$ (McInnes et al., 1994; O'Dowd et al., 1997). These secondary sulfates are normally referred as non-sea-salt sulfates, to be distinguished from

sea-salt sulfate that is naturally in sea waters (Tang et al., 1997). Many studies have reported sulfate-containing sea salt aerosols with some degrees of internal mixing (Andreae et al., 1986; McInnes et al., 1994; Murphy et al., 1998; Laskin et al., 2002; Bondy et al., 2018). In summary, a realistic external mixing state of the SOAS fine particles is that most of $NH_4^+$ and $SO_4^{2-}$ are in $PM_1$, whereas $Na^+$ with associated anions ($NO_3^-$ and $Cl^-$) and at least small amounts of $NH_4^+$ and $SO_4^{2-}$ are associated in $PM_{1-2.5}$ (particles with sizes 1-2.5 µm). This is consistent with the single particle mixing state observations by Bondy et al. (2018) from

the SOAS study. The interactions between aerosols with gases are illustrated in Fig. 7a. Particle size distributions measured in the southeast US also support these types of particle mixing state (Fang et al., 2017).

*Explanation for role of NVCs on R based on bulk (internal mixture) analysis:* Recapping the bulk analysis above where ions are all assumed to by internally mixed, we have shown that the observations relating $R$ to NVCs, and deviations in $R$ between

models and observations can be readily explained. First, when NVC such as $Na^+$ are present in the ambient aerosol and not included in the thermodynamic model, but some fraction of the associated anion pair is, the thermodynamic model will predict higher $NH_4^+$ than observed because the model will partition greater levels of available semivolatile cations (i.e., $NH_3$) to the particle phase ($NH_4^+$) to conserve $NH_x$ and make up for the missing NVCs. This leads to a predicted $R$ near 2. The trends in measured $R$ with measured $Na^+$ are also expected. As noted before, measured $R$ becomes increasingly less than 2 as measured

$Na^+$ increases because at higher $Na^+$, bulk aerosol pH increases (Fig. 3c), resulting in lower $\varepsilon(NH_4^+)$ (see $NH_4^+$ S curve in supplemental Fig. S9), shifting $NH_4^+$ to gas phase $NH_3$. Other NVCs have similar effects as $Na^+$, as long as soluble forms of the salts are observed (e.g., $NaNO_3$, $Na_2SO_4$, $KNO_3$, $K_2SO_4$, $Ca(NO_3)_2$, $Mg(NO_3)_2$). We have shown with this bulk analysis that accurately including NVCs in the thermodynamic analysis appears to largely resolves the disparity in predicted and measured $R$ for the data sets we analyzed. But the bulk analysis is only an approximation of the actual aerosol mixing state. We next test if

assuming an internal mixture will roughly represent the behavior of externally mixed aerosols in terms of the effect of NVCs on $R$, pH, and partitioning of semivolatile species? To assess this, we consider the behavior of external mixing cases.

*Explanation for the role of NVCs on R based on external mixture analysis:* An extreme (and unrealistic at the timescale of aerosol lifetime (Zaveri et al., 2010)) external mixture is where $PM_1$ is composed of all the measured $NH_4^+$, $SO_4^{2-}$ and $PM_{1-2.5}$ is

composed of all the measured $Na^+$ (all NVCs), $NO_3^-$, and $Cl^-$. $NH_3$, $HNO_3$, $HCl$, and $H_2O$ (water vapor) can still equilibrate between these externally mixed particle types (see Fig. 7a), given the relatively short equilibrating time scales for these sizes of particles (Dassios and Pandis, 1999; Cruz et al., 2000; Fountoukis et al., 2009). As Fig. 7b shows, for the extreme external mixing case (i.e. 0% sulfate in $PM_{1-2.5}$), predicted $R$, combined from $PM_1$ and $PM_{1-2.5}$, is close to 2, deviating from the lower predicted $R$ of $1.66 \pm 0.13$ from the internal mixture. This is due to the vastly different pH of $PM_1$ (0.6) and $PM_{1-2.5}$ (4.1) (Fig.

7c), where all $NH_4^+$ is predicted to be in $PM_1$, and all $NO_3^-$ is predicted to be in $PM_{1-2.5}$.

For more realistic mixing cases, where some fraction of the sulfate is mixed with NVCs (Bondy et al., 2018), the combined $R$ of the external mixture decreases rapidly as more $SO_4^{2-}$ is mixed with $Na^+$ in $PM_{1-2.5}$. At ~20% $SO_4^{2-}$ fraction in $PM_{1-2.5}$, the average levels of predicted $R$ start to converge between external and internal mixtures (Fig. 7b). The difference in pH between $PM_1$ and

$PM_{1-2.5}$ is also reduced to within one pH unit (Fig. 7c). With these small differences in pH, $NH_4^+$ can condense on both

externally-mixed aerosol groups. For example, $PM_1$ and $PM_{1-2.5}$ $NH_4^+$ are predicted to be 0.67 µg m$^{-3}$ and 0.04 µg m$^{-3}$, respectively (equal to the sum of the measured $PM_{2.5}$ $NH_4^+$ of 0.71 µg m$^{-3}$). From this analysis, based only on data when $Na^+$ was above the LOD, predicted $R$ for the bulk and external mixture are the same when on average $18 \pm 7\%$ (by mass) of the $PM_{2.5}$ $SO_4^{2-}$ is in the $PM_{1-2.5}$ size range (i.e., mixed with $Na^+$). This is comparable to inferences of mixing based on size-resolved

aerosol measurements in the southeast (e.g., Fang et al., (2017) shows ~30% $PM_{2.5}$ $SO_4^{2-}$ mass in $PM_{1-2.5}$). Less internal mixing of $SO_4^{2-}$ with $Na^+$ is needed when $Na^+$ concentrations are lower. For the SOAS 12-day $Na^+$ average level of 0.07 µg m$^{-3}$, only 5% of the $SO_4^{2-}$ (by mass) when mixed with $Na^+$ produces the same results as the bulk totally internal mixture case (see supplemental Fig. S11). Note that higher $Na^+$ concentrations generally require more $SO_4^{2-}$ to obtain agreement in $R$ between external and internal mixtures (scatter plots are shown as supplemental Fig. S12).

The difference between the internally and externally mixed system is not as great as may be expected, especially for particle pH and liquid water ($W_i$) (Fig. 7c and Fig. 7d). Since liquid water levels are determined as the sum of the water associated with the various salts, the bulk liquid water generally equals the sum of the two externally mixed liquid water concentrations, based on the Zdanovskii-Stokes-Robinson (ZSR) relationship (Zdanovskii, 1936; Stokes and Robinson, 1966). Because the most

hygroscopic salts (i.e., $NH_4^+$ and $SO_4^{2-}$; $NO_3^-$ concentrations are low) are in $PM_1$, $PM_1$ liquid water dominates over $PM_{1-2.5}$, making the combined pH of the external mixture nearly identical to $PM_1$ pH (see Equation 4 for combined pH calculation). The combined pH of the external mixture is also similar to that of internal mixture, regardless of the $SO_4^{2-}$ fractions (see Fig. 7c).

## 5. Summary

We have shown that including NVCs in the thermodynamic model largely resolves the ammonium-sulfate molar ratio ($R =$
$NH_4^+/SO_4^{2-}$) discrepancy, based on our data sets. (We have not utilized the CSN data set as other researchers have due to a large low bias in $R$.) Since only small amounts of NVC can significantly affect $R$, measurement limitations, such as high NVC LODs or NVCs not measured at all (e.g., AMS measurements), can lead to substantial differences in observed and thermodynamic model predicted $R$. We show that this bias in $R$ (ISORROPIA-predicted $R$ with $Na^+$ minus ISORROPIA-predicted $R$ without $Na^+$) is correlated with and highly sensitive to measured $Na^+$, but not correlated with organic aerosol mass or mass fraction.
Similarly, the difference in measured $R$ from a ratio of 2 (2 minus observed $R$) is correlated to measured $Na^+$ (NVCs) and not correlated with organic aerosol mass or mass fraction. If organic films were limiting mass transfer, the discrepancy in $R$ should worsen as the films become thicker. We find the opposite. Furthermore, ISORROPIA-predicted $NH_3$-$NH_4^+$ partitioning (with measured $Na^+$ as input) agrees well with the observation, showing an equilibrium state of the partitioning and no significant $NH_3$ mass transfer limit caused by organic film. These results provide evidence for the role of NVCs, but not bulk organic aerosol
species or organic films in the molar ratio discrepancy observed in the southeastern US.

Excluding minor amounts of fine mode NVC in thermodynamic calculations results in predicted $R$ near 2, which is generally higher than observed values. This results from the model criteria for aerosol electrical neutrality and because semivolatile $NH_4^+$ has to be increased to compensate the missing NVCs. Less absolute discrepancy is associated with predicted particle pH with or
without NVC because pH is on a logarithmic scale of $H_{aq}^+$ and the range of pH is larger than that of $R$ (or $R_{SO4}$) in the eastern US. However, neglecting NVC can induce pH biases that could result in significant partitioning errors for semivolatile species like ammonium, nitrate, chloride, and even organic acids, under certain conditions. Because NVCs are often minor constituents of fine particles, especially for submicron particles, implying low ambient concentrations and high measurement uncertainties,

assessing thermodynamic model predictions through molar ratios is problematic. If NVCs were not measured or significantly below the measurement LOD, an ion charge balance could be used to infer an upper limit on NVC concentrations, but addition of measurement uncertainties can lead to uncertain results. Note that the ion charge balance on its own generally cannot be used to infer $H^+$ since the $H^+$ concentrations are generally very low, even at the low pH of the southeastern US aerosols, and the

dissociation states of acids must be known (e.g., proportions of $HSO_4^-$ and $SO_4^{2-}$), which requires a full thermodynamic analysis.

A motivation for the organic effects on ammonia partitioning (Silvern et al., 2017) was the observed $R_{SO4}$ decreasing trend over the past 15 years in the southeastern US. Fully considering NVCs does not change the finding of nearly constant fine particle pH in the southeast (summertime) despite the large sulfate reductions in the past 15 years, but it does now lead to agreement with the

observed $R_{SO4}$ decreasing trend. Although the analysis was performed assuming internal mixtures of aerosol constituents, since only bulk $PM_{2.5}$ composition data were available, we show that external mixtures of NVCs and sulfate produce similar molar ratios, with the requirement that only small amounts of sulfate are needed to be mixed with the NVC-rich particle, qualitatively consistent with the particle mixing state measured for the SOAS study reported by Bondy et al. (2018). In contrast to molar ratio, the average pH for externally mixed aerosol is not sensitive to the mixing fraction of $SO_4^{2-}$ and $Na^+$.-Further assessments on

possible effects of organic species on semi-volatile partitioning of inorganic species should be carried out, especially for regions that are chemically different from the eastern US conditions evaluated in this study.

**Acknowledgements.** This work was supported by the National Science Foundation (NSF) under grant AGS-1360730. The WINTER data is provided by NCAR/EOL under sponsorship of the National Science Foundation (http://data.eol.ucar.edu/). We also acknowledge support from an EPA STAR grant and the European Research Council Consolidator Grant 726165 - PyroTRACH.

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

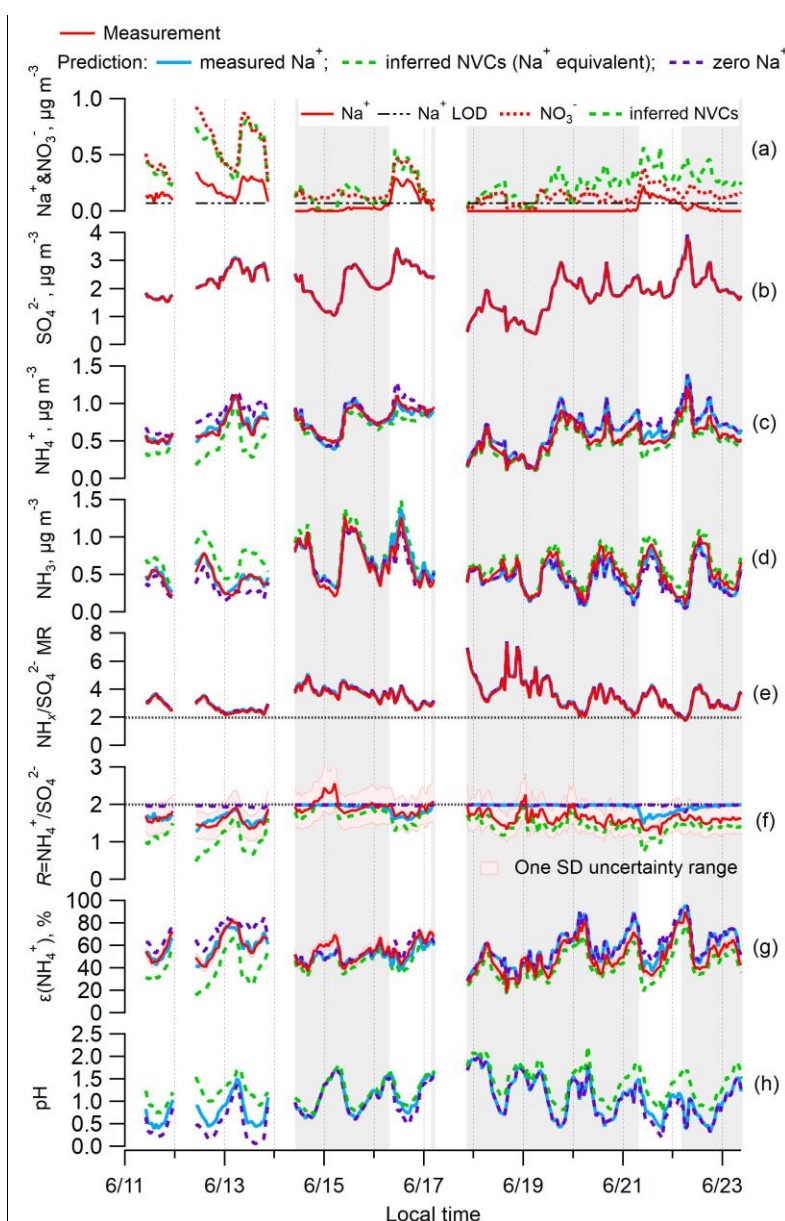

**Figure 1.** Time series of various measured and ISORROPIA-predicted parameters and PM$_{2.5}$ component concentrations for the SOAS study. Specific plots are as follows: (a) Na$^+$ and NO$_3^-$, (b) SO$_4^{2-}$, (c) NH$_4^+$, (d) NH$_3$, (e) total ammonium (NH$_x$ = NH$_4^+$ + NH$_3$) to sulfate molar ratio (NH$_x$/SO$_4^{2-}$), (f) ammonium-sulfate ratio ($R$ = NH$_4^+$/SO$_4^{2-}$), (g) particle-phase fractions of total

5  ammonium, ε(NH$_4^+$), and (h) particle pH. ISORROPIA-predicted results for the base case and three different Na$^+$ inputs are shown: measured Na$^+$ in blue, inferred nonvolatile cations (NVCs) from an ion charge balance (where the overall NVCs are represented here by Na$^+$; Na$^+$ = 2SO$_4^{2-}$ + NO$_3^-$ + Cl$^-$ − NH$_4^+$, μmol m$^{-3}$) in green, and zero Na$^+$ in purple. The periods with measured Na$^+$ below LOD are marked with grey backgrounds.

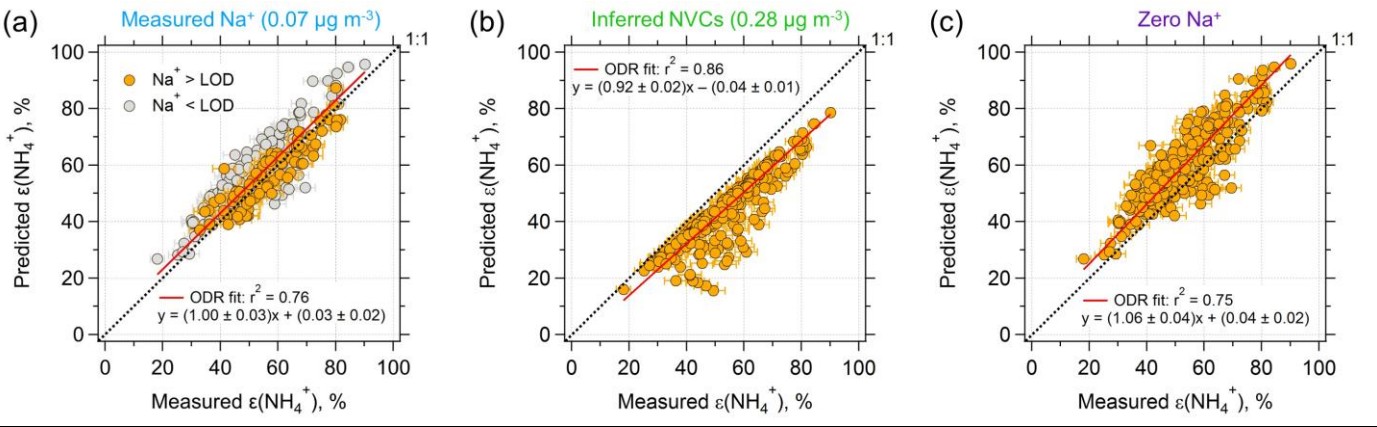

**Figure 2.** Comparisons of predicted and measured particle phase fractions of total ammonium, $\varepsilon(NH_4^+) = NH_4^+/NH_x$. (a) The model prediction is based on an ISORROPIA input of measured $Na^+$, $NH_x$, $SO_4^{2-}$, $NO_3^-$, $Cl^-$. (b) Same model input, but NVCs (represented by $Na^+$) is inferred from an ion charge balance and (c) $Na^+$ is set to zero. Orthogonal distance regression (ODR) fits are shown and uncertainties in the fits are one standard deviation (SD) (The ODR fit in (a) is based on all the data points). Uncertainty of measured $\varepsilon(NH_4^+)$ is derived from error propagation of $NH_4^+$ (20%) and $NH_3$ (6.8%) measurements. Best agreement is achieved by using measured $Na^+$ as input.

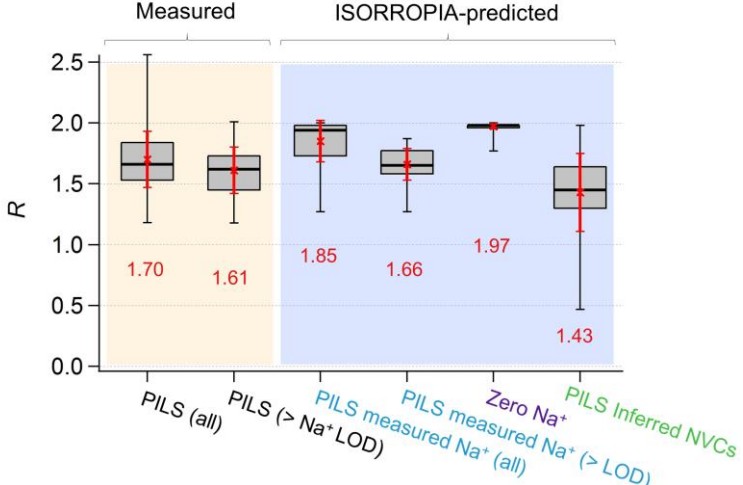

**Figure 3.** Comparisons of PM$_{2.5}$ ammonium-sulfate molar ratios (*R*) between measurements and ISORROPIA-predictions for the base case but with differing Na$^+$ inputs. Data are from the SOAS study. Red numbers are the means and red error bars are one SD. Standard box-whisker plots are shown, with 100% and 0% data indicated by black error bars. Top and bottom of box are the interquartile ranges (75% and 25%) centered around the median value (50%). Comparisons include all data and periods when measured Na$^+$ > LOD of 0.07 µg m$^{-3}$.

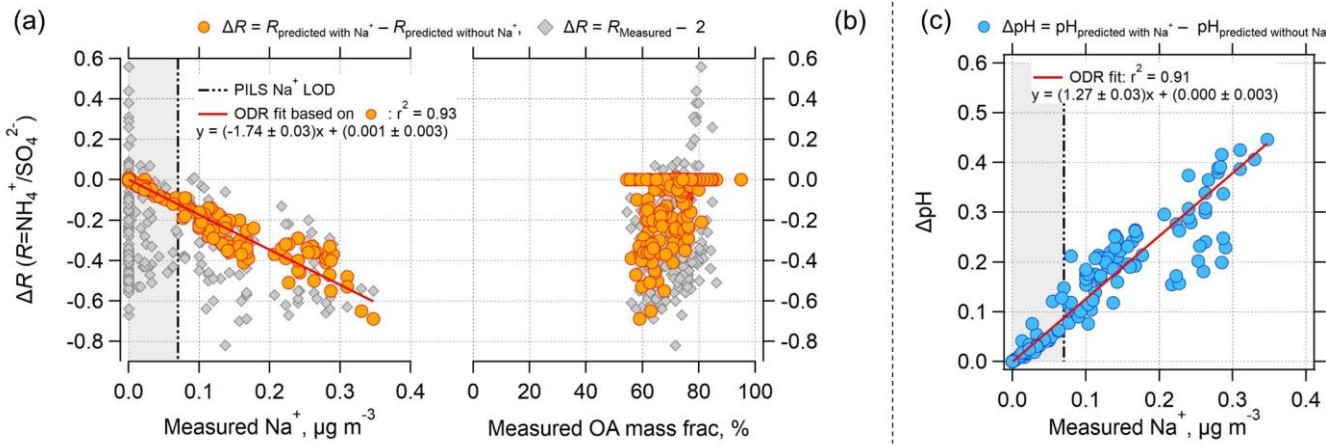

**Figure 4.** Effect of nonvolatile cations (NVC) on the PM$_{2.5}$ ammonium-sulfate molar ratios ($R$) and pH as a function of measured Na$^+$ concentration and organic aerosol (OA) mass fractions for the SOAS data set studied. Plot (a) is $\Delta R$ versus measured Na$^+$,

5  (b) $\Delta R$ versus measured OA mass fraction (OA mass divided total particle mass reported from AMS), and (c) $\Delta$pH versus measured Na$^+$. Grey diamonds in plots (a) and (b) are for $\Delta R$ equal to the measured $R$ minus 2. Orange circular points are for $\Delta R$ equal to ISORROPIA-predicted $R$ with measured Na$^+$ included in the model input minus ISORROPIA-predicted $R$ without Na$^+$ in the model input. $\Delta$pH in plot (c) is determined in a similar way. $\Delta R$ is negative since including Na$^+$ in the thermodynamic model results in $R$ lower than 2, whereas not including Na$^+$ results in an $R$ close to 2 (see Fig. 3). ODR fits are shown and

10  uncertainties in the fits are one standard deviation. A plot similar to (b), but versus OA mass concentration can be found as Fig. S6. The vertical dotted line is the Na$^+$ LOD of 0.07 µg m$^{-3}$. Regions where Na$^+$ is below LOD are marked with grey backgrounds.

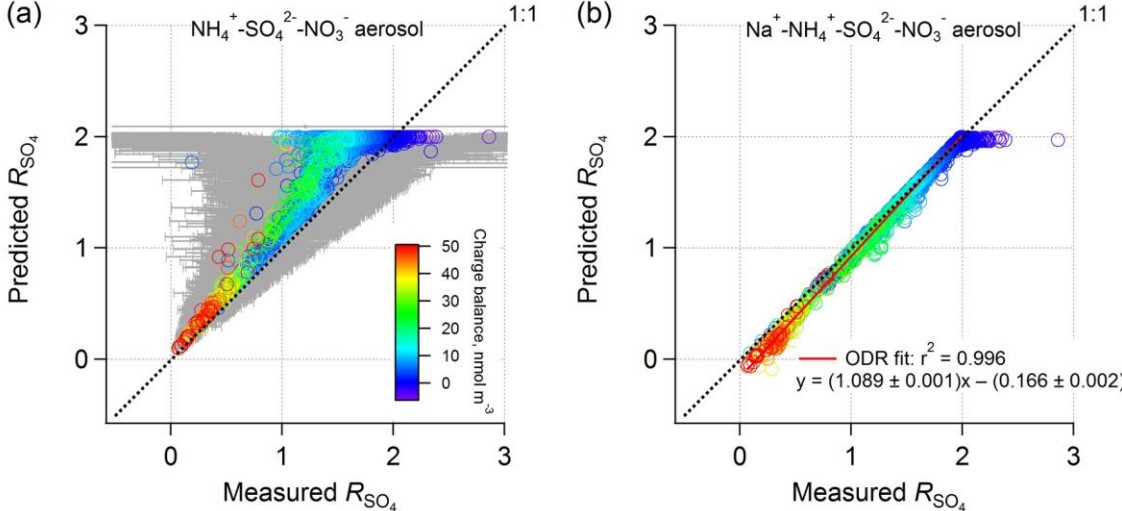

**Figure 5.** Comparison between PM$_1$ ISORROPIA-predicted $R_{SO4}$ and AMS-measured $R_{SO4}$ ($R_{SO4}$ = (NH$_4^+$ − NO$_3^-$)/SO$_4^{2-}$) (mol mol$^{-1}$), where the ISORROPIA-prediction is based on (a) NH$_4^+$, SO$_4^{2-}$, NO$_3^-$ aerosol and (b) Na$^+$, NH$_4^+$, SO$_4^{2-}$, NO$_3^-$ aerosol, and both include HNO$_3$ to calculate total nitrate for the model input. All measurement data are from the WINTER study. NVCs were determined by an ion charge balance with the predicted molar concentration shown by symbol color. Error bars were determined by propagated uncertainties for $R_{SO4}$ based on a 35% AMS measurement uncertainty for NH$_4^+$, SO$_4^{2-}$, and NO$_3^-$ (Bahreini et al., 2009). Error bars are larger at higher ratios due to subtraction of higher concentrations of nitrate and so subject to greater measurement error. Data points with low SO$_4^{2-}$ levels (< 0.2 µg m$^{-3}$; 9% of the total points) were excluded due to high uncertainties.

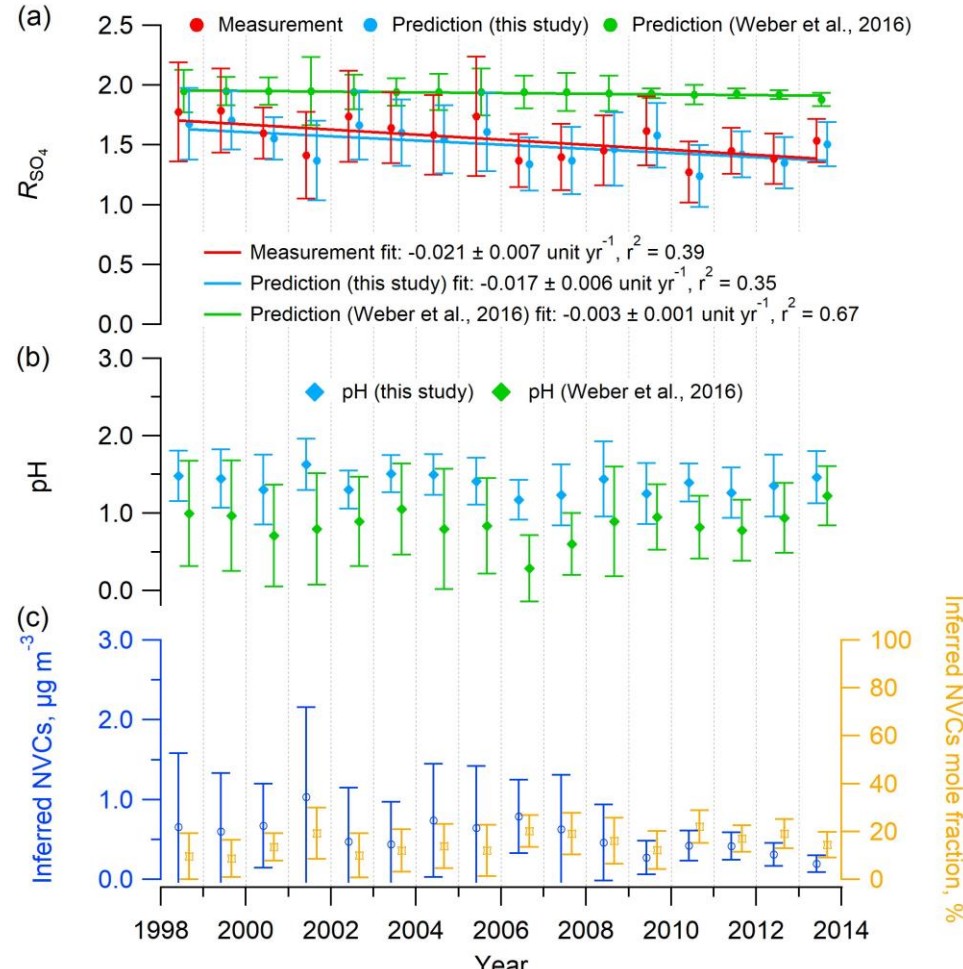

**Figure 6.** Mean summer (June–August) trends in (a) measured and predicted $R_{SO4}$, (b) predicted $PM_{2.5}$ pH, and (c) inferred NVCs concentration and mole fraction at the SEARCH-CTR site. $Na^+$ was inferred from an ion charge balance of $Na^+$-$NH_4^+$-$SO_4^{2-}$-$NO_3^-$. ISORROPIA inputs include the measured $PM_{2.5}$ composition ($NH_4^+$, $SO_4^{2-}$, $NO_3^-$) and meteorological data (RH, T) at

5  CTR. In all cases, $R_{SO4}$ and pH were estimated with ISORROPIA-II run in forward mode with an assumed $NH_3$ level of 0.36 μg m$^{-3}$, the mean concentration from the SOAS study (CTR site, summer 2013), due to limited $NH_3$ data before 2008. Historical $NH_3$ mean summer concentrations at CTR were 0.2 μg m$^{-3}$ (2004-2007) (Blanchard et al., 2013) and 0.23 ± 0.14 μg m$^{-3}$ (2008-2013) (Weber et al., 2016). Error bars represent daily data ranges (SD). Linear regression fits are shown and uncertainties in the fits are one SD. 41 data points out of 609 (7%) with observed daily mean $R_{SO4}$ above 3 were considered outliers and not shown

10  (if included the fit slope is $-0.023 \pm 0.008$ unit yr$^{-1}$).

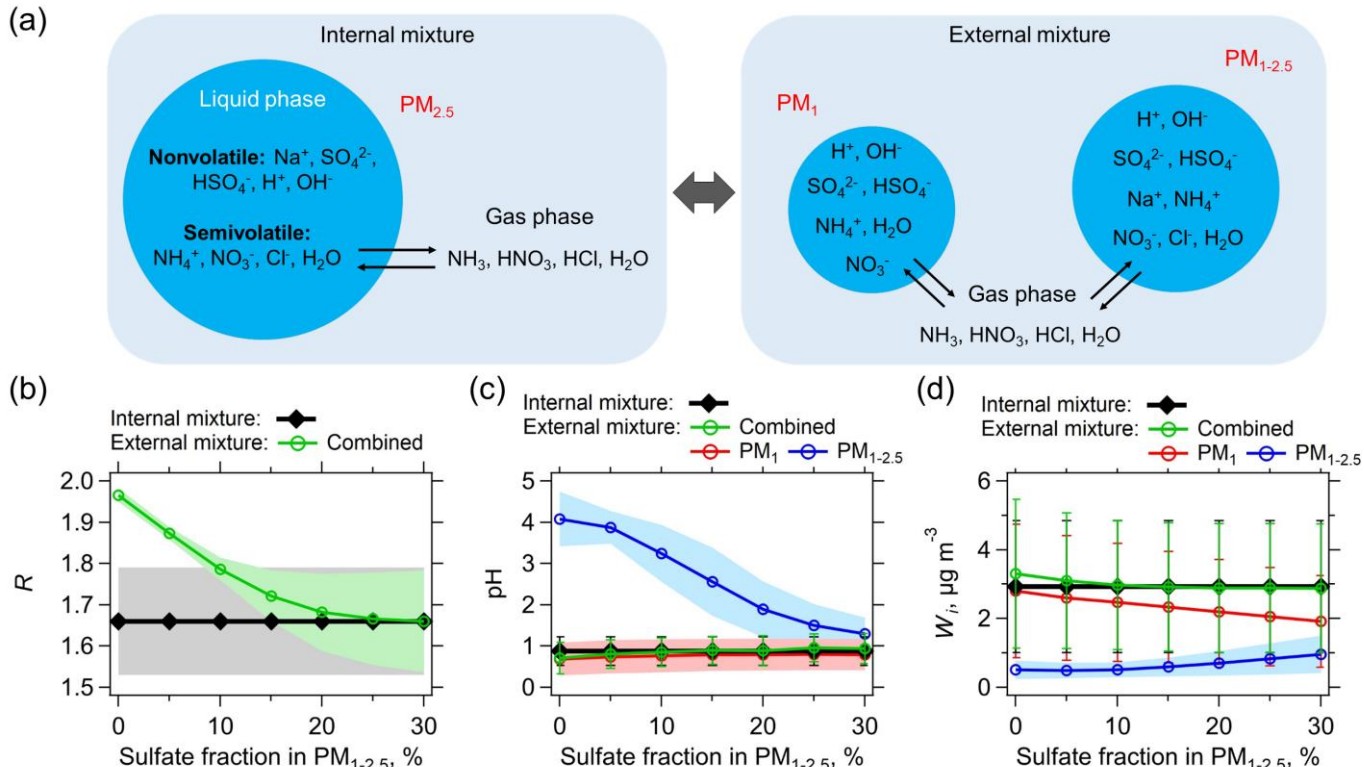

**Figure 7.** (a) Schematic of assumed internally and externally mixed aerosols. NVCs (here represented by $Na^+$) are all assumed in $PM_{1-2.5}$ for the external mixing case. The two externally mixed aerosol groups ($PM_1$ and $PM_{1-2.5}$) are in equilibrium with the same gases. The internal mixed case has bulk $PM_{2.5}$ compositions ($PM_1 + PM_{1-2.5}$) together with gases as model input. The predicted

5 molar ratio ($R$), pH, and liquid water ($W_i$) of the internally and externally mixed aerosols are summarized in (b), (c), and (d), respectively. The x-axis is the sulfate (mass) fraction assumed in $PM_{1-2.5}$, with the remaining sulfate in $PM_1$. For the analysis shown here only data for which measured $Na^+$ was above the LOD are utilized. Lower $Na^+$ concentrations require smaller fractions of $SO_4^{2-}$ in the $PM_{1-2.5}$ range for agreement with the bulk analysis (e.g., 5% for PILS-IC $Na^+$ LOD of 0.07 µg m$^{-3}$). Standard deviations of the data are shown as error bars or shaded zones.

