# Peer review of "The underappreciated role of nonvolatile cations on aerosol ammonium-sulfate molar ratios"

_Atmospheric Chemistry and Physics, 2017_

## Short Comment (SC1) · 1 Sep 2017

Readers may also be interested in a related discussion on Pye, H. O. T., Zuend, A., Fry, J. L., Isaacman-VanWertz, G., Capps, S. L., Appel, K. W., Foroutan, H., Xu, L., Ng, N. L., and Goldstein, A. H.: Coupling of organic and inorganic aerosol systems and the effect on gas-particle partitioning in the southeastern United States, Atmos. Chem. Phys. Discuss., https://doi.org/10.5194/acp-2017-623, in review, 2017.

Discussion comments available here: https://www.atmos-chem-phys-discuss.net/acp-2017-623/#discussion

---

## Referee Comment (RC1) · Anonymous Referee #1 · 8 Sep 2017

Large discrepancy existed between observed and model-predicted molar ratios of $NH_4^+/SO_4^{2-}$ (R) for the southeastern US datasets. The observed R differed among instruments, with averages ranging from 0.93 for AMS ground data to 1.7 for PILS data. In comparison, ISORROPIA predicted R are always near 2. To explain this discrepancy, mainly two hypothesis are proposed, namely the organic-film hypothesis (Pye et al., 2017) and the non-volatile cations (NVC) hypothesis (as shown in this manuscript). By including in the measured NVC, the authors could now decrease predicted R from 1.97 to 1.85, which is still higher than the corresponding observation of 1.7. The remaining difference could possibly be due to the presence of organic-film, or the size heterogeneity.

[Figure]

Considering the large disagreement in observation data, neither of the above hypothesis could be fully validated. However, they are still of scientific interests and worth publishing, as they both provided robust explanations to be further examined; while several concerns need to be addressed before that.

(1) What's the average activity coefficient of NH3·H2O(aq) and NH4+? Does that change with NVC levels? If so, how would the theoretical S-curve be influenced, or what's the potential range of S-curve in this study? In comparison, the S-curve range based on the activity coefficient of H+ as given in Pye et al. (2017) should also be indicated.

(2) At high or low pH ranges, the partitioning fraction of NH3(g) can be extremely low or large, but can never reach 0% or 100%. What's the accuracy of the ISORROPIA model? Or, at what value would the model treat the ratio actually as 0% or 100%? Since the observation data can never be zero, what's the discrepancy of predicted NH3 and observation NH3 at those extreme conditions, for gas- and aerosol-phase respectively? Similarly, how about the HNO3-NO3- pair?

(3) Adding Fig. 3 in the authors' comment to Pye et al. (2017) would help improve the current manuscript. To my eye, the theoretical S-curve in that figure is to the right edges of the corresponding observation data. What if the aerosol water associated with organics are taken into account? That dilution effect would increase pH, shift the corresponding observation data points to the right and may result in better agreement. In addition, the authors claim that corresponding S-curve of Pye et al. (2017) can be derived by shift the S-curve of 0.8 pH units. This argument looks confusing and should be better described.

(4) The authors attributed the data with R over 2 to "measurement uncertainty and error propagation at low SO42- concentrations". However, based on data shown in Figure 1, these periods are not the periods with the lowest SO42- concentration (and thus largest uncertainty). Also, these periods correspond to periods with negative inferred

Na+. The arbitrary exclusion of these data is problematic. Basically that is to say that ambient aerosols can never be neutral or basic. As mentioned in other papers (Allen et al., 2015), sometimes the sea-salt episodes can be observed. How could the authors prove that cation-abundant situations are wrong? Does those data have any common distinct features from others? The data can be discarded for better reasons, not just due to that they look abnormal.

References:

Allen, H. M., Draper, D. C., Ayres, B. R., Ault, A., Bondy, A., Takahama, S., Modini, R. L., Baumann, K., Edgerton, E., Knote, C., Laskin, A., Wang, B., and Fry, J. L.: Influence of crustal dust and sea spray supermicron particle concentrations and acidity on inorganic $NO_3^{-}$ aerosol during the 2013 Southern Oxidant and Aerosol Study, Atmos. Chem. Phys., 15, 10669-10685, 2015.

Pye, H. O. T., Zuend, A., Fry, J. L., Isaacman-VanWertz, G., Capps, S. L., Appel, K. W., Foroutan, H., Xu, L., Ng, N. L., and Goldstein, A. H.: Coupling of organic and inorganic aerosol systems and the effect on gas-particle partitioning in the southeastern United States, Atmos. Chem. Phys. Discuss., 2017, 1-25, 2017.

---

## Short Comment (SC2) · 14 Sep 2017

Comments on 'The underappreciated role of nonvolatile cations on aerosol ammonium-sulfate molar ratios', by Guo et al. (2017)

Comments submitted by Rachel Silvern and Daniel Jacob, Harvard University

Guo et al. (2017) criticize the results from the recent paper by Silvern et al. (2017) that built on the previous work by Kim et al. (2015). Silvern et al. (2017) found that the ammonium-sulfate molar ratio (R) in aerosol over the Southeast US in summer averages only $1.04 \pm 0.21$ mol mol-1, despite a large excess of ammonia, whereas

thermodynamic models such as ISORROPIA predict that R should approach 2 under these conditions. Silvern et al. (2017) further showed that R steadily decreased during the 2003-2013 period even as SO2 emissions decreased, inconsistent with thermodynamic models.

Guo et al. (2017) argue that the apparent inconsistency of observed R with thermodynamic models can be reconciled by taking into account non-volatile cations (NVCs) in the aerosol charge balance. They show that taking into account NVCs reconciles model and observed R at their SOAS site in the Southeast and for the WINTER aircraft campaign in the Northeast.

In our opinion, however, the paper does not resolve the discrepancies with thermodynamics found by Kim et al. (2015) and Silvern et al. (2017). Specifically:

1. Both Kim et al. (2015) and Silvern et al. (2017) included observed NVCs in their charge balance ratios and found them to have little effect because NVCs are present in very low concentrations relative to NH4+. Guo et al. (2017) do not acknowledge that both Kim et al. (2015) previously examined the effect of NVCs. The effect of NVCs was shown specifically in Figure 7 of Kim et al. (2015) and in p. 5110 of Silvern et al. (2017). So the inclusion of NVCs in the charge balance calculation is not new. It is also discussed by Pye et al. (ACPD 2017, https://doi.org/10.5194/acp-2017-623). Guo et al. (2017) may consider them 'underappreciated' (viz. the title) but they must invoke 'inferred' NVC concentrations over 4.5 times measured values in order to reconcile R at their SOAS site with thermodynamic calculations (bottom of page 4).

2. The 'inferred Na+' from Guo et al. (2017) seems like a self-fulfilling prophecy. Here Na+ (taken as a proxy for NVCs) is specified not from observations but from a charge balance 2[SO42-]+[NO3-]-[NH4+] excluding [H+]. Of course in that case the predicted R will match the observed values but that is nothing more than fitting NVC to match the observed R. The problem is that these fitted values of NVCs are much higher than observed. Guo et al. [2017] make the case that [H+] in the charge balance must be

small but this is based on inferred Na+ so the argument seems circular.

3. Guo et al. (2017) start from a value R = 1.70 mol mol-1 for a select time range at their SOAS site, and it does not take much correction to bring this value to approaching 2 through inclusion of NVCs. However, the CSN observations across the Southeast and the AMS observations from SOAS reported by Silvern et al. (2017) indicate much lower mean values of 1.04 ± 0.21 mol mol-1 and 0.93 ± 0.29 mol mol-1 respectively. This is not acknowledged by Guo et al. (2017). Silvern et al. (2017) noted the discrepancy with the higher values of R observed at the SEARCH sites and analyzed the differences in the measurement protocols, concluding that the R values could be biased low in the CSN data and biased high in the SEARCH data. Weber et al. (2016) reported a value of R = 1.54 for one representative SEARCH site in summer 2013 and Guo et al. (2015) reported R = 1.4 for SOAS. Pye et al. (2017) also pointed out this large difference in R between data sets. Guo et al. (2017) should acknowledge this difference in observed values of R between datasets because it has major implications for their assertion that NVCs can solve the problem.

4. The most puzzling and interesting behavior of R noted by Silvern et al. (2017) is its decrease over the 2003-2013 period as SO2 emissions decrease while ammonia emissions stay flat. Guo et al. (2017) need to acknowledge this discussion of the trend by Silvern et al (2017). The top panel of Figure 5 in Guo et al. (2017) shows that ISORROPIA cannot properly predict the trend, supporting the argument made by Silvern et al. (2017) that observed trends show a departure from equilibrium behavior. In order to capture observed trends, Guo et al. (2017) require n an increasing concentration of NVCs with time (an increasing 'inferred Na+') that is inconsistent with observed concentrations at SEARCH sites.

Beyond these specifics, Guo et al. (2017) misrepresent the results of Silvern et al. (2017) in three general ways:

5. They suggest that Silvern et al. (2017) did not think of NVCs. This is apparent

in the title "The underappreciated role…" and the fact that nowhere in the paper is it acknowledged that Silvern et al. (2017) and previously Kim et al. (2015) quantified the effect of NVCs on R – see comment 1 above.

6. They suggest (specifically in the abstract) that Silvern et al. (2017) is solely about the organic film hypothesis with claims that retardation of equilibrium would somehow not apply to water vapor or nitric acid. In fact, Silvern et al. (2017) only presented the organic film as a tentative hypothesis to explain the decrease in R over the 2003-2013 period (see their Abstract). Also, they did not claim that it would not retard equilibrium for H2O and HNO3, and instead simply discussed the implications for H2O and HNO3 as an open question (see their last paragraph).

7. They imply that Silvern et al. (2017) do not understand the basics of H2SO4-NH3 thermodynamics and the implications for pH, but this seems based on inadequate reading of the paper. Contrary to what Guo et al. (2017) state (see for example page 5, lines 20-22), Silvern et al. (2017) did not claim that thermodynamics imply complete neutralization of sufate by ammonia resulting in elevated pH. In fact Silvern et al. (2017) took pains to show that the thermodynamic argument is not about pH but about R. See their Figure 1 which shows a pH value below 2 even when total ammonia is in large excess. Guo et al. (2017) claim that Silvern et al. (2017) are using R as "acidity proxy" to get at pH (page 2, lines 9-12) but Silvern et al. (2017) in fact emphasize that their paper is not about pH. The Guo et al. (2017) paper suggests that the controversy is about pH (page 8, lines 5-7) but it is not; it's about R.

References:

Guo, H., Xu, L., Bougiatioti, A., Cerully, K. M., Capps, S. L., Hite, J. R., Carlton, A. G., Lee, S. H., Bergin, M. H.,Ng, N. L., Nenes, A., and Weber, R. J.: Fine-particle water and pH in the southeastern United States, Atm. Chem. Phys., 15, 5211-5228, doi: 10.5194/acp-15-5211-2015, 2015.

Kim, P.S., D.J. Jacob, J.A. Fisher, K. Travis, K. Yu, L. Zhu, R.M. Yantosca, M.P. Sulprizio,

J.L. Jimenez, P. Campuzano-Jost, K.D. Froyd, J. Liao, J.W. Hair, M.A. Fenn, C.F. Butler, N.L. Wagner, T.D. Gordon, A. Welti, P.O. Wennberg, J.D. Crounse, J.M. St. Clair, A.P. Teng, D.B. Millet, J.P. Schwarz, M.Z. Markovic, and A.E. Perring, Sources, seasonality, and trends of Southeast US aerosol: an integrated analysis of surface, aircraft, and satellite observations with the GEOS-Chem model, Atmos. Chem. Phys., 15, 10,411-10,433, 2015.

Silvern, R.F., D.J. Jacob, P.S. Kim, E.A. Marais, J.R. Turner, Campuzano-Jost, P., and Jimenez, J. L., Inconsistency of ammonium-sulfate aerosol ratios with thermodynamic models in the eastern US: a possible role of organic aerosol, Atmos. Chem. and Phys., 17, 5107-5118, doi:10.5194/acp-17-5107-2017, 2017.

Weber, R. J., Guo, H., Russell, A. G., and Nenes, A.: High aerosol acidity despite declining atmospheric sulfate concentrations over the past 15 years, Nature Geoscience, 9, 282-285, doi: 10.1038/ngeo2665, 2016.

---

## Short Comment (SC3) · 4 Oct 2017

We thank Rachel Silvern and Daniel Jacob for their interest in our paper, and for challenging our work with thought-provoking arguments (that we thoroughly enjoy responding to!). Before a point-by-point response to the issues raised, we would like to articulate the main points of our work:

- Including NVCs in the thermodynamic model largely resolves the molar ratio discrepancy, based on our data set, which is representative of the southeast. Only small amounts of NVC are often required, therefore the practice of omitting NVCs from fine mode calculations (which may seem unavoidable for many datasets) induces important biases in molar ratios, which have to be considered in any relevant interpretations (especially on the role of organics).
- The bias in $R$ (ISORROPIA predicted $R$ with $Na^+$ minus ISORROPIA predicted $R$ without $Na^+$, where $R = NH_4^+/SO_4^{2-}$, mole/mole) is highly correlated with measured $Na^+$ ($r^2 = 0.93$), but not correlated with OA mass or OA mass fraction. Furthermore, the difference in observed $R$ from a ratio of 2 ($R$ observed minus 2) is correlated with NVCs and not correlated with OA mass or OA mass fraction. Both results provide strong evidence for NVCs, and not OA, as the underlying driver of the molar ratio discrepancy.

We now proceed with detailed responses (in normal font) to the comments (in italics).

*Guo et al. (2017) criticize the results from the recent paper by Silvern et al. (2017) that built on the previous work by Kim et al. (2015). Silvern et al. (2017) found that the ammonium-sulfate molar ratio (R) in aerosol over the Southeast US in summer averages only 1.04 ± 0.21 mol mol-1, despite a large excess of ammonia, whereas thermodynamic models such as ISORROPIA predict that R should approach 2 under these conditions. Silvern et al. (2017) further showed that R steadily decreased during the 2003-2013 period even as SO2 emissions decreased, inconsistent with thermodynamic models.*

Steadily decreasing $R$ was reported in the southeast by Hidy et al. (2014) (see Fig 5) and in Weber et al. (2016) (see Fig 3). Weber et al. (2016) provided the first explanation; based on simple thermodynamic principles it could be explained by the contrast in volatility between ammonia/ammonium and sulfate. The Silvern et al. (2017) film is an alternative explanation that accounts for both the $R$ trend and reconciles observed and thermodynamic model-predicted molar ratios. From analysis of our data sets, the latter may be explained by NVCs.

In the first study, Weber et al. (2016) over-predicted $R_{SO4}$ due to limited consideration of NVCs. Despite that, the decreasing trend of $R_{SO4}$ was captured by ISORROPIA (see Fig 2 in Weber et al. (2016)). In this study, we show a good agreement between measured and ISORROPIA-predicted $R_{SO4}$ with inferred $Na^+$ from a charge balance, which is useful when NVC measurements are not available (Fig 5a). We do note that while not including NVC results in a decreasing $R_{SO4}$ trend, the rate is lower than when NVCs are included (contrast is shown in this paper Fig 5a green vs blue lines). These analyses further suggest the thermodynamic model can explain the observed $R_{SO4}$ trends, with no need for organic films.

*1. Both Kim et al. (2015) and Silvern et al. (2017) included observed NVCs in their charge balance ratios and found them to have little effect because NVCs are present in very low concentrations relative to NH4+. Guo et al. (2017) do not acknowledge that both Kim et al. (2015) previously examined the effect of NVCs. The effect of NVCs was shown specifically in Figure 7 of Kim et al. (2015) and in p. 5110 of Silvern et al. (2017). So the inclusion of NVCs in the charge balance calculation is not new. It is also discussed by Pye et al. (ACPD 2017, https://doi.org/10.5194/acp-2017-623). Guo et al. (2017) may consider them 'underappreciated' (viz. the title) but they must invoke 'inferred' NVC concentrations over 4.5 times measured values in order to reconcile R at their SOAS site with thermodynamic calculations (bottom of page 4).*

To clarify, the question goes beyond how NVCs affect the measured molar ratio or charge balance, but

how they affect the thermodynamic model prediction of $R$. For example, Kim et al. (2015) Fig 7 only shows measured $R$ that includes NVC, no thermodynamic modeling with NVCs is included (as far as we can tell). Silvern et al. (2017) does include a thermodynamic analysis.

The assertion that much higher NVC concentrations are needed than observed to reconcile $R$ is incorrect. Here we show that when measured NVC > measurement LOD (i.e., reliable NVC concentrations, an admittedly a small fraction of the study period) there is good agreement between measured and ISORROPIA-predicted $R$ (Fig 1b and 1c, and associated text; no difference at a significance level of $\alpha = 0.05$). Issues arise when the NVC concentrations are below or near LOD and they have to be estimated (this is not surprising). Most defensible conclusions can be made for periods with the most reliable data (NVCs above LODs). Assessments and critiques that focus solely on only periods of the least reliable data sets (NVCs below LODs) are weak. However, even periods with NVCs below LODs, often provide consistent results. For example:
- For SOAS, when measured $Na^+$ was below LOD (0.07 $\mu$g m$^{-3}$), a $Na^+$ of 0.1-0.3 $\mu$g m$^{-3}$ was needed to bring measured and predicted $R$ into agreement, which is near the detection limit and so difficult to measure. Despite this, the trends are the same and the predicted $R$ is generally smaller than the observed, and so including the estimated LVCs in this case over compensated.
- For WINTER data, inferred NVC results in agreement in $R$ prediction with the observation, but inferred $Na^+$ is smaller than observed $PM_1$ $Na^+$, 0.15 vs. 0.23 $\mu$g m$^{-3}$, stated in the caption of Fig 4. Again, the results are uncertain since measurements of NVC were not measured online in that study.
- For the case (we guess) where the 4.5 factor is referred to, inferred $Na^+ = 0.28 \pm 0.18$ $\mu$g m$^{-3}$ vs. measured $Na^+ = 0.06 \pm 0.18$ $\mu$g m$^{-3}$. This does cause an under-prediction of $R$ (predicted $R = 1.43 \pm 0.32$ vs. measured $R = 1.70 \pm 0.23$ ($\pm$ SD of the data)). T-test shows a statistical difference, but because the results are not based on measured NVCs we are reluctant to make inferences on the cause of the discrepancy.

Our main finding is that the NVCs, even at low levels, have significant impact on predicted molar ratios (i.e., predicted ammonium concentration), in contrast to findings of Kim et al. (2015), Silvern et al. (2017), and Pye et al. (2017). If our findings are correct, 'underappreciated' is appropriate. We will cite the above papers with their findings in the revised draft.

Why the different conclusions? Maybe use of different (and possibly inconsistent) data sets? We note that NVC measurement limitations (measurement uncertainties) could be a cause. Also, we have used what we feel is one of the best data sets, which allows more precise, time-resolved analysis and hence more robust conclusions: Online PILS-IC for anions and cations, CIMS ammonia in our analysis, versus Kim et al. (2015), Silvern et al. (2017), or Pye et al. (2017) that use a combination of AMS data that does not include NVCs and CSN 24-hr filter $PM_{2.5}$ data which does include NVCs, but have known negative $NH_4^+$ biases ($R$ will be biased low).

All of our results can be examined as we have provided the ISORROPIA-II input files in the supplement.

*2. The 'inferred Na+' from Guo et al. (2017) seems like a self-fulfilling prophecy. Here Na+ (taken as a proxy for NVCs) is specified not from observations but from a charge balance 2[SO42-]+[NO3-]-[NH4+] excluding [H+]. Of course in that case the predicted R will match the observed values but that is nothing more than fitting NVC to match the observed R. The problem is that these fitted values of NVCs are much higher than observed. Guo et al. [2017] make the case that [H+] in the charge balance must be small but this is based on inferred Na+ so the argument seems circular.*

We understand the point raised, but do not agree; the "self-fulfilling prophecy" would occur if we assumed $NH_4$ and $NO_3$ were not allowed to equilibrate with the (observed) gas phase and volatilize, or if the pH of the computed aerosol were very close to neutral. Neither happens, as ions volatilize and the

aerosol becomes acidic (with a pH between 0 and 2 in this study). Therefore, we stand by our suggestion that the NVC derived from a charge balance to the data poses an upper limit for NVCs.

*3. Guo et al. (2017) start from a value R = 1.70 mol mol-1 for a select time range at their SOAS site, and it does not take much correction to bring this value to approaching 2 through inclusion of NVCs. However, the CSN observations across the Southeast and the AMS observations from SOAS reported by Silvern et al. (2017) indicate much lower mean values of **1.04 ± 0.21** mol mol-1 and 0.93 ± 0.29 mol mol-1 respectively. This is not acknowledged by Guo et al. (2017). Silvern et al. (2017) noted the discrepancy with the higher values of R observed at the SEARCH sites and analyzed the differences in the measurement protocols, concluding that the R values could be biased low in the CSN data and biased high in the SEARCH data. Weber et al. (2016) reported a value of R = 1.54 for one representative SEARCH site in summer 2013 and Guo et al. (2015) reported R = 1.4 for SOAS. Pye et al. (2017) also pointed out this large difference in R between data sets. Guo et al. (2017) should acknowledge this difference in observed values of R between datasets because it has major implications for their assertion that NVCs can solve the problem.*

The inclusion of NVCs moves the predicted $R$ away from 2 into agreement with measured $R$. The selected 13-days period of the SOAS study is representative of the SOAS PILS-IC $PM_{2.5}$ data as it is 13 days of the 17 days (76%) of the $PM_{2.5}$ data. The first 4 days of the study in which $PM_{2.5}$ data is available, 06/07-06/10, were more affected by precipitation and so not included, but still gave a good prediction of $NH_3$, as shown by Fig 10 in Guo et al. (2015).

We did not selectively pick data that had $R$ values close to 2. The measured $R$ for the selected period (1.70 ± 0.23) is representative of the whole SOAS campaign (1.60 ± 0.23 for all of SOAS by MARGA measurement (Allen et al., 2015), the other $PM_{2.5}$ $R$ data = 1.90 ± 1.01 (all points) and 1.68 ± 0.47 (excluding $R$ over 3, 8% points) also for all of SOAS by SEARCH $PM_{2.5}$ online measurement) (± is the SD of the data). The PILS-IC measured $R$ during the SOAS study falls right in the range of the long-term SEARCH observations at CTR; in 2013, summer mean $R$ and SD was 1.63 ± 0.16. Hidy et al. (2014) plot $R$ for all SEARCH sites over many years (Fig 5). For all SEARCH sites in the southeast, Silvern et al. (2017) give 1.62 ± 0.17. Our smaller SOAS data set is consistent with all these data. The CSN data, which Silvern et al. (2017) also used and reports at an $R$ of 1.44 ± 0.34 (the 1.04 ± 0.21 in the comment is misquoted; that is for CSN $R_{SO4}$ value), is lower due to the known low $NH_4^+$ bias due to CSN use of nylon filters. In our analysis, we have focused what we believed are the best data (PILS vs. Filter). Finally, we also have shown that a relatively small change in NVC can lead to substantial change in predicted $R$. For example, Fig 3a shows that the change in $R$ is quite sensitive to concentrations of NVCs; the slope in $\Delta R$ vs $Na^+$ concentration (in $\mu g\ m^{-3}$) is near 2.

We agree that there are differences in $R$ observations (ignoring CSN due to a known bias). A significant fraction of the differences in $R$ between data sets can be attributed to the differences in measured particle size ranges (in contrast to what Pye et al. (2017) finds). A unique feature of our SOAS data is that we have both $PM_1$ and $PM_{2.5}$ results measured with the same instrumentation (only cyclone inlet changed, $PM_{2.5}$ PILS-IC data for the 1st half and $PM_1$ for the 2nd half). $PM_1$ vs $PM_{2.5}$ has an effect on measured $R$. For example, see the wide variation in SOAS measured $R$ in the figure below. For the 1st half, the PILS-IC and MARGA $PM_{2.5}$ $R$ tend to agree, AMS $PM_1$ $R$ is much lower, for the 2nd half there is much less difference between AMS and PILS-IC $PM_1$ $R$ compared to the larger difference between these $PM_1$ and MARGA $PM_{2.5}$ $R$. Overall, the figure shows an increase in $R$ from $PM_1$ to $PM_{2.5}$ measurements with minor differences possibly due to instrumentation issues. (Aside: The comparison of AMS vs PILS-IC for the whole SOAS study can be found at Guo et al. (2015) in Fig 2. MARGA sulfate and ammonium are also highly correlated to and close to PILS-IC; ODR fits, MARGA $SO_4^{2-}$ = (1.00 ± 0.01) PILS-IC $SO_4^{2-}$ + (0.51 ± 0.02 $\mu g\ m^{-3}$), $r^2$ = 0.96; MARGA $NH_4^+$ = (1.04 ± 0.01) PILS-IC $NH_4^+$ + (0.21 ± 0.01 $\mu g\ m^{-3}$), $r^2$ = 0.91, see our comment to Pye et al. (2017) for a plot.)

[Figure]

Figure I. Comparison of measured $R$ by AMS, PILS-IC, and MARGA for (left) the entire SOAS campaign, (center) the 1st half of SOAS, and (right) the 2nd half of SOAS. PILS-IC measured $PM_{2.5}$ in the 1st half and switched to $PM_1$ in the 2nd half of the study, while AMS measured $PM_1$ and MARGA measured $PM_{2.5}$ for the entire study. For a consistent comparison, we excluded the data if any $R$ from the three instruments is not available (therefore, the statistics may be slightly different from the above in the text). 10% and 90% percentiles are shown as vertical black sticks; 25%, 50% (median) and 75% percentiles are shown as horizontal black sticks and grey shades. Means are marked by solid red circles and numbers are listed above each instrument. Red error bars indicate propagated measurement uncertainties calculated with quadrature sum of squares based on measurement uncertainties of AMS 35%, PILS-IC 15%, and MARGA 10% (assumed based on Rumsey et al. (2014).

*4. The most puzzling and interesting behavior of R noted by Silvern et al. (2017) is its decrease over the 2003-2013 period as SO2 emissions decrease while ammonia emissions stay flat. Guo et al. (2017) need to acknowledge this discussion of the trend by Silvern et al (2017). The top panel of Figure 5 in Guo et al. (2017) shows that ISORROPIA cannot properly predict the trend, supporting the argument made by Silvern et al. (2017) that observed trends show a departure from equilibrium behavior. In order to capture observed trends, Guo et al. (2017) require n an increasing concentration of NVCs with time (an increasing 'inferred Na+') that is inconsistent with observed concentrations at SEARCH sites.*

We agree this is interesting; this was also noted by Weber et al. (2016) together with its explanation based on thermodynamic principles. We do not require increasing NVCs with time; in fact we show (Figure 5c) that $Na^+$ *decreases* over time (with a linear trend of $-0.030 \pm 0.009$ µg m$^{-3}$ yr$^{-1}$, r$^2$ = 0.49, consistent with the SEARCH observations). What is happening is that $SO_4^{2-}$ is decreasing more rapidly than $Na^+$, so that in relative terms, the latter (and NVCs overall) become an increasingly important driver of the aerosol thermodynamic state (slope = 0.5% yr$^{-1}$, r$^2$ = 0.32).

*5. They suggest that Silvern et al. (2017) did not think of NVCs. This is apparent in the title "The underappreciated role. . ." and the fact that nowhere in the paper is it acknowledged that Silvern et al. (2017) and previously Kim et al. (2015) quantified the effect of NVCs on R – see comment 1 above.*

The importance of NVC in aerosol thermodynamics was known for decades (that is why ISORROPIA-II was developed to begin with!), and we will make this very clear in the revised version. However, effects of NVC in the fine mode (especially in the size ranges 1.0-2.5 µm) *is underappreciated*, because they are assumed unimportant or not constrained well enough by measurements. By having access to arguably the most precise, comprehensive and high resolution (in time) datasets allows us to test and resolve this issue (vs. using the $PM_{2.5}$ datasets), because pH (and $R$) can drastically change between $PM_1$ and $PM_{2.5}$, owing to the effects of NVC (Fang et al., 2017; Guo et al., 2017).

*6. They suggest (specifically in the abstract) that Silvern et al. (2017) is solely about the organic film hypothesis with claims that retardation of equilibrium would somehow not apply to water vapor or nitric acid. In fact, Silvern et al. (2017) only presented the organic film as a tentative hypothesis to explain the decrease in R over the 2003-2013 period (see their Abstract). Also, they did not claim that it would not retard equilibrium for H2O and HNO3, and instead simply discussed the implications for H2O and HNO3 as an open question (see their last paragraph).*

We did not mean to imply that Silvern et al. (2017) is *only* about the film hypothesis. However, the film hypothesis is one of the few mechanisms suggested that could be tested. We support that films are not likely because: *i*) the difference in measured *R* from 2 is anticorrelated with organics altogether (with implications for the role of organics that go beyond just acting as a kinetic barrier), but correlated with NVC (or $Na^+$) concentrations, and, *ii*) it is hard to justify organic films selectively inhibiting $NH_3$ from reaching equilibrium, but allow equally diffusive and sticky $H_2O$ and $HNO_3$ molecules to "pass through" establish gas-particle equilibrium. Furthermore, Weber et al. (2016) provided the basis for explaining the *R* trend, based on the thermodynamics of the inorganics alone.

*7. They imply that Silvern et al. (2017) do not understand the basics of H2SO4- NH3 thermodynamics and the implications for pH, but this seems based on inadequate reading of the paper. Contrary to what Guo et al. (2017) state (see for example page 5, lines 20-22), Silvern et al. (2017) did not claim that thermodynamics imply complete neutralization of sufate by ammonia resulting in elevated pH. In fact Silvern et al. (2017) took pains to show that the thermodynamic argument is not about pH but about R. See their Figure 1 which shows a pH value below 2 even when total ammonia is in large excess. Guo et al. (2017) claim that Silvern et al. (2017) are using R as "acidity proxy" to get at pH (page 2, lines 9-12) but Silvern et al. (2017) in fact emphasize that their paper is not about pH. The Guo et al. (2017) paper suggests that the controversy is about pH (page 8, lines 5-7) but it is not; it's about R.*

"Incomplete neutralization" was the focus of Silvern et al. (2017), and central to that is the thermodynamic state of the system. pH is an important expression of the state, and governs the partitioning of ammonia and nitric acid onto the aerosol. Therefore, thermodynamics (and pH) controls the aerosol $NH_4^+$ (and if relevant, $NO_3^-$) concentration and *R* (= $NH_4^+/SO_4^{2-}$). To state that "it is about *R* and not about pH" is therefore incorrect or at least incomplete, as *R* is decoupled from aerosol acidity only when liquid water is absent. We will make sure that these points are clarified in the revised manuscript.

**References:**

Allen, H. M., Draper, D. C., Ayres, B. R., Ault, A., Bondy, A., Takahama, S., Modini, R. L., Baumann, K., Edgerton, E., Knote, C., Laskin, A., Wang, B., and Fry, J. L.: Influence of crustal dust and sea spray supermicron particle concentrations and acidity on inorganic $NO_3^-$ aerosol during the 2013 Southern Oxidant and Aerosol Study, Atmospheric Chemistry and Physics, 15, 10669-10685, doi: 10.5194/acp-15-10669-2015, 2015.

Fang, T., Guo, H., Zeng, L., Verma, V., Nenes, A., and Weber, R. J.: Highly Acidic Ambient Particles, Soluble Metals, and Oxidative Potential: A Link between Sulfate and Aerosol Toxicity, Environmental science & technology, 51, 2611-2620, doi: 10.1021/acs.est.6b06151, 2017.

Guo, H., Xu, L., Bougiatioti, A., Cerully, K. M., Capps, S. L., Hite, J. R., Carlton, A. G., Lee, S. H., Bergin, M. H., Ng, N. L., Nenes, A., and Weber, R. J.: Fine-particle water and pH in the southeastern United States, Atmospheric Chemistry and Physics, 15, 5211-5228, doi: 10.5194/acp-15-5211-2015, 2015.

Guo, H., Liu, J., Froyd, K. D., Roberts, J. M., Veres, P. R., Hayes, P. L., Jimenez, J. L., Nenes, A., and Weber, R. J.: Fine particle pH and gas–particle phase partitioning of inorganic species in Pasadena, California, during the 2010 CalNex campaign, Atmospheric Chemistry and Physics, 17, 5703-5719, doi: 10.5194/acp-17-5703-2017, 2017.

Hidy, G. M., Blanchard, C. L., Baumann, K., Edgerton, E., Tanenbaum, S., Shaw, S., Knipping, E.,

Tombach, I., Jansen, J., and Walters, J.: Chemical climatology of the southeastern United States, 1999-2013, Atmospheric Chemistry and Physics, 14, 11893-11914, doi: 10.5194/acp-14-11893-2014, 2014.

Kim, P. S., Jacob, D. J., Fisher, J. A., Travis, K., Yu, K., Zhu, L., Yantosca, R. M., Sulprizio, M. P., Jimenez, J. L., Campuzano-Jost, P., Froyd, K. D., Liao, J., Hair, J. W., Fenn, M. A., Butler, C. F., Wagner, N. L., Gordon, T. D., Welti, A., Wennberg, P. O., Crounse, J. D., St Clair, J. M., Teng, A. P., Millet, D. B., Schwarz, J. P., Markovic, M. Z., and Perring, A. E.: Sources, seasonality, and trends of southeast US aerosol: an integrated analysis of surface, aircraft, and satellite observations with the GEOS-Chem chemical transport model, Atmospheric Chemistry and Physics, 15, 10411-10433, doi: 10.5194/acp-15-10411-2015, 2015.

Pye, H. O. T., Zuend, A., Fry, J. L., Isaacman-VanWertz, G., Capps, S. L., Appel, K. W., Foroutan, H., Xu, L., Ng, N. L., and Goldstein, A. H.: Coupling of organic and inorganic aerosol systems and the effect on gas-particle partitioning in the southeastern United States, Atmospheric Chemistry and Physics Discussions, 1-25, doi: 10.5194/acp-2017-623, 2017.

Rumsey, I. C., Cowen, K. A., Walker, J. T., Kelly, T. J., Hanft, E. A., Mishoe, K., Rogers, C., Proost, R., Beachley, G. M., Lear, G., Frelink, T., and Otjes, R. P.: An assessment of the performance of the Monitor for AeRosols and GAses in ambient air (MARGA): a semi-continuous method for soluble compounds, Atmospheric Chemistry and Physics, 14, 5639-5658, doi: 10.5194/acp-14-5639-2014, 2014.

Silvern, R. F., Jacob, D. J., Kim, P. S., Marais, E. A., Turner, J. R., Campuzano-Jost, P., and Jimenez, J. L.: Inconsistency of ammonium–sulfate aerosol ratios with thermodynamic models in the eastern US: a possible role of organic aerosol, Atmospheric Chemistry and Physics, 17, 5107-5118, doi: 10.5194/acp-17-5107-2017, 2017.

Weber, R. J., Guo, H., Russell, A. G., and Nenes, A.: High aerosol acidity despite declining atmospheric sulfate concentrations over the past 15 years, Nature Geoscience, 9, 282-285, doi: 10.1038/ngeo2665, 2016.

---

## Referee Comment (RC2) · Anonymous Referee #2 · 4 Nov 2017

ACP Guo Review

**Summary**

This manuscript describes the potential role of nonvolatile cations, such as sodium (or magnesium, potassium, or calcium, if soluble) , on ammonium to sulfate ratios. Further the data presented is used as evidence that organic films do not form on particles, which could inhibit partitioning between the particle and the gas phase. A positive of the work is that it considers species beyond the "traditional" non-refractory species measured by AMS, which are often excluded from discussions of submicron aerosols. However, there are a number of severe shortcomings that constitute fundamental flaws to the logic of the article and it should not be published in its present form.

The largest issue is that a fully internal mixture (with all species present in each particle at their bulk atmospheric concentrations) is assumed, meaning that any $Na^+$ is assumed to be present in the same particles that have ammonium, sulfate, and nitrate. Literature from co-located sampling over the same time period shows that in fact most $Na^+$ present during SOAS was not mixed with SOA, but was present in sea spray aerosol or other mechanically generated particles, such as mineral dust (Allen et al., 2015; Bondy et al., 2017). Thus $Na^+$ cannot explain the values of R that the manuscript is using $Na^+$ to explain. The concept of "inferred-sodium" is particularly worrisome, as it is not necessarily supported even by bulk measurements and $Na^+$. Another serious concerns is that values of $Na^+$ are reported that are below LOD, which is not appropriate and that the measured $Na^+$ is below the LOD when "inferred $Na^+$" is >4 times higher than a value which is unreliable. Studies going back almost 20 years in Atlanta and the southeast U.S. have shown that NVCs are not present in SOA, particularly with single particle mass spectrometry, which is extremely sensitive to $Na^+$ and $K^+$. Overall, the topic is important and interesting, but the conclusions drawn from the manuscript overreach the data, the main result is fundamentally flawed and as a result the conclusions drawn (that NVCs are important in SOA) are not supported. The authors are clear leaders in the topic of aerosol acidity and thermodynamic modeling, but there is considerable concern is that if the manuscript is published with the current results and conclusions, this could spread the misconception that SOA contains sufficient NVCs to impact the sulfate-ammonium ratio, which is not the case.

**Major Comments**

- Page 4 Line 18: "We also assumed that the particles were internally mixed, and that pH did not vary with size"… and gas-particle partitioning was at thermodynamic equilibrium." The use of the non-volatile cations ($Na^+$, $Mg^{2+}$, etc.) without consideration of mixing state is very likely to lead to incorrect results, and numerous studies with different methods from Atlanta and the southeast have shown this over the past 20 years. Single particle mass spectrometry, such as ATOFMS is extremely sensitive to $Na^+$ and $K^+$ due to their ionization energies and can be detected even when well under 0.1% of the particle. Citations are discussed below, but briefly, there is essentially no sodium, magnesium, potassium, or calcium in SOA particles in the southeast US. Measurements are not as readily available in the northeast, but the expectation is that they would not be present mixed with SOA there either. There are NVCs present in the atmosphere at the same time, but they are present in other types of particles (industrial, biomass burning, sea salt, etc.) and thus NVCs cannot explain the discrepancies in R that this manuscript attributes the discrepancies to. Despite the interesting premise and analysis, the fundamental fact that NVCs are not in SOA renders many of the conclusions invalid.

- o 2003 JGR Middlebrook et al. (2003) compared multiple single particle mass spectrometers. The PALMS (Murphy) organic/sulfate class of particles does not have an sodium peak, but the Na/K sulfate (likely SSA) and mineral dust do (though not labeled in dust, it is present in the spectrum). The ATOFMS spectra also shows no $Na^+$ in organic/sulfate particles. The same lack of $Na^+$ is also seen with RSMS-II (Johnston).
- o 2003 JGR-A Liu et al. (2003) showed EC/OC and OC particles and the individual mas spectra did not contain $Na^+$ (Figure 3). In Wenzel et al a missing particle type was identified (Wenzel et al., 2003). This missing type corresponds to ammonium sulfate mixed with organic carbon, which have low 266 nm cross sections. If Na was present in these particles, they would have been more likely to be ionized by the 266 laser and not been a "missing" type.
- o 2011a&b ES&T Hatch et al. shows 41000 organic carbon particles from Atlanta during ANARChE, which showed no $Na^+$ with the OC (Figure 1). That paper also showed for ANARChE and a study in 2008 (AMIGAS) that particles containing organosulfates (likely many OC-sulfate particles) have different sources than particles without organosulfates (Figure 3, 4, 5) (Hatch et al., 2011a; Hatch et al., 2011b).
- o Future work (unpublished) from SOAS will show that Na is present is at levels > 1% of dry particle mass (thus excluding water), in less than 2% of SOA particles. That manuscript will show that Na is present in 5 other types of particles, which account for almost all sodium present.
- Page 4 Lines 33-36 and Page 5 lines 1-7: Of the three different $Na^+$ levels tested, option 1 infers that any lack of charge balance can be attributed to $Na^+$. On line 35-36, the authors then note that inferring the amount of $Na^+$ leads to a value more than 4 times higher than the measured value. There are a number of other possibilities that could explain these strongly different results and inferring shouldn't work as there is almost no $Na^+$ mixed with SOA particles. This is in fact supported by the fact that the reported values for $Na^+$ are below the limit of detection of the measurement (0.06 value when LOD is 0.07).
- Page 5 Lines 2-3: The authors choose to use values for $Na^+$ below LOD for the rest of the study, even though they are below LOD. This is a substantial issue and values below LOD should not published or used as the basis for the main analysis over the remainder of the paper.
- Page 5 Lines 9-15: Most $Na^+$ at SOAS is present from sea spray aerosol (SSA), which has been shown in multiple papers (Allen et al., 2015; Bondy et al., 2017). This authors' observation that $Cl^-$ and $NO_3^-$ have high $R^2$ values with $Na^+$ and agrees with the papers showing that $Na^+$ is from other sources than SOA, which further highlights that $Na^+$ is not present in SOA particles. It is also worth noting that June 10-13 at SOAS (during the period Guo et al focus on) there was a large contribution of sea spray aerosol at Centreville (Bondy et al., 2017). This includes the submicron where 20-40% of particles between 200-1000 nm were not SOA-dominant particles and are likely where nearly all of the $Na^+$ was located.
- Page 6 Lines 7-10: The paper here discusses June 11-13, which was a high sea salt time period, leading to a more externally mixed aerosol and the $Na^+$ was not present in SOA particles.
- Page 6 Last Paragraph: While $Na^+$ somewhat tracks with delta R, since $Na^+$ is not in the same particles the paragraph comes to close to attributing causation to a correlation. The wording in this section needs to be weakened. Also, the presumption that organic mass should be correlated with organic film thickness is overstated. Composition, viscosity, etc. all matter when a film might form and OA mass fractions would not provide meaningful estimates of this.

- Page 7 Lines 10-13: The authors mention higher $Na^+$ near coastlines, which is again likely from sea spray and would not be mixed with SOA in the same particles in all likelihood (unfortunately there is not single particle data to provide information on mixing state from that study to my knowledge).

**Minor Comments**

- The clarity of the writing, particularly in the introduction, could be improved as there are numerous long sentences and confusing wording that could be improved. Also there numerous missing words, incorrect plural versus singular, and conjugation throughout the manuscript, which needs to be cleaned up.
- Page 2 Lines 7-11: "Despite its importance, the inability to directly measure fine mode particle pH (e.g. Rindelaub et al. (2016) presents an indirect method that infers particle H+ activity for sizes above 10 μm and requires activity coefficient predicted by a thermodynamic modeling. This method reports the pH for a HSO4-/SO42- aerosol system similar to the fine particle pH predicted by a thermodynamic modeling used in this study (Guo et al., 2015)) has led to the use of measurable aerosol properties as acidity proxies, such as aerosol ammonium sulfate ratio or ion balances (e.g. (Paulot and Jacob, 2014; Wang et al., 2016; Silvern et al., 2017)). Recent work has shown that acidity proxies are not uniquely related to pH, which in turn strongly questions any conclusions derived from its use. There are numerous reasons why acidity proxies do not represent pH well; they do not capture the variability in particle water content, ion activity coefficients, or partial dissociation of species in the aerosol phase (Guo et al., 2015; Hennigan et al., 2015; Guo et al., 2016)."
    - There are a number of problems with this section that need to be addressed.
    - It is not fair to state that the use of the sulfate/bisulfate ratio in Rindelaub et al. led to the use of molar ratios or ion balances, since those have been used for decades and Rindelaub came out in 2016 (Rindelaub et al., 2016).
    - Secondly, the use of the bisulfate-to-sulfate ratio to determine pH is not indirect and is in fact more direct than thermodynamic modeling of gas particle partitioning, as both species are in the aerosol phase where pH is being determined and the effects of coatings or other non-ideal behavior is avoided. The method does use a thermodynamic model to determine activity coefficients, but the fact this is combined with multiple concentrations directly measured in the aerosol phase makes it very different than ion balance or molar ratio methods.
    - The manuscript states that Rindelaub only used particles above 10 microns, but fails to mention Craig et al. from this summer, which showed this direct method working down to 2-3 microns for a range of systems (Craig et al., 2017).
- Page 2, 2nd Paragraph: The authors go to great lengths to make clear that organic coatings or glassy particles which inhibit equilibrium between gases and particles are not possible based on "established literature", but most of that literature is from the groups who authored this manuscript. If there are limitations to the thermodynamic model approach used in those studies, citing their prior work does not invalidate the other work suggesting films might be important, such as Havala Pye's modeling paper from earlier this year.
- Page 3 Line 26: The authors state that "14% of "sulfate" is predicted to be $HSO_4^-$ and the rest as $SO_4^{2-}$ in the winter dataset." Based on a simple acid dissociation constant calculation at pH =1, bisulfate should be >80% of the combination of sulfate and bisulfate. Some explanation should be included to explain why this does not follow basic acid dissociation rules. Is it related to activity coefficients somehow? This need to be addressed in a revised manuscript. Also it is confusing to refer to sulfate,

bisulfate, and sulfuric acid together as "total sulfate", as they are each distinct species. Total S(VI) sulfur could work or some other nomenclature.

- Page 2 Line 9: I believe the word "and" is missing after (Guo et al., 2015)). Also the second parenthesis is not needed.
- Page 3 Line 13: "or if there is free $H_2SO_4$ in the aerosol". For the pH values in this manuscript and others using this method, the aerosol acidity is never sufficient for any $H_2SO_4$ to exist. Below pH = 2, sulfate will transition to bisulfate, but not sulfuric acid.
- Page 3 Line 14: missing the word "are", should be ", but are rare for"
- Page 5 Line 39: "mode R with measure $Na^+$ input", should be "measured"
- A constant throughout the manuscript is that strong statements are supported primarily by prior work from the authors of this study. It would strengthen the manuscript to either make less strong statements or cite work from other groups to support the claims being made.

**References**

Allen, H. M., Draper, D. C., Ayres, B. R., Ault, A., Bondy, A., Takahama, S., Modini, R. L., Baumann, K., Edgerton, E., Knote, C., Laskin, A., Wang, B., and Fry, J. L.: Influence of crustal dust and sea spray supermicron particle concentrations and acidity on inorganic NO3- aerosol during the 2013 Southern Oxidant and Aerosol Study, Atmos. Chem. Phys., 15, 10669-10685, 2015.

Bondy, A. L., Wang, B., Laskin, A., Craig, R. L., Nhliziyo, M. V., Bertman, S. B., Pratt, K. A., Shepson, P. B., and Ault, A. P.: Inland Sea Spray Aerosol Transport and Incomplete Chloride Depletion: Varying Degrees of Reactive Processing Observed during SOAS, Environ. Sci. Technol., 51, 9533-9542, 2017.

Craig, R. L., Nandy, L., Axson, J. L., Dutcher, C. S., and Ault, A. P.: Spectroscopic Determination of Aerosol pH from Acid–Base Equilibria in Inorganic, Organic, and Mixed Systems, J. Phys. Chem. A, 121, 5690-5699, 2017.

Hatch, L. E., Creamean, J. M., Ault, A. P., Surratt, J. D., Chan, M. N., Seinfeld, J. H., Edgerton, E. S., Su, Y., and Prather, K. A.: Measurements of Isoprene-Derived Organosulfates in Ambient Aerosols by Aerosol Time-of-Flight Mass Spectrometry - Part 1: Single Particle Atmospheric Observations in Atlanta, Environ. Sci. Technol., 45, 5105-5111, 2011a.

Hatch, L. E., Creamean, J. M., Ault, A. P., Surratt, J. D., Chan, M. N., Seinfeld, J. H., Edgerton, E. S., Su, Y. X., and Prather, K. A.: Measurements of Isoprene-Derived Organosulfates in Ambient Aerosols by Aerosol Time-of-Flight Mass Spectrometry-Part 2: Temporal Variability and Formation Mechanisms, Environ. Sci. Technol., 45, 8648-8655, 2011b.

Liu, D. Y., Wenzel, R. J., and Prather, K. A.: Aerosol time-of-flight mass spectrometry during the Atlanta Supersite Experiment: 1. Measurements, J. Geophys. Res.: Atmos., 108, 2003.

Middlebrook, A. M., Murphy, D. M., Lee, S. H., Thomson, D. S., Prather, K. A., Wenzel, R. J., Liu, D. Y., Phares, D. J., Rhoads, K. P., Wexler, A. S., Johnston, M. V., Jimenez, J. L., Jayne, J. T., Worsnop, D. R., Yourshaw, I., Seinfeld, J. H., and Flagan, R. C.: A comparison of particle mass spectrometers during the 1999 Atlanta Supersite Project, J. Geophys. Res.: Atmos., 108, 2003.

Rindelaub, J. D., Craig, R. L., Nandy, L., Bondy, A. L., Dutcher, C. S., Shepson, P. B., and Ault, A. P.: Direct Measurement of pH in Individual Particles via Raman Microspectroscopy and Variation in Acidity with Relative Humidity, J. Phys. Chem. A, 120, 911-917, 2016.

Wenzel, R. J., Liu, D. Y., Edgerton, E. S., and Prather, K. A.: Aerosol time-of-flight mass spectrometry during the Atlanta Supersite Experiment: 2. Scaling procedures, J. Geophys. Res.: Atmos., 108, 2003.

---

## Author Comment (AC1) · 11 Jan 2018

**Responses to Referee #1**

We thank the referee for the thoughtful and constructive comments. Before a point-by-point response to the issues raised, we would like to articulate the main points of our work:

- Including NVCs in the thermodynamic model largely resolves the molar ratio discrepancy, based on our data set, which is representative of the southeast. Only small amounts of NVC are often required, therefore the practice of omitting NVCs from fine mode calculations (which may seem unavoidable for many datasets) induces important biases in molar ratios, which have to be considered in any relevant interpretations (especially on the role of organics).

- The bias in $R$ (ISORROPIA predicted $R$ with $Na^+$ minus ISORROPIA predicted $R$ without $Na^+$, where $R = NH_4^+/SO_4^{2-}$, mole/mole) is highly correlated with measured $Na^+$ ($r^2 = 0.93$), but not correlated with OA mass or OA mass fraction. Furthermore, the difference in observed $R$ from a ratio of 2 ($R$ observed minus 2) is correlated with NVCs and not correlated with OA mass or OA mass fraction. Both results provide strong evidence for NVCs, and not OA, as the underlying driver of the molar ratio discrepancy.

We have addressed the comments (numbered, below), with referee comments in quotes and italics, and our responses in plain text.

1. *"To explain this discrepancy, mainly two hypothesis are proposed, namely the organic-film hypothesis (Pye et al., 2017) and the non-volatile cations (NVC) hypothesis (as shown in this manuscript). By including in the measured NVC, the authors could now decrease predicted R from 1.97 to 1.85, which is still higher than the corresponding observation of 1.7. The remaining difference could possibly be due to the presence of organic-film, or the size heterogeneity. Considering the large disagreement in observation data, neither of the above hypothesis could be fully validated."*

   The above statistics are for the whole Fig. 1 period including many data points where we had to estimate $Na^+$ since the measured $Na^+$ was below our LOD. Focusing only on the periods with measured $Na^+$ above LOD (reliable NVC concentrations), there is no statistical difference between predicted and measured $R$ (t- test $\alpha = 0.05$).

   See also our response to the comments from Daniel Jacob and Rachel Silvern (Figure I in that response), where we also discuss differences in $R$ between model-predicted and observations and also point out that a significant fraction of the differences in $R$ between some data sets (e.g., AMS $PM_1$ vs various $PM_{2.5}$ data) can be attributed to the differences in measured particle size ranges.

In summary, our NVC analysis can fully explain the discrepancy in molar ratio predictions for either PM$_{2.5}$ or PM$_1$ data sets.

2. *"What's the average activity coefficient of NH3·uH2O(aq) and NH4+? Does that change with NVC levels? If so, how would the theoretical S-curve be influenced, or what's the potential range of S-curve in this study? In comparison, the S-curve range based on the activity coefficient of H+ as given in Pye et al. (2017) should also be indicated"*

The activity coefficient of dissolved NH$_3$, $\gamma(NH_3)$, has a negligible effect on the S curve and so not considered (Guo et al., 2017). For example, at 298 K, the acid dissociation constant of NH$_4^+$, $K_a = 5.69 \times 10^{-10}$ mole L$^{-1}$ (Clegg et al., 1998), results in $\frac{K_a}{\gamma_{NH_3}} \ll \frac{\gamma_{H^+}[H^+]}{\gamma_{NH_4^+}}$ as long as the solution is not too basic. SOAS fine particles were very acidic with pH on average $0.94 \pm 0.59$ (SD). The measured Na$^+$ (above zero) for the SOAS study doesn't change $\frac{\gamma_{H^+}}{\gamma_{NH_4^+}}$ significantly; including or excluding Na$^+$ gives the same $\frac{\gamma_{H^+}}{\gamma_{NH_4^+}}$ of $1.38 \pm 0.12$ (no statistical difference as confirmed by t-test at $\alpha = 0.05$).

3. *"At high or low pH ranges, the partitioning fraction of NH3(g) can be extremely low or large, but can never reach 0% or 100%. What's the accuracy of the ISORROPIA model? Or, at what value would the model treat the ratio actually as 0% or 100%? Since the observation data can never be zero, what's the discrepancy of predicted NH3 and observation NH3 at those extreme conditions, for gas- and aerosol-phase respectively? Similarly, how about the HNO3-NO3- pair?"*

Theoretically, the partitioning fraction of NH$_3$ may never be 0 or 100%, but practically this is not an issue. We only use the semivolatile pairs with fractions close to 50% to constrain our pH predictions since this is the region of greatest sensitivity (e.g. (Guo et al., 2015)). Propagated uncertainty in the partitioning fraction can be determined from both gas and particle measurement uncertainties. The average propagated uncertainty in $\varepsilon(NH_4^+)$ is ~4% (absolute value, not percentage of $\varepsilon(NH_4^+)$) for SOAS, the pH prediction is accurate within 0.08 for $\varepsilon(NH_4^+)$ at 50% and 0.22 for $\varepsilon(NH_4^+)$ at 10%. A similar result is derived for HNO$_3$-NO$_3^-$ partitioning (0.07 for $\varepsilon(NO_3^-)$ at 50% and 0.22 for $\varepsilon(NO_3^-)$ at 10%).

4. *"Adding Fig. 3 in the authors' comment to Pye et al. (2017) would help improve the current manuscript. To my eye, the theoretical S-curve in that figure is to the right edges of the corresponding observation data. What if the aerosol water associated with organics are taken into account? That dilution effect would increase pH, shift the corresponding observation data points to the right and may result in better agreement. In addition, the authors claim that corresponding S-curve of Pye et al. (2017) can be derived by shift the S-curve of 0.8 pH units. This argument looks confusing and should be better described."*

To actually do this properly we need the Pye et al. (2017) data set, which is not yet available. Furthermore, this paper does not directly address the claims of Pye et al. (2017).

5. *"The authors attributed the data with R over 2 to "measurement uncertainty and error propagation at low SO42-concentrations". However, based on data shown in Figure 1, these periods are not the periods with the lowest SO42-concentration (and thus largest uncertainty). Also, these periods correspond to periods with negative inferred Na+. The arbitrary exclusion of these data is problematic. Basically that is to say that ambient aerosols can never be neutral or basic. As mentioned in other papers (Allen et al., 2015), sometimes the sea-salt episodes can be observed. How could the authors prove that cation-abundant situations are wrong? Do those data have any common distinct features from others? The data can be discarded for better reasons, not just due to that they look abnormal."*

The observed PILS-IC data points with $R$ over 2 are within the measurement uncertainty range and are periods of lower sulfate concentrations than average. For example, lowest sulfate was record near June 19 midnight and $R$ slightly above 2. We don't find the $R$ above 2 points distinctly different from other periods, e.g., enhanced $Na^+$ or $NO_3^-$ was not simultaneously observed, indicating no significant change in aerosol composition (see Figure I below).

Similar results are found for other measurements of $PM_{2.5}$ ions during SOAS, e.g., MARGA data (Allen et al., 2015). The figure below shows good consistency between PILS and MARGA measured $R$ and $Na^+$. MARGA and PILS sulfate and ammonium also agree well; ODR fits, MARGA $SO_4^{2-}$ = (1.00 ± 0.01) PILS-IC $SO_4^{2-}$ + (0.51 ± 0.02 µg m$^{-3}$), $r^2$ = 0.96; MARGA $NH_4^+$ = (1.04 ± 0.01) PILS-IC $NH_4^+$ + (0.21 ± 0.01 µg m$^{-3}$), $r^2$ = 0.91, see our comment to Pye et al. (2017) for a plot.)

The sea-salt episodes mentioned by Allen et al. (2015) are included in our studies. Consistently low pH was predicted despite the occasionally enhanced $Na^+$ level. In response to reviewer 2 question of mixing state, we have added more details on the topic to the manuscript. Finally, including or removing the data when $R$ is over 2 does not change the findings of the paper.

[Figure]

**Figure I**. (a) Comparison of $PM_{2.5}$ PILS and MARGA $Na^+$. (b) Comparison of inferred $Na^+$ (from ion charge balance; $Na^+ = 2SO_4^{2-} + NO_3^- + Cl^- - NH_4^+$, nmol m$^{-3}$) by PILS and MARGA to total measured NVCs by MARGA (represented by $Na^+$), and (c) comparison of PILS and MARGA ammonium-sulfate molar ratios ($R$). Data are from the SOAS study.

**References:**

Allen, H. M., Draper, D. C., Ayres, B. R., Ault, A., Bondy, A., Takahama, S., Modini, R. L., Baumann, K., Edgerton, E., Knote, C., Laskin, A., Wang, B., and Fry, J. L.: Influence of crustal dust and sea spray supermicron particle concentrations and acidity on inorganic $NO_3^-$ aerosol during the 2013 Southern Oxidant and Aerosol Study, Atmospheric Chemistry and Physics, 15, 10669-10685, doi: 10.5194/acp-15-10669-2015, 2015.

Clegg, S. L., Brimblecombe, P., and Wexler, A. S.: Thermodynamic model of the system $H^+-NH_4^+-SO_4^{2-}-NO_3^--H_2O$ at tropospheric temperatures, Journal of Physical Chemistry A, 102, 2137-2154, doi: 10.1021/Jp973042r, 1998.

Guo, H., Xu, L., Bougiatioti, A., Cerully, K. M., Capps, S. L., Hite, J. R., Carlton, A. G., Lee, S. H., Bergin, M. H., Ng, N. L., Nenes, A., and Weber, R. J.: Fine-particle water and pH in the southeastern United States, Atmospheric Chemistry and Physics, 15, 5211-5228, doi: 10.5194/acp-15-5211-2015, 2015.

Guo, H., Liu, J., Froyd, K. D., Roberts, J. M., Veres, P. R., Hayes, P. L., Jimenez, J. L., Nenes, A., and Weber, R. J.: Fine particle pH and gas–particle phase partitioning of inorganic species in Pasadena, California, during the 2010 CalNex campaign, Atmospheric Chemistry and Physics, 17, 5703-5719, doi: 10.5194/acp-17-5703-2017, 2017.

Pye, H. O. T., Zuend, A., Fry, J. L., Isaacman-VanWertz, G., Capps, S. L., Appel, K. W., Foroutan, H., Xu, L., Ng, N. L., and Goldstein, A. H.: Coupling of organic and inorganic aerosol systems and the effect on gas-particle partitioning in the southeastern United States, Atmospheric Chemistry and Physics Discussions, 1-25, doi: 10.5194/acp-2017-623, 2017.

---

## Author Comment (AC2) · 11 Jan 2018

**Responses to Referee #2**

We thank the referee for the thoughtful and constructive comments. Before a point-by-point response to the issues raised, we would like to articulate the main points of our work:

- Including NVCs in the thermodynamic model largely resolves the molar ratio discrepancy, based on our data set, which is representative of the southeast. Only small amounts of NVC are often required, therefore the practice of omitting NVCs from fine mode calculations (which may seem unavoidable for many datasets) induces important biases in molar ratios, which have to be considered in any relevant interpretations (especially on the role of organics).

- The bias in $R$ (ISORROPIA predicted $R$ with $Na^+$ minus ISORROPIA predicted $R$ without $Na^+$, where $R = NH_4^+/SO_4^{2-}$, mole/mole) is highly correlated with measured $Na^+$ ($r^2 = 0.93$), but not correlated with OA mass or OA mass fraction. Furthermore, the difference in observed $R$ from a ratio of 2 ($R$ observed minus 2) is correlated with NVCs and not correlated with OA mass or OA mass fraction. Both results provide strong evidence for NVCs, and not OA, as the underlying driver of the molar ratio discrepancy.

We have addressed the comments (numbered, below), with referee comments in quotes and italics, and our responses in plain text.

1. *"The largest issue is that a fully internal mixture (with all species present in each particle at their bulk atmospheric concentrations) is assumed, meaning that any Na+ is assumed to be present in the same particles that have ammonium, sulfate, and nitrate. Literature from co‐located sampling over the same time period shows that in fact most Na+ present during SOAS was not mixed with SOA, but was present in sea spray aerosol or other mechanically generated particles, such as mineral dust (Allen et al., 2015; Bondy et al., 2017). Thus Na+ cannot explain the values of R that the manuscript is using Na+ to explain. The concept of "inferred‐sodium" is particularly worrisome, as it is not necessarily supported even by bulk measurements and Na+. Another serious concerns is that values of Na+ are reported that are below LOD, which is not appropriate and that the measured Na+ is below the LOD when "inferred Na+" is >4 times higher than a value which is unreliable. Studies going back almost 20 years in Atlanta and the southeast U.S. have shown that NVCs are not present in SOA, particularly with single particle mass spectrometry, which is extremely sensitive to Na+ and K+."*

   Mixing state of NVCs with ammonium, sulfate, and nitrate is a valid concern that we had not addressed in the original study. The assertion that there is *absolutely no mixing* of $Na^+$ with sulfate is incorrect. $Na^+$, which is largely from sea salt (as noted by the reviewer), will contain some fraction of sulfate since it is well known that there is sulfate in sea water (~25% $SO_4^{2-}/Na^+$ mass ratio) (DOE, 1994), apart from any secondary non-sea-salt sulfate or subsequent effects from aerosol aging processes. Nevertheless, we have now considered the impact of incomplete mixing of the non-volatile

inorganic species (sulfate, sodium and other NVC) on $R$ and pH in a separate section in the main text. The analysis shows that with a small fraction of sulfate internally mixed with $Na^+$-rich particles gives acidity and partitioning levels consistent with that of complete internal mixing. The issue of mixing state of NVCs with SOA does not affect our conclusions as well, especially given that the molar ratio discrepancies (one of the main points in our paper) can be explained by inorganic species without any consideration of organic effects. Therefore, effects from incomplete mixing of particles do not alter our conclusions obtained in the original analysis.

The use of data below LOD is justified as the LOD is simply an estimate (i.e. three times of field blanks standard deviation). We note throughout the text when we are using data that is below the LOD. Below we show that the use of inferred $Na^+$ is reasonable as an upper limit of NVCs through comparisons with measurements by other instruments at the SOAS study. Here we show that when measured NVC > measurement LOD (i.e., reliable NVC concentrations, admittedly a small fraction of the study period) there is good agreement between measured and ISORROPIA-predicted $R$ (updated Fig. 1f and Fig. 3, and associated text; no difference at a significance level of $\alpha = 0.05$). Issues arise (which is not surprising) when the NVC concentrations are below or near LOD and they have to be estimated. Most defensible conclusions can be made for periods with the most reliable data (NVCs above LODs). Assessments and critiques that focus solely on only periods of the least reliable data sets (NVCs below LODs) are weak. However, even periods with NVCs below LODs, often provide consistent results. For example:

- For SOAS, when measured $Na^+$ was below LOD (0.07 µg m$^{-3}$), a $Na^+$ of 0.1-0.3 µg m$^{-3}$ was needed to bring measured and predicted $R$ into agreement, which is near the detection limit and so difficult to measure. Despite this, the trends are the same and the predicted $R$ is generally smaller than the observed, and so including the estimated NVCs in this case overcompensates.
- For WINTER data, inferred NVC results in agreement in $R$ prediction with the observation, but inferred $Na^+$ is smaller than observed $PM_1$ $Na^+$, 0.15 vs. 0.23 µg m$^{-3}$, stated in the caption of Figure 4. Again, the results are uncertain since measurements of NVC were not measured online in that study.

The reviewer criticizes the inferred $Na^+$ from ion charge balance to be worrisome and much higher than measured $Na^+$. The inferred $Na^+$ represents an upper limit of NVCs, including $K^+$, $Mg^{2+}$, and $Ca^{2+}$ (if soluble), that could be present in the aqueous aerosols based on charge neutrality of the solution. Therefore, it is supposed to be higher than a single measurement of NVC, such as $Na^+$. We have results from the WINTER study showing the inferred $Na^+$ is comparable to the offline measurement. Also, in Figure I (b), we show a comparison between inferred $Na^+$ from PILS-IC data and MARGA data and measured $Na^+$-equivalent NVCs ($K^+$, $Mg^{2+}$, and $Ca^{2+}$ are represented by $Na^+$) from MARGA for the SOAS study (Allen et al., 2015). Before June 18, the three $Na^+$ levels are similar, indicating the inferred $Na^+$ level is

reasonable. After June 18, the two inferred Na$^+$ levels are higher than MARGA total measured NVCs, but the differences are within uncertainties (propagated from PILS-IC ammonium, sulfate, nitrate, and chloride measurements).

[Figure]

**Figure I**. (a) Comparison of PM$_{2.5}$ PILS and MARGA Na$^+$. (b) Comparison of inferred Na$^+$ (from ion charge balance; Na$^+$ = 2SO$_4^{2-}$ + NO$_3^-$ + Cl$^-$ − NH$_4^+$, nmol m$^{-3}$) by PILS and MARGA to total measured NVCs by MARGA (represented by Na$^+$), and (c) comparison of PILS and MARGA ammonium-sulfate molar ratios ($R$). Data are from the SOAS study.

2. *"Page 4 Lines 33 ‑ 36 and Page 5 lines 1 ‑ 7: Of the three different Na+ levels tested, option 1 infers that any lack of charge balance can be attributed to Na+. On line 35 ‑ 36, the authors then note that inferring the amount of Na+ leads to a value more than 4 times higher than the measured value. There are a number of other possibilities that could explain these strongly different results and inferring shouldn't work as there is almost no Na+ mixed with SOA particles. This is in fact supported by the fact that the reported values for Na+ are below the limit of detection of the measurement (0.06 value when LOD is 0.07)"*

We have addressed this question above. The inferred Na$^+$ is at reasonable levels in both SOAS and WINTER studies.

3. *"Page 5 Lines 2 ‐ 3: The authors choose to use values for Na+ below LOD for the rest of the study, even though they are below LOD. This is a substantial issue and values below LOD should not published or used as the basis for the main analysis over the remainder of the paper."*

We believe there is a slight misinterpretation of results here. First of all, our PILS-IC measurement of $Na^+$ is believed to be accurate; it is in agreement with MARGA data, as showed in Figure I (a) above (this figure has also been added to the supplemental material as Fig. S2). The periods of below $Na^+$ LOD are intended to show that when close to zero $Na^+$ and is included in the thermodynamic model, a near 2 molar ratio ($R$) is predicted. We focus on the periods, where $Na^+$ is above the LOD for added confidence. For these periods, as noted before, we find good agreement between predicted and measured $R$ (no statistical difference at $\alpha = 0.05$). The roles of NVCs and organics on molar ratio are based mainly based on observed data above LOD (Fig. 4). We have added more details on the comparisons of measured and predicted $R$ for various $Na^+$ levels (see added Fig. 3), and included a comparison with MARGA data in the supplemental material (Fig. S2 and S3).

4. *"Page 5 Lines 9 ‐ 15: Most Na+ at SOAS is present from sea spray aerosol (SSA), which has been shown in multiple papers (Allen et al., 2015; Bondy et al., 2017). This authors' observation that Cl ‐ and NO3 ‐ have high R2 values with Na+ and agrees with the papers showing that Na+ is from other sources than SOA, which further highlights that Na+ is not present in SOA particles. It is also worth noting that June 10 ‐ 13 at SOAS (during the period Guo et al focus on) there was a large contribution of sea spray aerosol at Centreville (Bondy et al., 2017). This includes the submicron where 20 ‐ 40% of particles between 200 ‐ 1000 nm were not SOA ‐ dominant particles and are likely where nearly all of the Na+ was located."*

We have addressed this question above.

5. *"Page 6 Lines 7 ‐ 10: The paper here discusses June 11 ‐ 13, which was a high sea salt time period, leading to a more externally mixed aerosol and the Na+ was not present in SOA particles."*

We have addressed this question above. Also note that a higher concentration of sea salt doesn't necessarily indicate a more externally mixed aerosol; the latter depends on the age of the aerosol and the processes that act upon it from emission.

6. *"Page 6 Last Paragraph: While Na+ somewhat tracks with delta R, since Na+ is not in the same particles the paragraph comes to close to attributing causation to a correlation. The wording in this section needs to be weakened. Also, the presumption that organic mass should be correlated with organic film thickness is overstated. Composition, viscosity, etc. all matter when a film might form and OA mass fractions would not provide meaningful estimates of this."*

We don't agree. Our results from assuming an external mixture are consistent with bulk analysis (internal mixture), see added section on internal vs. external mixture. It is true that we have no data on organic film thickness, but if the organic film is sufficient to impede $NH_3$ uptake it *has* to comprise a reasonable fraction of the overall OA (a timescale analysis easily shows this), and concurrently impede equilibration of water and nitrate with the gas phase. We show that neither OA mass or mass fraction is related to observed bias in *R*. If the argument is that an organic film has a widespread effect throughout the southeast, it can't be argued that it is some unique property of the OA (i.e. unique composition, unique viscosity, selectivity with respect to $NH_3$, etc.) unless if a specific mechanism can be proposed to support it.

7. *"Page 7 Lines 10 ‑ 13: The authors mention higher Na+ near coastlines, which is again likely from sea spray and would not be mixed with SOA in the same particles in all likelihood (unfortunately there is not single particle data to provide information on mixing state from that study to my knowledge)."*

We have answered this question above.

8. *"The clarity of the writing, particularly in the introduction, could be improved as there are numerous long sentences and confusing wording that could be improved. Also there numerous missing words, incorrect plural versus singular, and conjugation throughout the manuscript, which needs to be cleaned up"*

We thank the reviewer for pointing this out. The paper has been edited for clarity.

9. *"Page 2 Lines 7 ‑ 11: "Despite its importance, the inability to directly measure fine mode particle pH (e.g. Rindelaub et al. (2016) presents an indirect method that infers particle H+ activity for sizes above 10 μm and requires activity coefficient predicted by a thermodynamic modeling. This method reports the pH for a HSO4 ‑/SO42 ‑ aerosol system similar to the fine particle pH predicted by a thermodynamic modeling used in this study (Guo et al., 2015)) has led to the use of measurable aerosol properties as acidity proxies, such as aerosol ammonium sulfate ratio or ion balances (e.g. (Paulot and Jacob, 2014; Wang et al., 2016; Silvern et al., 2017)). Recent work has shown that acidity proxies are not uniquely related to pH, which in turn strongly questions any conclusions derived from its use. There are numerous reasons why acidity proxies do not represent pH well; they do not capture the variability in particle water content, ion activity coefficients, or partial dissociation of species in the aerosol phase (Guo et al., 2015; Hennigan et al., 2015; Guo et al., 2016)." o There are a number of problems with this section that need to be addressed. o It is not fair to state that the use of the sulfate/bisulfate ratio in Rindelaub et al. led to the use of molar ratios or ion balances, since those have been used for decades and Rindelaub came out in 2016 (Rindelaub et al., 2016). o Secondly, the use of the bisulfate ‑ to ‑ sulfate ratio to determine pH is not indirect and is in fact more direct than thermodynamic modeling of gas particle partitioning, as both species are in the aerosol phase where pH is being determined and the effects of coatings or other nonideal behavior is avoided. The method does use a thermodynamic model to determine activity coefficients, but the fact this is combined with multiple concentrations directly measured in the aerosol phase makes it very different than ion balance or molar ratio methods. o The manuscript states that Rindelaub only used particles above 10 microns, but*

*fails to mention Craig et al. from this summer, which showed this direct method working down to 2‐3 microns for a range of systems (Craig et al., 2017)."*

We have edited the sentence citing Rindelaub et al. (2016) to minimize any confusion.

We clearly have a different understanding of "direct" pH measurement than the reviewer. We cited the method of Rindelaub et al. (2016) as an indirect (but clearly particle-level) measurement of aerosol pH because it doesn't quantify the hydronium ion aqueous phase activity directly. Instead, it infers the $H^+$ concentration by $HSO_4^-/SO_4^{2-}$ ratio, equilibrium constant, and activity coefficients. Deliquesced ambient fine particles are very concentrated liquids. The average ionic strength was 29 mol $L^{-1}$ for this study. Therefore, non-ideality cannot be ignored and avoided by the method of Rindelaub et al. (2016) as the reviewer claims. Furthermore, the activity coefficients of $HSO_4^-$ and $SO_4^{2-}$ cannot be determined simply by $HSO_4^-$ and $SO_4^{2-}$ concentrations because the non-ideal effects are caused by interactions of all water-soluble ions in aqueous aerosols, such as $NH_4^+$ and $NO_3^-$. In terms of determining activity coefficients, a thermodynamic model with an input of all measured inorganic ions should be more accurate than one with only a fraction of the ions input.

We thank the reviewer from bringing attention on Craig et al. (2017) and this paper will be cited in future work.

10. *"Page 2, 2nd Paragraph: The authors go to great lengths to make clear that organic coatings or glassy particles which inhibit equilibrium between gases and particles are not possible based on "established literature", but most of that literature is from the groups who authored this manuscript. If there are limitations to the thermodynamic model approach used in those studies, citing their prior work does not invalidate the other work suggesting films might be important, such as Havala Pye's modeling paper from earlier this year."*

We agree that we mainly cite our past work, because there are very few other detailed assessments of pH predictions by thermodynamic models using in-situ observations. What is meant by "established literature" is that assumption of equilibrium between gas and particle phases is widely used and agrees with observations. Take water vapor for example; LWC is predicted based on the equilibrium assumption and there is a rich body of published literature (spanning decades) comparing predicted and measured LWC. As noted in the paper, given their similar molecular weights and particle uptake properties, it is hard to argue that $NH_3$ and $H_2O$ could interact completely differently with a hypothesized organic film so that equilibrium is not established for one species but established for another. (This is all articulated in the 2nd paragraph of the Introduction).

As for our reported results, we have always tested the thermodynamic predictions in our past studies. We use gas-particle partitioning of semivolatile species that are sensitive to pH, to predict particle pH and compare predicted vs.

measured partitioning, based on the equilibrium assumption, to evaluate pH accuracy. At least in our past analysis, these comparisons show that results based on the equilibrium assumption agree with observations, when RH is sufficiently high (e.g. > 40% or higher) and particles are completely deliquesced. We have cited more papers from other groups (Ansari and Pandis, 2000; Moya et al., 2001; Morino et al., 2006; Liu et al., 2017; Paulot et al., 2017), to strengthen our point, but it nevertheless remains the same.

Pye et al. (2017) is still in review, therefore we do not address the issues raised in that paper. Nevertheless it is important to note it is an equilibrium model study, so kinetic limitations from hypothetical organic films are not important in that study as well.

11. *"Page 3 Line 26: The authors state that "14% of "sulfate" is predicted to be HSO4 ‐ and the rest as SO4 2 ‐ in the winter dataset." Based on a simple acid dissociation constant calculation at pH =1, bisulfate should be >80% of the combination of sulfate and bisulfate. Some explanation should be included to explain why this does not follow basic acid dissociation rules. Is it related to activity coefficients somehow? This need to be addressed in a revised manuscript. Also it is confusing to refer to sulfate, bisulfate, and sulfuric acid together as "total sulfate", as they are each distinct species. Total S(VI) sulfur could work or some other nomenclature."*

The simple calculation as referred by the reviewer is likely based on ideal solutions, where all activity coefficients are treated as one. In this case, $HSO_4^-$ is the dominant form at pH of 1 (shown as dash lines in Figure II). After taking into account of the activity coefficients predicted by ISORROPIA, the curves move left by 2 units (comparing solid vs. dash lines in Figure b). As a result, $SO_4^{2-}$ is the dominant form at pH of 1. Therefore, our statement is consistent with the basic acid dissociation rules. Instead of citing 14% as $HSO_4^-$ from WINTER study, the Figure II has been added to supplemental material to explain how activity coefficients affect the relative fractions of $SO_4^{2-}$ and $HSO_4^-$.

[Figure]

Figure II. Relative fractions of $SO_4^{2-}$ (red) and $HSO_4^-$ (blue) calculated based on ideal solutions (all activity coefficients equal one) and the SOAS non-ideal conditions. The average activity coefficients of $\gamma_{SO_4^{2-}}/\gamma_{HSO_4^-} = 0.01$ are predicted by ISORROPIA for the SOAS fine particles. $\gamma_{H^+} = 1$ is assumed; a smaller $\gamma_{H^+}$ shifts the red and blue curves towards the left, increasing $SO_4^{2-}$ relative fraction at a given pH. The dissociation constant of $HSO_4^-$ is $1.015 \times 10^{-2}$ mol kg$^{-1}$ at 298.15 K (Fountoukis and Nenes, 2007).

12. *"Page 2 Line 9: I believe the word "and" is missing after (Guo et al., 2015)). Also the second parenthesis is not needed."*

    This has been edited.

13. *"Page 3 Line 13: "or if there is free H2SO4 in the aerosol". For the pH values in this manuscript and others using this method, the aerosol acidity is never sufficient for any H2SO4 to exist. Below pH = 2, sulfate will transition to bisulfate, but not sulfuric acid."*

    We agree that at on average pH ~ 1 in this study, free from of $H_2SO_4$ doesn't exist. The sentence in the text aims to explain the theoretical possibilities causing $R = 0$. For example, $NH_4HSO_4$ cannot give $R = 0$. We have revised the sentence to "the lower limit is 0 for $R$ when $SO_4^{2-}$ is associated with other cations instead of $NH_4^+$ (e.g. $Na_2SO_4$) or if there is free $H_2SO_4$ in the aerosol (in theory but not the reason at pH ~ 1 in this study)". Below pH of 2, sulfate transforms to bisulfate; but below pH of -2, bisulfate transforms to sulfuric acid.

14. *"Page 3 Line 14: missing the word "are", should be ", but are rare for"."*

    We have revised accordingly.

15. *"Page 5 Line 39: "mode R with measure Na+ input", should be "measured"."*

    We have revised accordingly.

16. *"A constant throughout the manuscript is that strong statements are supported primarily by prior work from the authors of this study. It would strengthen the manuscript to either make less strong statements or cite work from other groups to support the claims being made."*

    This point is well taken. We have cited more work from other groups. We believe that our statements are justified by our analysis.

**References:**

Allen, H. M., Draper, D. C., Ayres, B. R., Ault, A., Bondy, A., Takahama, S., Modini, R. L., Baumann, K., Edgerton, E., Knote, C., Laskin, A., Wang, B., and Fry, J. L.: Influence of crustal dust and sea spray supermicron particle concentrations and acidity on inorganic $NO_3^-$ aerosol during the 2013 Southern Oxidant and Aerosol Study, Atmospheric Chemistry and Physics, 15, 10669-10685, doi: 10.5194/acp-15-10669-2015, 2015.

Ansari, A. S., and Pandis, S. N.: The effect of metastable equilibrium states on the partitioning of nitrate between the gas and aerosol phases, Atmospheric Environment, 34, 157-168, doi: 10.1016/s1352-2310(99)00242-3, 2000.

Craig, R. L., Nandy, L., Axson, J. L., Dutcher, C. S., and Ault, A. P.: Spectroscopic Determination of Aerosol pH from Acid-Base Equilibria in Inorganic, Organic, and Mixed Systems, J Phys Chem A, 121, 5690-5699, doi: 10.1021/acs.jpca.7b05261, 2017.

DOE: Handbook of Methods for the Analysis of the Various Parameters of the Carbon Dioxide System in Sea Water. Version 2, edited by: Dickson, A. G., and Goyet, C., ORNL/CDIAC-74, 1994.

Fountoukis, C., and Nenes, A.: ISORROPIA II: a computationally efficient thermodynamic equilibrium model for $K^+$-$Ca^{2+}$-$Mg^{2+}$-$NH_4^+$-$Na^+$-$SO_4^{2-}$-$NO_3^-$-$Cl^-$-$H_2O$ aerosols, Atmospheric Chemistry and Physics, 7, 4639-4659, doi: 10.5194/acp-7-4639-2007, 2007.

Guo, H., Xu, L., Bougiatioti, A., Cerully, K. M., Capps, S. L., Hite, J. R., Carlton, A. G., Lee, S. H., Bergin, M. H., Ng, N. L., Nenes, A., and Weber, R. J.: Fine-particle water and pH in the southeastern United States, Atmospheric Chemistry and Physics, 15, 5211-5228, doi: 10.5194/acp-15-5211-2015, 2015.

Guo, H., Liu, J., Froyd, K. D., Roberts, J. M., Veres, P. R., Hayes, P. L., Jimenez, J. L., Nenes, A., and Weber, R. J.: Fine particle pH and gas–particle phase partitioning of inorganic species in Pasadena, California, during the 2010 CalNex campaign, Atmospheric Chemistry and Physics, 17, 5703-5719, doi: 10.5194/acp-17-5703-2017, 2017.

Liu, M., Song, Y., Zhou, T., Xu, Z., Yan, C., Zheng, M., Wu, Z., Hu, M., Wu, Y., and Zhu, T.: Fine particle pH during severe haze episodes in northern China, Geophysical Research Letters, 44, 5213-5221, doi: 10.1002/2017gl073210, 2017.

Morino, Y., Kondo, Y., Takegawa, N., Miyazaki, Y., Kita, K., Komazaki, Y., Fukuda, M., Miyakawa, T., Moteki, N., and Worsnop, D. R.: Partitioning of $HNO_3$ and particulate nitrate over Tokyo: Effect of vertical mixing, Journal of Geophysical Research, 111, doi: 10.1029/2005jd006887, 2006.

Moya, M., Ansari, A. S., and Pandis, S. N.: Partitioning of nitrate and ammonium between the gas and particulate phases during the 1997 IMADA-AVER study in Mexico City, Atmospheric Environment, 35, 1791-1804, doi: 10.1016/s1352-2310(00)00292-2, 2001.

Paulot, F., Paynter, D., Ginoux, P., Naik, V., Whitburn, S., Van Damme, M., Clarisse, L., Coheur, P. F., and Horowitz, L. W.: Gas-aerosol partitioning of ammonia in biomass burning plumes: Implications for the interpretation of spaceborne observations of ammonia and the radiative forcing of ammonium nitrate, Geophysical Research Letters, doi: 10.1002/2017gl074215, 2017.

Pye, H. O. T., Zuend, A., Fry, J. L., Isaacman-VanWertz, G., Capps, S. L., Appel, K. W., Foroutan, H., Xu, L., Ng, N. L., and Goldstein, A. H.: Coupling of organic and inorganic aerosol systems and the effect on gas-particle partitioning in the southeastern United States, Atmospheric Chemistry and Physics Discussions, 1-25, doi: 10.5194/acp-2017-623, 2017.

Rindelaub, J. D., Craig, R. L., Nandy, L., Bondy, A. L., Dutcher, C. S., Shepson, P. B., and Ault, A. P.: Direct Measurement of pH in Individual Particles via Raman Microspectroscopy and Variation in Acidity with Relative Humidity, J Phys Chem A, 120, 911-917, doi: 10.1021/acs.jpca.5b12699, 2016.

---

## Author Response (AR1)

**Style Definition:** Heading 2: Font: Not Italic

[revised manuscript text omitted]

---

## Referee Report (RR1)

**Responses to Referee #1**

We thank the referee for the thoughtful and constructive comments. Before a point-by-point response to the issues raised, we would like to articulate the main points of our work:

- Including NVCs in the thermodynamic model largely resolves the molar ratio discrepancy, based on our data set, which is representative of the southeast. Only small amounts of NVC are often required, therefore the practice of omitting NVCs from fine mode calculations (which may seem unavoidable for many datasets) induces important biases in molar ratios, which have to be considered in any relevant interpretations (especially on the role of organics).

- The bias in $R$ (ISORROPIA predicted $R$ with $Na^+$ minus ISORROPIA predicted $R$ without $Na^+$, where $R = NH_4^+/SO_4^{2-}$, mole/mole) is highly correlated with measured $Na^+$ ($r^2 = 0.93$), but not correlated with OA mass or OA mass fraction. Furthermore, the difference in observed $R$ from a ratio of 2 ($R$ observed minus 2) is correlated with NVCs and not correlated with OA mass or OA mass fraction. Both results provide strong evidence for NVCs, and not OA, as the underlying driver of the molar ratio discrepancy.

We have addressed the comments (numbered, below), with referee comments in quotes and italics, and our responses in plain text.

1. *"To explain this discrepancy, mainly two hypothesis are proposed, namely the organic-film hypothesis (Pye et al., 2017) and the non-volatile cations (NVC) hypothesis (as shown in this manuscript). By including in the measured NVC, the authors could now decrease predicted R from 1.97 to 1.85, which is still higher than the corresponding observation of 1.7. The remaining difference could possibly be due to the presence of organic-film, or the size heterogeneity. Considering the large disagreement in observation data, neither of the above hypothesis could be fully validated."*

The above statistics are for the whole Fig. 1 period including many data points where we had to estimate $Na^+$ since the measured $Na^+$ was below our LOD. Focusing only on the periods with measured $Na^+$ above LOD (reliable NVC concentrations), there is no statistical difference between predicted and measured $R$ (t- test $\alpha = 0.05$).

See also our response to the comments from Daniel Jacob and Rachel Silvern (Figure I in that response), where we also discuss differences in $R$ between model-predicted and observations and also point out that a significant fraction of the differences in $R$ between some data sets (e.g., AMS $PM_1$ vs various $PM_{2.5}$ data) can be attributed to the differences in measured particle size ranges.

In summary, our NVC analysis can fully explain the discrepancy in molar ratio predictions for either PM$_{2.5}$ or PM$_1$ data  sets.

I don't think that the authors have satisfactorily addressed in the text the much lower *R* values found in the CSN data, which cannot be explained by the NVC hypothesis.

2. *"What's the average activity coefficient of NH3·uH2O(aq) and NH4+? Does that change with NVC levels? If so, how would the theoretical S-curve be influenced, or what's the potential range of S-curve in this study? In comparison, the S-curve range based on the activity coefficient of H+ as given in Pye et al. (2017) should also be indicated"*

The activity coefficient of dissolved NH$_3$, $\gamma(NH_3)$, has a negligible effect on the S curve  and so not considered (Guo et al., 2017). For example, at 298 K, the acid dissociation constant of NH$_4^+$, $K_a = 5.69 \times 10^{-10}$ mole L$^{-1}$ (Clegg et al., 1998),

results in $\dfrac{K_a}{\gamma_{NH_4^+}/\gamma_{NH_3}} \ll \gamma_{H^+}[H^+]$ as long as the solution is not too basic. SOAS fine particles were very acidic with pH on average $0.94 \pm 0.59$ (SD). The measured Na$^+$ (above zero) for the SOAS study doesn't change $\dfrac{\gamma_{H^+}}{\gamma_{NH_4^+}}$ significantly; including or excluding Na$^+$ gives the same $\dfrac{\gamma_{H^+}}{\gamma_{NH_4^+}}$ of $1.38 \pm 0.12$ (no statistical difference as confirmed by t-test at $\alpha = 0.05$).

I agree with the authors.

3. *"At high or low pH ranges, the partitioning fraction of NH3(g) can be extremely low or large, but can never reach 0% or 100%. What's the accuracy of the ISORROPIA model? Or, at what value would the model treat the ratio actually as 0% or 100%? Since the observation data can never be zero, what's the discrepancy of predicted NH3 and observation NH3 at those extreme conditions, for gas- and aerosol-phase respectively? Similarly, how about the HNO3-NO3- pair?"*

Theoretically, the partitioning fraction of NH$_3$ may never be 0 or 100%, but practically this is not an issue. We only use the semivolatile pairs with fractions close to 50% to constrain our pH predictions since this is the region of greatest sensitivity (e.g. (Guo et al., 2015)). Propagated uncertainty in the partitioning fraction can be determined from both gas and particle measurement uncertainties. The average propagated uncertainty in $\varepsilon(NH_4^+)$ is ~4% (absolute value, not percentage of $\varepsilon(NH_4^+)$) for SOAS, the pH prediction is accurate within 0.08 for $\varepsilon(NH_4^+)$ at 50% and 0.22 for $\varepsilon(NH_4^+)$ at 10%. A similar result is derived for HNO$_3$-NO$_3^-$ partitioning (0.07 for $\varepsilon(NO_3^-)$ at 50% and 0.22 for $\varepsilon(NO_3^-)$ at 10%).

I agree with the authors.

4. *"Adding Fig. 3 in the authors' comment to Pye et al. (2017) would help improve the current manuscript. To my eye, the theoretical S-curve in that figure is to the right edges of the corresponding observation data. What if the aerosol water associated with organics are taken into account? That dilution effect would increase pH, shift the corresponding observation data points to the right and may result in better agreement. In addition, the authors claim that*

*corresponding S-curve of Pye et al. (2017) can be derived by shift the S-curve of 0.8 pH units. This argument looks confusing and should be better described."*

To actually do this properly we need the Pye et al. (2017) data set, which is not yet available. Furthermore, this paper does not directly address the claims of Pye et al. (2017).

I don't have any comment on this. It seemed to just be a suggestion by the reviewer.

5. *"The authors attributed the data with R over 2 to "measurement uncertainty and error propagation at low SO42-concentrations". However, based on data shown in Figure 1, these periods are not the periods with the lowest SO42-concentration (and thus largest uncertainty). Also, these periods correspond to periods with negative inferred Na+. The arbitrary exclusion of these data is problematic. Basically that is to say that ambient aerosols can never be neutral or basic. As mentioned in other papers (Allen et al., 2015), sometimes the sea-salt episodes can be observed. How could the authors prove that cation-abundant situations are wrong? Do those data have any common distinct features from others? The data can be discarded for better reasons, not just due to that they look abnormal."*

The observed PILS-IC data points with $R$ over 2 are within the measurement uncertainty range and are periods of lower sulfate concentrations than average. For example, lowest sulfate was record near June 19 midnight and $R$ slightly above 2. We don't find the $R$ above 2 points distinctly different from other periods, e.g., enhanced $Na^+$ or $NO_3^-$ was not simultaneously observed, indicating no significant change in aerosol composition (see Figure I below).

Similar results are found for other measurements of $PM_{2.5}$ ions during SOAS, e.g., MARGA data (Allen et al., 2015). The figure below shows good consistency between PILS and MARGA measured $R$ and $Na^+$. MARGA and PILS sulfate and ammonium also agree well; ODR fits, MARGA $SO_4^{2-} = (1.00 \pm 0.01)$ PILS-IC $SO_4^{2-} + (0.51 \pm 0.02 \text{ µg m}^{-3})$, $r^2 = 0.96$; MARGA $NH_4^+ = (1.04 \pm 0.01)$ PILS-IC $NH_4^+ + (0.21 \pm 0.01 \text{ µg m}^{-3})$, $r^2 = 0.91$, see our comment to Pye et al. (2017) for a plot.)

The sea-salt episodes mentioned by Allen et al. (2015) are included in our studies. Consistently low pH was predicted despite the occasionally enhanced $Na^+$ level. In response to reviewer 2 question of mixing state, we have added more details on the topic to the manuscript. Finally, including or removing the data when $R$ is over 2 does not change the findings of the paper.

I agree with the authors.

[Figure]

**Figure I**. (a) Comparison of PM$_{2.5}$ PILS and MARGA Na$^+$. (b) Comparison of inferred Na$^+$ (from ion charge balance; Na$^+$ = 2SO$_4^{2-}$ + NO$_3^-$ + Cl$^-$ − NH$_4^+$, nmol m$^{-3}$) by PILS and MARGA to total measured NVCs by MARGA (represented by Na$^+$), and (c) comparison of PILS and MARGA ammonium-sulfate molar ratios ($R$). Data are from the SOAS study.

**References:**

Allen, H. M., Draper, D. C., Ayres, B. R., Ault, A., Bondy, A., Takahama, S., Modini, R. L., Baumann, K., Edgerton, E., Knote, C., Laskin, A., Wang, B., and Fry, J. L.: Influence of crustal dust and sea spray supermicron particle concentrations and acidity on inorganic NO$_3^-$ aerosol during the 2013 Southern Oxidant and Aerosol Study, Atmospheric Chemistry and Physics, 15, 10669-10685, doi: 10.5194/acp-15-10669-2015, 2015.

Clegg, S. L., Brimblecombe, P., and Wexler, A. S.: Thermodynamic model of the system H$^+$-NH$_4^-$SO$_4^{2-}$-NO$_3^-$-H$_2$O at tropospheric temperatures, Journal of Physical Chemistry A, 102, 2137-2154, doi: 10.1021/Jp973042r, 1998.

Guo, H., Xu, L., Bougiatioti, A., Cerully, K. M., Capps, S. L., Hite, J. R., Carlton, A. G., Lee, S. H., Bergin, M. H., Ng, N. L., Nenes, A., and Weber, R. J.: Fine-particle water and pH in the southeastern United States, Atmospheric Chemistry and Physics, 15, 5211-5228, doi: 10.5194/acp-15-5211-2015, 2015.

Guo, H., Liu, J., Froyd, K. D., Roberts, J. M., Veres, P. R., Hayes, P. L., Jimenez, J. L., Nenes, A., and Weber, R. J.: Fine particle pH and gas–particle phase partitioning of inorganic species in Pasadena, California, during the 2010 CalNex campaign, Atmospheric Chemistry and Physics, 17, 5703-5719, doi: 10.5194/acp-17-5703-2017, 2017.

Pye, H. O. T., Zuend, A., Fry, J. L., Isaacman-VanWertz, G., Capps, S. L., Appel, K. W., Foroutan, H., Xu, L., Ng, N. L., and Goldstein, A. H.: Coupling of organic and inorganic aerosol systems and the effect on gas-particle partitioning in the southeastern United States, Atmospheric Chemistry and Physics Discussions, 1-25, doi: 10.5194/acp-2017-623, 2017

**. Responses to Referee #2**

We thank the referee for the thoughtful and constructive comments. Before a point-by-point response to the issues raised, we would like to articulate the main points of our work:

- Including NVCs in the thermodynamic model largely resolves the molar ratio discrepancy, based on our data set, which is representative of the southeast. Only small amounts of NVC are often required, therefore the practice of omitting NVCs from fine mode calculations (which may seem unavoidable for many datasets) induces important biases in molar ratios, which have to be considered in any relevant interpretations (especially on the role of organics).

- The bias in $R$ (ISORROPIA predicted $R$ with $Na^+$ minus ISORROPIA predicted $R$ without $Na^+$, where $R =$ $NH_4^+/SO_4^{2-}$, mole/mole) is highly correlated with measured $Na^+$ ($r^2 = 0.93$), but not correlated with OA mass or OA mass fraction. Furthermore, the difference in observed $R$ from a ratio of 2 ($R$ observed minus 2) is correlated with NVCs and not correlated with OA mass or OA mass fraction. Both results provide strong evidence for NVCs, and not OA, as the underlying driver of the molar ratio discrepancy.

We have addressed the comments (numbered, below), with referee comments in quotes and italics, and our responses in plain text.

1. *"The largest issue is that a fully internal mixture (with all species present in each particle at their bulk atmospheric concentrations) is assumed, meaning that any Na+ is assumed to be present in the same particles that have ammonium, sulfate, and nitrate. Literature from co‐located sampling over the same time period shows that in fact most Na+ present during SOAS was not mixed with SOA, but was present in sea spray aerosol or other mechanically generated particles, such as mineral dust (Allen et al., 2015; Bondy et al., 2017). Thus Na+ cannot explain the values of R that the manuscript is using Na+ to explain. The concept of "inferred‐sodium" is particularly worrisome, as it is not necessarily supported even by bulk measurements and Na+. Another serious concerns is that values of Na+ are reported that are below LOD, which is not appropriate and that the measured Na+ is below the LOD when "inferred Na+" is >4 times higher than a value which is unreliable. Studies going back almost 20 years in Atlanta and the southeast U.S. have shown that NVCs are not present in SOA, particularly with single particle mass spectrometry, which is extremely sensitive to Na+ and K+."*

   Mixing state of NVCs with ammonium, sulfate, and nitrate is a valid concern that we had not addressed in the original study. The assertion that there is *absolutely no mixing* of $Na^+$ with sulfate is incorrect. $Na^+$, which is largely from sea salt (as noted by the reviewer), will contain some fraction of sulfate since it is well known that there is sulfate in sea water (~25% $SO_4^{2-}/Na^+$ mass ratio) (DOE, 1994), apart from any secondary non-sea-salt sulfate or subsequent effects from aerosol aging processes.

But isn't standard protocol to remove that sea-salt sulfate and just report non-sea-salt sulfate? Just checking, I know it's standard protocol in research data sets.

Nevertheless, we have now considered the impact of incomplete mixing of the non-volatile

inorganic species (sulfate, sodium and other NVC) on $R$ and pH in a separate section in the main text. The analysis shows that with a small fraction of sulfate internally mixed with $Na^+$-rich particles gives acidity and partitioning levels consistent with that of complete internal mixing. The issue of mixing state of NVCs with SOA does not affect our conclusions as well, especially given that the molar ratio discrepancies (one of the main points in our paper) can be explained by inorganic species without any consideration of organic effects. Therefore, effects from incomplete mixing of particles do not alter our conclusions obtained in the original analysis.

I agree with the authors that internal vs. external mixing shouldn't have a large effect on $R$ and pH. That calculation is a nice addition to the paper.

The use of data below LOD is justified as the LOD is simply an estimate (i.e. three times of field blanks standard deviation). We note throughout the text when we are using data that is below the LOD. Below we show that the use of inferred $Na^+$ is reasonable as an upper limit of NVCs through comparisons with measurements by other instruments at the SOAS study. Here we show that when measured NVC > measurement LOD (i.e., reliable NVC concentrations, admittedly a small fraction of the study period) there is good agreement between measured and ISORROPIA-predicted $R$ (updated Fig. 1f and Fig. 3, and associated text; no difference at a significance level of $\alpha = 0.05$).

But then of course $R$ would be less than 2. These are unusual conditions when NVCs are high.

Issues arise (which is not surprising) when the NVC concentrations are below or near LOD and they have to be estimated. Most defensible conclusions can be made for periods with the most reliable data (NVCs above LODs). Assessments and critiques that focus solely on only periods of the least reliable data sets (NVCs below LODs) are weak. However, even periods with NVCs below LODs, often provide consistent results. For example:
-   For SOAS, when measured $Na^+$ was below LOD (0.07 µg m$^{-3}$), a $Na^+$ of 0.1-0.3 µg m$^{-3}$ was needed to bring measured and predicted $R$ into agreement, which is near the detection limit and so difficult to measure. Despite this, the trends are the same and the predicted $R$ is generally smaller than the observed, and so including the estimated NVCs in this case overcompensates.
-   For WINTER data, inferred NVC results in agreement in $R$ prediction with the observation, but inferred $Na^+$ is smaller than observed PM$_1$ $Na^+$, 0.15 vs. 0.23 µg m$^{-3}$, stated in the caption of Figure 4. Again, the results are uncertain since measurements of NVC were not measured online in that study.

The reviewer criticizes the inferred $Na^+$ from ion charge balance to be worrisome and much higher than measured $Na^+$. The inferred $Na^+$ represents an upper limit of NVCs, including $K^+$, $Mg^{2+}$, and $Ca^{2+}$ (if soluble), that could be present in

the aqueous aerosols based on charge neutrality of the solution. Therefore, it is supposed to be higher than a single measurement of NVC, such as $Na^+$. We have results from the WINTER study showing the inferred $Na^+$ is comparable to the offline measurement. Also, in Figure I (b), we show a comparison between inferred $Na^+$ from PILS-IC data and MARGA data and measured $Na^+$-equivalent NVCs ($K^+$, $Mg^{2+}$, and $Ca^{2+}$ are represented by $Na^+$) from MARGA for the SOAS study (Allen et al., 2015). Before June 18, the three $Na^+$ levels are similar, indicating the inferred $Na^+$ level is

reasonable. After June 18, the two inferred Na+ levels are higher than MARGA total measured NVCs, but the differences are within uncertainties (propagated from PILS-IC ammonium, sulfate, nitrate, and chloride measurements).

I agree with the reviewer that inferred Na+ is highly problematic and I don't see that the authors have addressed that concern in the revised text. A major problem is that the charge balance equation used to infer Na+ doesn't include H+ and thus forces H+ to be low so $R$ to be high resulting in a circular argument. The authors justify this by arguing that H+ is very low compared to other cations but that is based on their thermodynamic calculation for H+ assumed to be correct (note that in their example [NH4+] >> [H+], effectively meaning R close to 2), so it is self-fulfilling.

So a problem with this paper right now is that it pushes its argument either by presenting unusual situations where Na+ is above LOD or by forcing the argument to be correct through the upper limit of inferred Na+.

[Figure]

**Figure I**. (a) Comparison of PM$_{2.5}$ PILS and MARGA Na+. (b) Comparison of inferred Na+ (from ion charge balance; Na+ = 2SO$_4^{2-}$ + NO$_3^-$ + Cl$^-$ − NH$_4^+$, nmol m$^{-3}$) by PILS and MARGA to total measured NVCs by MARGA (represented by Na+), and (c) comparison of PILS and MARGA ammonium-sulfate molar ratios ($R$). Data are from the SOAS study.

2. *"Page 4 Lines 33 ‑ 36 and Page 5 lines 1 ‑ 7: Of the three different Na+ levels tested, option 1 infers that any lack of charge balance can be attributed to Na+. On line 35 ‑ 36, the authors then note that inferring the amount of Na+ leads to a value more than 4 times higher than the measured value. There are a number of other possibilities that could explain these strongly different results and inferring shouldn't work as there is almost no Na+ mixed with SOA particles. This is in fact supported by the fact that the reported values for Na+ are below the limit of detection of the measurement (0.06 value when LOD is 0.07)"*

We have addressed this question above. The inferred $Na^+$ is at reasonable levels in both SOAS and WINTER studies.

I don't see that the authors have addressed that issue in the revised text.

3. *"Page 5 Lines 2 - 3: The authors choose to use values for Na+ below LOD for the rest of the study, even though they are below LOD. This is a substantial issue and values below LOD should not published or used as the basis for the main analysis over the remainder of the paper."*

We believe there is a slight misinterpretation of results here. First of all, our PILS-IC measurement of $Na^+$ is believed to be accurate; it is in agreement with MARGA data, as showed in Figure I (a) above (this figure has also been added to the supplemental material as Fig. S2). The periods of below $Na^+$ LOD are intended to show that when close to zero $Na^+$ and is included in the thermodynamic model, a near 2 molar ratio ($R$) is predicted. We focus on the periods, where $Na^+$ is above the LOD for added confidence. For these periods, as noted before, we find good agreement between predicted and measured $R$ (no statistical difference at $\alpha = 0.05$). The roles of NVCs and organics on molar ratio are based mainly based on observed data above LOD (Fig. 4). We have added more details on the comparisons of measured and predicted $R$ for various $Na^+$ levels (see added Fig. 3), and included a comparison with MARGA data in the supplemental material (Fig. S2 and S3).

I agree with the reviewer and I don't think that the authors have addressed the issue satisfactorily. Conditions where $Na^+$ > LOD are unusual and conducive to their argument. Conditions where $Na^+$ < LOD should indeed not be used and the inferred $Na^+$ seems misleading.

4. *"Page 5 Lines 9 - 15: Most Na+ at SOAS is present from sea spray aerosol (SSA), which has been shown in multiple papers (Allen et al., 2015; Bondy et al., 2017). This authors' observation that Cl - and NO3 - have high R2 values with Na+ and agrees with the papers showing that Na+ is from other sources than SOA, which further highlights that Na+ is not present in SOA particles. It is also worth noting that June 10 - 13 at SOAS (during the period Guo et al focus on) there was a large contribution of sea spray aerosol at Centreville (Bondy et al., 2017). This includes the submicron where 20 - 40% of particles between 200 - 1000 nm were not SOA - dominant particles and are likely where nearly all of the Na+ was located."*

We have addressed this question above.

5. *"Page 6 Lines 7 - 10: The paper here discusses June 11 - 13, which was a high sea salt time period, leading to a more externally mixed aerosol and the Na+ was not present in SOA particles."*

We have addressed this question above. Also note that a higher concentration of sea salt doesn't necessarily indicate a more externally mixed aerosol; the latter depends on the age of the aerosol and the processes that act upon it from emission.

6. *"Page 6 Last Paragraph: While Na+ somewhat tracks with delta R, since Na+ is not in the same particles the*

*paragraph comes to close to attributing causation to a correlation. The wording in this section needs to be weakened. Also, the presumption that organic mass should be correlated with organic film thickness is overstated. Composition, viscosity, etc. all matter when a film might form and OA mass fractions would not provide meaningful estimates of this."*

We don't agree. Our results from assuming an external mixture are consistent with bulk analysis (internal mixture), see added section on internal vs. external mixture. It is true that we have no data on organic film thickness, but if the organic film is sufficient to impede NH$_3$ uptake it *has* to comprise a reasonable fraction of the overall OA (a timescale analysis easily shows this), and concurrently impede equilibration of water and nitrate with the gas phase. We show that neither OA mass or mass fraction is related to observed bias in *R*. If the argument is that an organic film has a widespread effect throughout the southeast, it can't be argued that it is some unique property of the OA (i.e. unique composition, unique viscosity, selectivity with respect to NH$_3$, etc.) unless if a specific mechanism can be proposed to support it.

I agree with the reviewer and think that the authors make too much hay off this organic film hypothesis. It seems to have been suggested by Silvern et al. as a speculative explanation for the low *R* in the CSN and AMS data, but here the paper misleadingly characterizes Silvern et al. as being all about the organic film hypothesis and misses their main point which was to draw attention to the low *R* in the CSN data (a problem ignored by this paper).

7. *"Page 7 Lines 10 ‑ 13: The authors mention higher Na+ near coastlines, which is again likely from sea spray and would not be mixed with SOA in the same particles in all likelihood (unfortunately there is not single particle data to provide information on mixing state from that study to my knowledge)."*

We have answered this question above.

8. *"The clarity of the writing, particularly in the introduction, could be improved as there are numerous long sentences and confusing wording that could be improved. Also there numerous missing words, incorrect plural versus singular, and conjugation throughout the manuscript, which needs to be cleaned up"*

We thank the reviewer for pointing this out. The paper has been edited for clarity.

9. *"Page 2 Lines 7 ‑ 11: "Despite its importance, the inability to directly measure fine mode particle pH (e.g. Rindelaub et al. (2016) presents an indirect method that infers particle H+ activity for sizes above 10 μm and requires activity coefficient predicted by a thermodynamic modeling. This method reports the pH for a HSO4 ‑/SO42 ‑ aerosol system similar to the fine particle pH predicted by a thermodynamic modeling used in this study (Guo et al., 2015)) has led to the use of measurable aerosol properties as acidity proxies, such as aerosol ammonium sulfate ratio or ion balances (e.g. (Paulot and Jacob, 2014; Wang et al., 2016; Silvern et al., 2017)). Recent work has shown that acidity proxies are not uniquely related to pH, which in turn strongly questions any conclusions derived from its use. There are numerous reasons why acidity proxies do not represent pH well; they do not capture the variability in particle water content, ion activity coefficients, or partial dissociation of species in the aerosol phase (Guo et al., 2015; Hennigan et al., 2015; Guo et al., 2016)." o There are a number of problems with this section that need to be addressed. o It is not fair to state that the use of the sulfate/bisulfate ratio in Rindelaub et al. led to the use of molar ratios or ion balances, since those have been used for decades and Rindelaub came out in 2016 (Rindelaub et al., 2016). o Secondly, the use of the bisulfate ‑*

*to‑sulfate ratio to determine pH is not indirect and is in fact more direct than thermodynamic modeling of gas particle partitioning, as both species are in the aerosol phase where pH is being determined and the effects of coatings or other nonideal behavior is avoided. The method does use a thermodynamic model to determine activity coefficients, but the fact this is combined with multiple concentrations directly measured in the aerosol phase makes it very different than ion balance or molar ratio methods. o The manuscript states that Rindelaub only used particles above 10 microns, but*

*fails to mention Craig et al. from this summer, which showed this direct method working down to 2 ‑ 3 microns for a range of systems (Craig et al., 2017)."*

We have edited the sentence citing Rindelaub et al. (2016) to minimize any confusion.

We clearly have a different understanding of "direct" pH measurement than the reviewer. We cited the method of Rindelaub et al. (2016) as an indirect (but clearly particle-level) measurement of aerosol pH because it doesn't quantify the hydronium ion aqueous phase activity directly. Instead, it infers the $H^+$ concentration by $HSO_4^-/SO_4^{2-}$ ratio, equilibrium constant, and activity coefficients. Deliquesced ambient fine particles are very concentrated liquids. The average ionic strength was 29 mol $L^{-1}$ for this study. Therefore, non-ideality cannot be ignored and avoided by the method of Rindelaub et al. (2016) as the reviewer claims. Furthermore, the activity coefficients of $HSO_4^-$ and $SO_4^{2-}$ cannot be determined simply by $HSO_4^-$ and $SO_4^{2-}$ concentrations because the non-ideal effects are caused by interactions of all water-soluble ions in aqueous aerosols, such as $NH_4^+$ and $NO_3^-$. In terms of determining activity coefficients, a thermodynamic model with an input of all measured inorganic ions should be more accurate than one with only a fraction of the ions input.

We thank the reviewer from bringing attention on Craig et al. (2017) and this paper will be cited in future work.

The authors again seem to make misleading claims to dismiss previous literature – here that they used *R* as an acidity proxy. They did not.

10. *"Page 2, 2nd Paragraph: The authors go to great lengths to make clear that organic coatings or glassy particles which inhibit equilibrium between gases and particles are not possible based on "established literature", but most of that literature is from the groups who authored this manuscript. If there are limitations to the thermodynamic model approach used in those studies, citing their prior work does not invalidate the other work suggesting films might be important, such as Havala Pye's modeling paper from earlier this year."*

We agree that we mainly cite our past work, because there are very few other detailed assessments of pH predictions by thermodynamic models using in-situ observations. What is meant by "established literature" is that assumption of equilibrium between gas and particle phases is widely used and agrees with observations. Take water vapor for example; LWC is predicted based on the equilibrium assumption and there is a rich body of published literature (spanning decades) comparing predicted and measured LWC. As noted in the paper, given their similar molecular weights and particle uptake properties, it is hard to argue that $NH_3$ and $H_2O$ could interact completely differently with a hypothesized organic film so that equilibrium is not established for one species but established for another. (This is all articulated in the 2$^{nd}$ paragraph of the Introduction).

Again, the authors misleadingly describe Silvern et al. as claiming that the organic film limitation would apply to $NH_3$ but not to $H_2O$ and $HNO_3$. They said nothing of the sort and instead pointed out that the organic film limitation hypothesis was

problematic precisely because it would have to also apply to $H_2O$ and $HNO_3$.

As for our reported results, we have always tested the thermodynamic predictions in our past studies. We use gas-particle partitioning of semivolatile species that are sensitive to pH, to predict particle pH and compare predicted vs.

measured partitioning, based on the equilibrium assumption, to evaluate pH accuracy. At least in our past analysis, these comparisons show that results based on the equilibrium assumption agree with observations, when RH is sufficiently high (e.g. > 40% or higher) and particles are completely deliquesced. We have cited more papers from other groups (Ansari and Pandis, 2000; Moya et al., 2001; Morino et al., 2006; Liu et al., 2017; Paulot et al., 2017), to strengthen our point, but it nevertheless remains the same.

Pye et al. (2017) is still in review, therefore we do not address the issues raised in that paper. Nevertheless it is important to note it is an equilibrium model study, so kinetic limitations from hypothetical organic films are not important in that study as well.

I agree with the reviewer that the authors' propensity to cite their own work and to dismiss others' borders on the embarrassing. I don't think that they fixed this in revision. The message one gets from the paper is that the authors are the only ones who understand particle thermodynamics and acidity, and everyone else doesn't know what they're doing; that doesn't come across very well.

11. *"Page 3 Line 26: The authors state that "14% of "sulfate" is predicted to be HSO4 ⁻ and the rest as SO4 2 ⁻ in the winter dataset." Based on a simple acid dissociation constant calculation at pH =1, bisulfate should be >80% of the combination of sulfate and bisulfate. Some explanation should be included to explain why this does not follow basic acid dissociation rules. Is it related to activity coefficients somehow? This need to be addressed in a revised manuscript. Also it is confusing to refer to sulfate, bisulfate, and sulfuric acid together as "total sulfate", as they are each distinct species. Total S(VI) sulfur could work or some other nomenclature."*

The simple calculation as referred by the reviewer is likely based on ideal solutions, where all activity coefficients are treated as one. In this case, $HSO_4^-$ is the dominant form at pH of 1 (shown as dash lines in Figure II). After taking into account of the activity coefficients predicted by ISORROPIA, the curves move left by 2 units (comparing solid vs. dash lines in Figure b). As a result, $SO_4^{2-}$ is the dominant form at pH of 1. Therefore, our statement is consistent with the basic acid dissociation rules. Instead of citing 14% as $HSO_4^-$ from WINTER study, the Figure II has been added to supplemental material to explain how activity coefficients affect the relative fractions of $SO_4^{2-}$ and $HSO_4^-$.

[Figure]

Figure II. Relative fractions of $SO_4^{2-}$ (red) and $HSO_4^-$ (blue) calculated based on ideal solutions (all activity coefficients equal one) and the SOAS non-ideal conditions. The average activity coefficients of $\gamma_{SO_4^-}/\gamma_{HSO_4} = 0.01$ are predicted by ISORROPIA for the SOAS fine particles. $\gamma_{H+} = 1$ is assumed; a smaller $\gamma_{H+}$ shifts the red and blue curves towards the left, increasing $SO_4^{2-}$ relative fraction at a given pH. The dissociation constant of $HSO_4^-$ is $1.015\times10^{-2}$ mol kg$^{-1}$ at 298.15 K (Fountoukis and Nenes, 2007).

12. *"Page 2 Line 9: I believe the word "and" is missing after (Guo et al., 2015)). Also the second parenthesis is not needed."*

This has been edited.

13. *"Page 3 Line 13: "or if there is free H2SO4 in the aerosol". For the pH values in this manuscript and others using this method, the aerosol acidity is never sufficient for any H2SO4 to exist. Below pH = 2, sulfate will transition to bisulfate, but not sulfuric acid."*

We agree that at on average pH ~ 1 in this study, free from of $H_2SO_4$ doesn't exist. The sentence in the text aims to explain the theoretical possibilities causing $R = 0$. For example, $NH_4HSO_4$ cannot give $R = 0$. We have revised the sentence to "the lower limit is 0 for $R$ when $SO_4^{2-}$ is associated with other cations instead of $NH_4^+$ (e.g. $Na_2SO_4$) or if there is free $H_2SO_4$ in the aerosol (in theory but not the reason at pH ~ 1 in this study)". Below pH of 2, sulfate transforms to bisulfate; but below pH of -2, bisulfate transforms to sulfuric acid.

14. *"Page 3 Line 14: missing the word "are", should be ", but are rare for"."*

We have revised accordingly.

15. *"Page 5 Line 39: "mode R with measure Na+ input", should be "measured"."*

We have revised accordingly.

16. *"A constant throughout the manuscript is that strong statements are supported primarily by prior work from the authors of this study. It would strengthen the manuscript to either make less strong statements or cite work from other groups to support the claims being made."*

This point is well taken. We have cited more work from other groups. We believe that our statements are justified by our analysis.

I agree with the reviewer and I don't think that the authors have significantly corrected that in revision.

---

## Referee Report (RR2)

Review of "The underappreciated role of nonvolatile cations on aerosol ammonium-sulfate molar ratios"

The authors have added a section considering mixing state, which is a helpful addition for considering that different sources of aerosols are present in the southeast U.S. Despite this addition, there are still concerns about the conclusion NVCs are driving this process and that mixing state is not impacted by a lack of NVCs the smaller OC-sulfate particles. The very strong arguments in the intro and remainder of the paper that organic coatings and viscous aerosols cannot have an effect on pH and partitioning are still too strong and should be weakened or acknowledged as at least potentially playing a role.

*Authors: Including NVCs in the thermodynamic model largely resolves the molar ratio discrepancy, based on our data set, which is representative of the southeast. Only small amounts of NVC are often required, therefore the practice of omitting NVCs from fine mode calculations (which may seem unavoidable for many datasets) induces important biases in molar ratios, which have to be considered in any relevant interpretations (especially on the role of organics).*

Reviewer Comment: The concern with this statement is that, even if including NVCs in the thermodynamic model resolves the discrepancy, if the NVCs are not in the particles this is referring to, then the correct answer is being obtained, but not for the correct reasons. Without evidence that NVCs are present in the SOA/sulfate particles that dominate in the SE US, I am still concerned about the overall finding of this manuscript. For SOAS, if ~5% of sulfate is mixed with sea salt or dust particles and ~95% of sulfate is mixed with SOA particles (simplifying here), but all of the NVCs are present in the salt/dust particles, then the NVCs likely do not play a large role in the ammonium-sulfate molar ratio. Hence, my overall concern that the title "underappreciated role" of NVCs could mislead readers if NVCs do not in fact have much effect on ammonium-sulfate ratios. Recently, we've seen other groups from SOAS point out in a paper under review for ACPD (Bondy et al. 2018) that NVC's are present in < 3% of SOA-dominated particles during SOAS. If >95% of SOA particles contain nearly all the sulfate, but contain no NVCs it is still not clear how the title reflects what is occurring in the aerosols. At a minimum, the authors should discuss that the Na$^+$ concentrations measured are not in the majority of the accumulation more and qualify their results, accordingly.

*Authors: The bias in R (ISORROPIA predicted R with Na+ minus ISORROPIA predicted R without Na+, where R = NH4+/SO42-, mole/mole) is highly correlated with measured Na+ (r2 = 0.93), but not correlated with OA mass or OA mass fraction. Furthermore, the difference in observed R from a ratio of 2 (R observed minus 2) is correlated with NVCs and not correlated with OA mass or OA mass fraction. Both results provide strong evidence for NVCs, and not OA, as the underlying driver of the molar ratio discrepancy.*

Reviewer Comment: When considering mixing state, this argument is not as clear as it might appear. If dust or salts are introduced (Allen et al. 2015, Bondy et al. 2017), Na$^+$ will go up, but so will sulfate from sea salt, though it is likely a low fraction of overall sulfate (that is externally mixed from the most of the sulfate mixed with OC in submicron particles). The equilibration time scale between (as modeled now in the paper) is not well known in the atmosphere, though if the authors can define that it would be useful. Thus, the correlation here may be indicative of shifting concentrations of different populations and not particularly strong evidence that Na$^+$ is leading to changes in R (e.g. correlation equals not causation).

*Authors: Mixing state of NVCs with ammonium, sulfate, and nitrate is a valid concern that we had not addressed in the original study. The assertion that there is absolutely no mixing of Na+ with sulfate is incorrect. Na+, which is largely from sea salt (as noted by the reviewer), will contain some fraction of sulfate since it is well known that there is sulfate in sea water (~25% SO42-/Na+ mass ratio) (DOE, 1994),*

*apart from any secondary non-sea-salt sulfate or subsequent effects from aerosol aging processes. Nevertheless, we have now considered the impact of incomplete mixing of the non-volatile inorganic species (sulfate, sodium and other NVC) on R and pH in a separate section in the main text. The analysis shows that with a small fraction of sulfate internally mixed with Na+-rich particles gives acidity and partitioning levels consistent with that of complete internal mixing. The issue of mixing state of NVCs with SOA does not affect our conclusions as well, especially given that the molar ratio discrepancies (one of the main points in our paper) can be explained by inorganic species without any consideration of organic effects. Therefore, effects from incomplete mixing of particles do not alter our conclusions obtained in the original analysis.*

Reviewer Comment: The argument that mixing does not alter the conclusions of the original analysis is surprising and I am still unclear as to how this can be. Particularly as the authors now cite work showing the pH is predicted to be higher for particles > 1 micron than < 1 micron. The authors are correct to point out that not all sulfate is in SOA particles, but based on filter measurements and AMS measurements at SOA, it would be safe to say the vast majority of sulfate is present in submicron SOA-sulfate particles. The concern is that if no NVCs are present in > 95% of SOA-sulfate particles, it seems highly unlikely that the R is being controlled by NVCs. The authors have addressed the inverse concern (e.g. does having a small amount of sulfate with NVCs change the overall finding), but this is not the major concern. The concern is that NVCs are not present in the very acidic SOA-sulfate particles that are the focus of this paper. If the NVCs in the 2nd externally mixed population are driving acidity in the other population that is a very surprising result, though perhaps I'm misunderstanding the external mixing section.

*The use of data below LOD is justified as the LOD is simply an estimate (i.e. three times of field blanks standard deviation). We note throughout the text when we are using data that is below the LOD. Below we show that the use of inferred Na+ is reasonable as an upper limit of NVCs through comparisons with measurements by other instruments at the SOAS study. Here we show that when measured NVC > measurement LOD (i.e., reliable NVC concentrations, admittedly a small fraction of the study period) there is good agreement between measured and ISORROPIA-predicted R (updated Fig. 1f and Fig. 3, and associated text; no difference at a significance level of $\alpha = 0.05$). Issues arise (which is not surprising) when the NVC concentrations are below or near LOD and they have to be estimated. Most defensible conclusions can be made for periods with the most reliable data (NVCs above LODs). Assessments and critiques that focus solely on only periods of the least reliable data sets (NVCs below LODs) are weak. However, even periods with NVCs below LODs, often provide consistent results. For example:*

*- For SOAS, when measured Na+ was below LOD (0.07 µg m-3), a Na+ of 0.1-0.3 µg m-3 was needed to bring measured and predicted R into agreement, which is near the detection limit and so difficult to measure. Despite this, the trends are the same and the predicted R is generally smaller than the observed, and so including the estimated NVCs in this case overcompensates.*
*- For WINTER data, inferred NVC results in agreement in R prediction with the observation, but inferred Na+ is smaller than observed PM1 Na+, 0.15 vs. 0.23 µg m-3, stated in the caption of Figure 4. Again, the results are uncertain since measurements of NVC were not measured online in that study.*

*The reviewer criticizes the inferred Na+ from ion charge balance to be worrisome and much higher than measured Na+. The inferred Na+ represents an upper limit of NVCs, including K+, Mg2+, and Ca2+ (if soluble), that could be present in the aqueous aerosols based on charge neutrality of the solution. Therefore, it is supposed to be higher than a single measurement of NVC, such as Na+. We have results from the WINTER study showing the inferred Na+ is comparable to the offline measurement. Also, in Figure I (b), we show a comparison between inferred Na+ from PILS-IC data and MARGA data and measured Na+-*

*equivalent NVCs (K+, Mg2+, and Ca2+ are represented by Na+) from MARGA for the SOAS study (Allen et al., 2015). Before June 18, the three Na+ levels are similar, indicating the inferred Na+ level is reasonable. After June 18, the two inferred Na+ levels are higher than MARGA total measured NVCs, but the differences are within uncertainties (propagated from PILS-IC ammonium, sulfate, nitrate, and chloride measurements).*

Review Comment: I am still uncomfortable with the extensive use of periods of data below LOD and inferred ion balance. LOD is admittedly arbitrarily defined, but it is an analytical standard, since as data approaches background noise, measurements are less reliable. I still have concerns that a higher concentration of "inferred" $Na^+$ is needed to make the model match the measurements, perhaps that means that $Na^+$ is not driving the R here? It is appreciated that the authors clearly define the time periods.

*Authors: We don't agree. Our results from assuming an external mixture are consistent with bulk analysis (internal mixture), see added section on internal vs. external mixture. It is true that we have no data on organic film thickness, but if the organic film is sufficient to impede NH3 uptake it has to comprise a reasonable fraction of the overall OA (a timescale analysis easily shows this), and concurrently impede equilibration of water and nitrate with the gas phase. We show that neither OA mass or mass fraction is related to observed bias in R. If the argument is that an organic film has a widespread effect throughout the southeast, it can't be argued that it is some unique property of the OA (i.e. unique composition, unique viscosity, selectivity with respect to NH3, etc.) unless if a specific mechanism can be proposed to support it.*

Reviewer Comment: More evidence continues to come out about diffusion limitations of viscous particles (see citations within (Reid et al., 2018)) and the role of coatings inhibiting partitioning that are not simply related to organic mass fraction (Zhang et al., 2018). Zhang et al. from the Surratt Group this year for example showed that even a 10 nm coating of oxidized monoterpene SOA, could decrease reactive uptake coefficients by a factor of 4 and that would decease isoprene SOA formation via IEPOX by 15-20% in conditions representative of the SE US. Clearly, IEPOX is a much larger molecule than ammonia, but at the ionic strengths listed below it seems very plausible that a distinct organic phase that could inhibit water or ammonia uptake could be present. I think given the caveats of the author's own assumptions (e.g. inferring sodium concentrations), that the statement ruling out a role for organic coatings and glassy organics should be weakened.

*Author Comment: We clearly have a different understanding of "direct" pH measurement than the reviewer. We cited the method of Rindelaub et al. (2016) as an indirect (but clearly particle-level) measurement of aerosol pH because it doesn't quantify the hydronium ion aqueous phase activity directly. Instead, it infers the H+ concentration by HSO4-/SO42- ratio, equilibrium constant, and activity coefficients. Deliquesced ambient fine particles are very concentrated liquids. The average ionic strength was 29 mol L-1 for this study. Therefore, non-ideality cannot be ignored and avoided by the method of Rindelaub et al. (2016) as the reviewer claims. Furthermore, the activity coefficients of HSO4- and SO42- cannot be determined simply by HSO4- and SO42- concentrations because the non-ideal effects are caused by interactions of all water-soluble ions in aqueous aerosols, such as NH4+ and NO3-. In terms of determining activity coefficients, a thermodynamic model with an input of all measured inorganic ions should be more accurate than one with only a fraction of the ions input.*

Reviewer comment: By the authors' standard of a direct measurement, would they consider a pH probe a direct measurement of $H^+$ activity/pH? If not, then it is likely the direct measurements of $H^+$ or pH are essentially impossible based on the standard the authors establish. pH probes and other measurements

of pH have a long established history, even if there are a few assumptions needed since the activity of each ion is solution is not known precisely in many systems (including aerosols). Direct in this discussion was intended to refer to measuring both the acid and conjugate base concentrations directly, of which the Rindelaub measurement is the first of its kind. Arguments over the definition of "direct" aside, the authors misunderstand the Rindelaub paper, as it does not assume an ideal solution. Activity calculations are conducted for all species in solution (since it is a model system). The followup work in Craig et al., includes an entire figure on $H^+$ activity coefficient versus ionic strength for inorganic, organic, and mixed system. Activity coefficients are calculated for all components in both Craig et al. and Rindelaub et al., so the non-ideality is not being ignored, as the authors claim.

The authors note that the average ionic strength for this study is 29 mol/liters, which is quite high and brings about three questions:

1) At that concentration are there concerns about the limited water present and the impact on the thermodynamic calculations? How are the authors handling the fact that most thermodynamic models struggle with non-dilute solutions? This may be a lack of knowledge of ISORROPIA, but Debye-Hückel breaks down above 0.1 M ionic strength I believe, does it not? How are the authors avoiding this, if so, it would be helpful to mention this high ionic strength and include a brief discussion.

2) How would this change if ISORROPIA accounted for organic components, a large mass fraction of the aerosols in question? My hunch is that it would lower the ionic strength, which would impact the figure shown below of shifting $K_a$ (moving less to left), so it would be helpful if the authors could address this.

3) At such high ionic strengths and lower water activities, it seems likely that many of the inorganics would end up salting out of the organic component, leading to core-shell liquid liquid phase separations. Despite the authors stating that the organics will not have an effect, it would seem this likely supports that these organic phases, possibly quite viscous, could impact partitioning and thus R. If the authors could address this it would be helpful.

Considering the challenging processes being studied by this paper and other papers focused on this topic, the authors would be well-served to acknowledge the possibility of other factors beyond their proposed explanation, which relies on its own assumptions.

*Author Comment: The simple calculation as referred by the reviewer is likely based on ideal solutions, where all activity coefficients are treated as one. In this case, HSO4- is the dominant form at pH of 1 (shown as dash lines in Figure II). After taking into account of the activity coefficients predicted by ISORROPIA, the curves move left by 2 units (comparing solid vs. dash lines in Figure b). As a result, SO42- is the dominant form at pH of 1. Therefore, our statement is consistent with the basic acid dissociation rules. Instead of citing 14% as HSO4- from WINTER study, the Figure II has been added to supplemental material to explain how activity coefficients affect the relative fractions of SO42- and HSO4-.*

Review comment: The authors make a fair point, presuming organic species do not lower the ionic strength significantly. How much would this then shift the $pK_a$ of the bisulfate/sulfuric acid system, the authors noted in comment 13? Would that likely mean that it would be much lower than could be observed in ambient aerosols? Thus, sulfate and bisulfate are the only forms of sulfate that need to be addressed for aerosols? Some clarification would be helpful.

Reid, J. P., Bertram, A. K., Topping, D. O., Laskin, A., Martin, S. T., Petters, M. D., Pope, F. D., and Rovelli, G.: The viscosity of atmospherically relevant organic particles, Nat. Commun., 9, 956, 2018.

Zhang, Y., Chen, Y., Lambe, A. T., Olson, N. E., Lei, Z., Craig, R. L., Zhang, Z., Gold, A., Onasch, T. B., Jayne, J. T., Worsnop, D. R., Gaston, C. J., Thornton, J. A., Vizuete, W., Ault, A. P., and Surratt, J. D.: Effect of Aerosol-Phase State on Secondary Organic Aerosol Formation from the Reactive Uptake of Isoprene-Derived Epoxydiols (IEPOX), Env. Sci. Tech. Lett., doi: 10.1021/acs.estlett.8b00044, 2018. 2018.

---

## Referee Report (RR3)

The authors' have made a number of revisions, which have improved the paper, but have not fully addressed a number of issues raised by multiple reviewers. Due to the lengthy (and quite combative) nature of the response, only the most concerning remaining issues with the manuscript are addressed.

- **Limit of Detection**: With respect to using measurements below the limit of detection (LOD) of the system. The authors' have made the statement that "The analysis is consistent for Na+ concentrations over the full Na+ measurement range, including below our assigned Na+ LOD." If a measurement is below its LOD it is not considered reliable, since it is within a signal-to-noise ratio of 3 to 1. When the signal is near the intensity of the noise a number of factors can lead to issues. The authors' claim that researchers define LOD in different ways, but for a standard analytical measurement, such as chromatography, there is an accepted definition that is broadly used and which should be here. The fact that the data is consistent with data above the LOD, is still not a good reason for using this data. Frankly data below the limit of quantification (LOQ) (10:1) shouldn't be used, but signal should at least be > 3:1 to be used. The authors' do not seem willing to only use properly quality controlled data, which should be the standard in a high quality journal like ACP.

- **Inferred Na$^+$**: The other major issue is the use of "Inferred Na$^+$", which I am still uncomfortable with. First, "Inferred Na$^+$" should be renamed since it accounts for all NVCs, not just sodium. Perhaps "Inferred NVCs"? Secondly, as one of the reviewers noted, inferred Na$^+$ is a bit of a circular argument and somewhat misleading. In prior work, the authors' have argued that proxy methods, such as molar ratio and ion balance have significant shortcomings, but to my understanding "inferred Na$^+$" uses the same basic concept since inferred Na$^+$ is used to help explain the ammonium-to-sulfate ratios, even when the actual Na$^+$ data don't match that. Maybe this could be reframed with a different name to help the issue. However, introducing a fudge factor and implying it has a real meaning (e.g. Na$^+$ concentration) to account for the fact that the measurements of Na$^+$ (which have their own issues) don't match the inferred value necessary to have NVCs explain the $R$ ratio, is not strong support for the authors' hypothesis and argument that NVCs explain the ammonium-sulfate ratio.

- **Exclusion of Organic Films, Glassy Aerosols, or other Possibilities:** The abstract and paper overall still very much imply that the authors' hypothesis about NVCs is the only reasonable explanation for overprediction of R values. However, given the issues with the Na$^+$ data (below LOD) and the need to use inferred Na$^+$ values higher than what is measured to make the hypothesis work, the strength of the authors' arguments against organic properties/films or other explanations does not appear sufficiently justified and should be weakened (as noted in previously).

---

## Author Response (AR2)

Below, we have addressed the reviewer comments that raise an issue. The reviewer comments are presented in quotes and italics, followed by our responses in plain text.

**Responses to Reviewer 3**

1. *"However, I agree with reviewer 2 that the data used by the authors to argue that all is well with thermodynamics are either unusual (high Na+) or misleading (inferred high Na+). The authors don't provide a satisfactory response in the revised text."*

   We *do not* argue that "all is well with thermodynamics", but *what we do* argue is that for major inorganic ions, like $NH_4^+$, the large discrepancy in predicted versus measured partitioning (expressed by the molar ratio) can be explained by careful attention to the presence of non-volatile cations (NVC), and a demonstrated large $NH_4^+$ sampling bias in the CSN data. Together with the observed strong correlation of molar ratio discrepancy with NVC and a lack of correlation with organic mass and mass fraction indicates that organics are unlikely associated with any inhibition in $NH_3$ uptake. The demonstrated equilibration of aerosol species (like water and nitrates) further excludes the presence of glassy aerosol or organic films that would strongly inhibit mass transfer.

   Furthermore:
   1. The $Na^+$ data used in our analysis is of good quality and not unusually high for the region. For the SOAS study, PILS-IC observed $Na^+$ was on average 0.07 µg m$^{-3}$ and in good agreement with MARGA $Na^+$ (see Fig. S2 in the supplement, the last submitted version, hereafter referred as V1). Average $Na^+$ over a full year at various sites in the southeast are in the range of 0.05 to 0.1 µg m$^{-3}$ (see new Supp material Table S1 and S2, which are also shown below).
   2. The analysis is consistent for $Na^+$ concentrations over the full $Na^+$ measurement range, including below our assigned $Na^+$ LOD. As an example, Fig. 4a shows that discrepancies in $R$ vs. $Na^+$ apply over the complete range of the observed $Na^+$ data. Nothing changes through the LOD transition (vertical line in Fig. 4a) to lower concentrations. The comparison between ISORROPIA predicted and measured ammonium partitioning in Fig. 2a is also consistent for all data points, which includes data below the $Na^+$ LOD. We do agree that the time series plot of Fig. 1 tends to emphasize the higher $Na^+$ events, but this is only one component of the paper.
   3. The ion balance does not produce misleading $Na^+$ data, it is, however, highly uncertain due to subtracting large numbers of small difference (i.e., $SO_4^{2-}$ and $NH_4^+$). We have discussed this in the text

(Line 27 Page 5 V1). The issue of not including $H^+$ in the ion balance, based on a full thermodynamic analysis, is discussed below. The statement that the inferred $Na^+$ is always consistently high is also incorrect; in SOAS this is true, but in the WINTER study it is consistently lower (Line 15 Page 8 V1).

2. *"I agree with reviewer 1 that the authors should not ignore the low values of R in the CSN data, and if they think that these values are biased they should say so and why."*

In response to this, we have added a paragraph and table showing there is a substantial bias in CSN $R$. (Note, this was recognized by Silvern et al. (2017), but was used anyway to argue that a low $R$ indicted problem with thermodynamic predictions, even if the bias by itself can account for the molar ratio discrepancy). The following is added on Page 4. "In addition to the SOAS and WINTER data sets, the Southeastern Aerosol Research and Characterization (SEARCH) CTR sampling site (the same as SOAS) historical data from year 1998 to 2013 is re-analyzed to show that thermodynamic model can reproduce the observed decreasing trend of $R_{SO4}$ when NVCs are considered. Molar ratios determined from the Chemical Speciation Network (CSN), which were utilized and discussed by Silvern et al. (2017) and Pye et al. (2018), are not used in this work because of a significant low bias when compared to the SEARCH and SOAS data (see Table S1 and S2 in the supplement)". For example, during the 12-day SOAS study period investigated here (11-23 June 2013), the online measurements in SOAS (CTR) reported $R$ values of $1.70 \pm 0.23$ for PILS and $1.78 \pm 0.18$ for MARGA. SEARCH filter-based measurements of $R$ were similar at $1.57 \pm 0.11$ at CTR and $1.64 \pm 0.14$ at Birmingham (BHM). In contrast, the two closest CSN sampling sites near CTR reported much lower $R$ of $0.70 \pm 0.36$ at BHM and $0.75 \pm 0.42$ at Montgomery (MTG). For the year of 2013, the CSN data at BHM showed a similar low bias compared to the SEARCH; SEARCH $R$ of $2.05 \pm 0.23$ was significantly higher than CSN $R$ of $1.26 \pm 0.59$. The discrepancy is likely due to the loss of semivolatile $NH_4^+$ collected on the CSN nylon filters, as noted by Silvern et al. (2017). Note that this bias is of the order of the discrepancy of $R$ that is postulated to arise from organics. This fact alone should be sufficient for any reviewer to recognize as a critical issue in the Silvern et al. (2017) analysis. The two tables below are added to the supplemental material.

**Table S1.** Comparisons of observed PM$_{2.5}$ ions and molar ratio between SOAS, SEARCH, and CSN ground sampling sites for the 11-23 June 2013 period (Fig. 1 in the main text). Since CSN (Chemical Speciation Network) doesn't have a site at CTR to be directly compared to SOAS and SEARCH, the two closest sites at

Birmingham (BHM) and Montgomery (MTG) are used. The most direct comparison is between Birmingham, SEARCH and CSN data. Means are shown with standard deviations.

| Network | SOAS | SOAS | SEARCH | SEARCH | CSN | CSN |
|---|---|---|---|---|---|---|
| Site location | CTR | CTR | CTR | BHM | BHM | MTG |
| Site coordinate | 32.90289, −87.24968 | 32.90289, −87.24968 | 32.90289, −87.24968 | 33.55302, −86.81485 | 33.49972, −86.92417 | 32.41281, −86.26339 |
| Method | PILS-IC | MARGA(-IC) | Teflon filter(-IC) | Teflon filter(-IC) | Nylon filter(-IC) | Nylon filter(-IC) |
| $NH_4^+$, µg m$^{-3}$ | $0.64 \pm 0.22$ | $0.79 \pm 0.22$ | $0.63 \pm 0.13^*$ | $0.69 \pm 0.20^*$ | $0.24 \pm 0.14$ | $0.25 \pm 0.19$ |
| $SO_4^{2-}$, µg m$^{-3}$ | $2.06 \pm 0.68$ | $2.38 \pm 0.66$ | $2.16 \pm 0.44$ | $2.23 \pm 0.51$ | $1.69 \pm 0.40$ | $1.46 \pm 0.84$ |
| $Na^+$, µg m$^{-3}$ | $0.07 \pm 0.09$ | $0.09 \pm 0.10$ | $0.06 \pm 0.04$ | $0.05 \pm 0.04$ | $0.13 \pm 0.06$ | $0.10 \pm 0.04$ |
| *R* | **1.70 ± 0.23** | **1.78 ± 0.18** | **1.57 ± 0.11** | **1.64 ± 0.14** | **0.70 ± 0.36** | **0.75 ± 0.42** |
| Data points | 229 | 229 | 13 | 4 | 5 | 3 |
| Notes on data | Hourly data | Hourly data | Daily data; every day | Daily data; every three days (6/12-6/21) | Daily data; every three days (6/9-6/24) | Daily data; every six days (6/9-6/21) |
| Reference | (Guo et al., 2015) | (Allen et al., 2015) | (Edgerton et al., 2005; Hidy et al., 2014) | | (Solomon et al., 2014) | |

$^*$ SEARCH $NH_4^+$ was measured by automated colorimetry.

**Table S2.** Comparisons of observed PM$_{2.5}$ ions and molar ratio between SEARCH and CSN ground sampling sites for the year 2013.

| Network | SEARCH | SEARCH | CSN | CSN |
|---|---|---|---|---|
| Site location | CTR | BHM | BHM | MTG |
| Site coordinate | 32.90289, −87.24968 | 33.55302, −86.81485 | 33.49972, −86.92417 | 32.41281, −86.26339 |
| Method | Teflon filter(-IC) | Teflon filter(-IC) | Nylon filter(-IC) | Nylon filter(-IC) |
| $NH_4^+$, µg m$^{-3}$ | $0.55 \pm 0.28^*$ | $0.72 \pm 0.31^*$ | $0.48 \pm 0.34$ | $0.41 \pm 0.29$ |
| $SO_4^{2-}$, µg m$^{-3}$ | $1.71 \pm 0.89$ | $1.96 \pm 0.90$ | $1.91 \pm 0.99$ | $1.65 \pm 0.89$ |
| $Na^+$, µg m$^{-3}$ | $0.05 \pm 0.05$ | $0.05 \pm 0.05$ | $0.13 \pm 0.30$ | $0.10 \pm 0.08$ |
| *R* | **1.75 ± 0.28** | **2.05 ± 0.05** | **1.26 ± 0.59** | **1.24 ± 0.59** |
| Data points | 154 | 111 | 93 | 61 |
| Notes on data | Daily data; every three days | Daily data; every three days | Daily data; every three days | Daily data; every six days |
| Reference | (Edgerton et al., 2005; Hidy et al., 2014) | | (Solomon et al., 2014) | |

$^*$ SEARCH $NH_4^+$ was measured by automated colorimetry.

3.  *"I also agree with reviewer 2 that the propensity of the authors to cite their own previous work as right and to misleadingly characterize the work of others as wrong borders on the embarrassing."*

The point is well taken, and we have attempted to reduce our self-citations, where possible. As the manuscript stands now, there are 21 self-citations to papers involving Guo, Weber, and Nenes, compared to 151 citations to the work of other groups. If the reviewer was referring to the discussion of Rindelaub et al. (2016) in the introduction, it has been removed in the last submitted version as it was not highly relevant to this paper. This should resolve the issue reviewer 2 had pertaining to that discussion (see response to reviewer 2 below). We have also removed from the abstract the statement that the organic film proposed by Silvern et al. (2017) selectively inhibiting $NH_3$, but not water vapor and $HNO_3$.

There are three issues the reviewer may wish to also consider on this matter:

1)  As stated in our response to reviewer 2, who brought this up, there is very little recent work on aerosol pH, other than our own, and so the number of available papers to cite is limited.

2)  This paper is in part a rebuttal of the Silvern et al. (2017) paper that proposes a different explanation for trends in *R* over the last decade, in contrast to what we had proposed (Weber et al., 2016). Thus, it is natural to cite our work on this topic. We do cite other publications in the introduction which show that the organic film hypothesis that predicts lack of ammonia gas-particle equilibrium is in contrast to established literature showing that $NH_3$, water vapor, and $HNO_3$ equilibrate with organic-rich aerosols (Ansari and Pandis, 2000; Moya et al., 2001; Morino et al., 2006; Fountoukis et al., 2009; Guo et al., 2015; Guo et al., 2016; Guo et al., 2017; Liu et al., 2017; Paulot et al., 2017) (V1). We have also added more citations from other research groups in this round of reviews. We note that to question hypotheses and then use well documented arguments for a revised picture of reality is exactly how science should progress, and is something that no one should be embarrassed of!

3)  It might be worth noting that Reviewer 2, who made this self-citation comment, has requested throughout their reviews that we cite various other papers (there are 6 of them, which are listed at the end of this response). We have attempted to include all of these papers in our manuscript, where appropriate. It is worthy to note, that 5 of these 6 requested manuscripts also have a common author, giving the appearance that the reviewer's main concern is that we cite their papers. The irony of this is not lost on us.

4. *"I don't think that the authors have satisfactorily addressed in the text the much lower R values found in the CSN data, which cannot be explained by the NVC hypothesis."*

We have addressed this question in bullet 2 above. The CSN data is biased so not considered in our analysis.

5. *"But isn't standard protocol to remove that sea-salt sulfate and just report non-sea-salt sulfate? Just checking, I know it's standard protocol in research data sets."*

The discussion was referring to particle mixing state and measurements of mixing state by single particle analysis, which of course cannot distinguish between sulfate from different sources. And it is not correct to do so either, because the thermodynamics is affected by all forms of sulfate, regardless of origin.

6. *"But then of course R would be less than 2. These are unusual conditions when NVCs are high."*

We have answered this question in bullet 1 above. Again, as an example, Fig. 4a shows, there is no discontinuity in the $R$ discrepancy for $Na^+$ above and below the LOD. As we show, the NVC levels we report are not unusual.

7. *"I agree with the reviewer that inferred $Na^+$ is highly problematic and I don't see that the authors have addressed that concern in the revised text. A major problem is that the charge balance equation used to infer $Na^+$ doesn't include $H^+$ and thus forces $H^+$ to be low so R to be high resulting in a circular argument. The authors justify this by arguing that $H^+$ is very low compared to other cations but that is based on their thermodynamic calculation for $H^+$ assumed to be correct (note that in their example $[NH_4^+] >> [H^+]$, effectively meaning R close to 2), so it is self-fulfilling."*

The numbers prove that our statement holds and is not a circular argument. To show this, we have added a comparison (see plots below) between $Na^+$ predicted with $H^+$ in the ion balance and without $H^+$ in the ion balance, where $H^+$ is determined iteratively with the full thermodynamic model. The following has been added to the main text: "For the three data sets used in this study, the difference in $Na^+$ predicted from an ion balance without considering $H^+$ compared to including $H^+$ is less than 1% for SOAS and SEARCH CTR, and 6% for the WINTER study, (see Fig. S3 in the supplement). In the following, we have not included $H^+$ in the ion balance."

We also point out that it is much more reliable to get aerosol $H^+$ through a full thermodynamic analysis and have added a sentence in the summary, "Note that the ion charge balance on its own generally cannot be used to infer $H^+$ since the $H^+$ concentrations are generally very low, even at the low pH of the southeastern US aerosols, and the dissociation states of acids must be known (e.g., proportions of $HSO_4^-$ and $SO_4^{2-}$), which requires a full thermodynamic analysis."

[Figure]

**Fig. S3.** Comparisons of ion charge balance inferred $Na^+$ including $H^+$ ($2SO_4^{2-} + NO_3^- + Cl^- - NH_4^+ - H^+$; y-axis) versus excluding $H^+$ ($2SO_4^{2-} + NO_3^- + Cl^- - NH_4^+$; x-axis) for three data sets used in the paper, (a) SOAS (Fig. 1), (b) WINTER (Fig. 5), and (c) SEARCH CTR (Fig. 6). The $H^+$ concentration was determined using ISORROPIA in an iterative approach. $Na^+$ is predicted from the ion balance is included with all other gas/particle species in the model, resulting in a predicted $H^+$. This $H^+$ is included in the ion balance to predict a new Na+, which is then used in a new model iteration. The procedure is repeated until the $Na^+$ concentration converges. The number of iterations for conversion are 1 for (a), 5 for (b), and 0 for (c), respectively, until inferred $Na^+$ converges. Orthogonal distance regression (ODR) fits are shown and uncertainties in the fits are one SD.

8.   *"So a problem with this paper right now is that it pushes its argument either by presenting unusual situations where $Na^+$ is above LOD or by forcing the argument to be correct through the upper limit of inferred $Na^+$."*

We have addressed these points above in Bullet 1. In a short summary, the above LOD $Na^+$ in the SOAS study is not unusually high compared to other co-collected data. Inferred $Na^+$ level is by nature an upper limit since it represents all the NVCs. A careful review of the data shows that the inferred $Na^+$ is no unrealistic. It is higher than SOAS measured $Na^+$ but lower than WINTER measured $Na^+$.

9. "*Page 4 Lines 33 ‑ 36 and Page 5 lines 1 ‑ 7: Of the three different $Na^+$ levels tested, option 1 infers that any lack of charge balance can be attributed to $Na^+$. On line 35 ‑ 36, the authors then note that inferring the amount of $Na^+$ leads to a value more than 4 times higher than the measured value. There are a number of other possibilities that could explain these strongly different results and inferring shouldn't work as there is almost no $Na^+$ mixed with SOA particles. This is in fact supported by the fact that the reported values for $Na^+$ are below the limit of detection of the measurement (0.06 value when LOD is 0.07).*"

This comment was made by reviewer 2, which we had addressed, and addressed again in Bullet 1 above. This reviewer seems satisfied with our added section addressing the mixing state issue. We further discuss it below in response to reviewer 2, who continues to raise the issue.

10. "*I agree with the reviewer and I don't think that the authors have addressed the issue satisfactorily. Conditions where $Na^+ > LOD$ are unusual and conducive to their argument. Conditions where $Na^+ < LOD$ should indeed not be used and the inferred Na+ seems misleading.*"

We once more emphasize that the $Na^+$ data above LOD is not unusual. To clarify this, we have added one sentence justifying why we use $Na^+$ below the LOD on Page 6, "In the following, $Na^+$ data below the LOD is used in the analysis to increase the size of the data set, given that there is no obvious discontinuity in the results for data above and below the $Na^+$ LOD (see Fig. 4) and that $Na^+$ below the PILS LOD still roughly agrees with MARGA measurements of $Na^+$, which has a lower LOD (see Fig. S2a in the supplement)."

11. "*I agree with the reviewer and think that the authors make too much hay off this organic film hypothesis. It seems to have been suggested by Silvern et al. as a speculative explanation for the low R in the CSN and AMS data, but here the paper misleadingly characterizes Silvern et al. as being all about the organic film hypothesis and misses their main point which was to draw attention to the low R in the CSN data (a problem ignored by this paper).*"

We apologize if we gave the wrong impression. We have deemphasized the references to organic films (although still point out to why they are unlikely), and broadened the discussion at the level of "organic aerosol", because the main point is that the discrepancy between predicted and observed $R$ is not correlated with organic mass or fraction.

We disagree that the main point of the Silvern et al. (2017) paper was to point to the CSN issue! The paper's title is "Inconsistency of ammonium–sulfate aerosol ratios with thermodynamic models in the eastern US: a possible role of organic aerosol". This paper discusses more than molar ratios in the southeast, it claims the low molar ratios are inconsistent with thermodynamic models and that an organic film that impedes $NH_3$ uptake could be an explanation. These assertions, if true, have large ramifications. Our point is that if one includes $Na^+$ data in the thermodynamic model (it is not included when using AMS data), and use high quality data (i.e., not CSN with a known $NH_4^+$ low bias, but SOAS study online data), there is no discrepancy in $R$ and so no need for organic films or organics overall. We show this is true with an additional data sets (WINTER study and SEARCH CTR study). Finally, we have noted the molar ratio discrepancy in an earlier paper (Weber et al., 2016), where instead of focusing on molar ratios ($R$), we focused on pH. We show that there is no inconsistency with a trend of lower sulfate relative to ammonia, a constant pH and small drop in molar ratios. In fact, we showed the opposite of Silvern et al., that thermodynamic models actually explain the observations.

12. "*The authors again seem to make misleading claims to dismiss previous literature – here that they used R as an acidity proxy. They did not.*"

There was a misunderstanding in the reference to the Rindelaub et al. (2016) paper in the first version of the manuscript. We intended to say that the lack of direct particle pH measurements contributes to the use of pH proxies. This issue was addressed in the first set of revisions; we have removed the sentence and the reference to Rindelaub and so the issue is resolved.

13. "*Again, the authors misleadingly describe Silvern et al. as claiming that the organic film limitation would apply to NH₃ but not to H₂O and HNO₃. They said nothing of the sort and instead pointed out that the organic film limitation hypothesis was problematic precisely because it would have to also apply to H₂O and HNO₃.*"

The reviewer is not correct here. Silvern et al. (2017) notes that "A mass transfer retardation of thermodynamic equilibrium may also have broader implications for the partitioning of semivolatile species and for hygroscopicity." We have added citations to papers showing that there is evidence for equilibrium of $NH_3$, $H_2O$, and $HNO_3$ in contrast to the organic film hypothesis. We have deemphasized the statements that the organic film proposed by Silvern et al. (2017) only selectively inhibits $NH_3$ in an attempt to be less confrontational (although it remains a critical point).

14. "*I agree with the reviewer that the authors' propensity to cite their own work and to dismiss others' borders on the embarrassing. I don't think that they fixed this in revision. The message one gets from the paper is that the authors are the only ones who understand particle thermodynamics and acidity, and everyone else doesn't know what they're doing; that doesn't come across very well.*"

We have addressed the claimed self-citation issue above (as the manuscript stands now, there are 21 self-citations to papers involving Guo, Weber, and Nenes, compared to 151 citations to the work of other groups). We are certainly not the only group that understands thermodynamics, but we do think it is reasonable to point out in the published literature any flaws, statements that are in disagreement with quantitative data, or conclusions based on data with a demonstrated bias. The reviewer's final comment is a subjective and personal critique, which we will not respond to.

15. *Reviewer 2: "A constant throughout the manuscript is that strong statements are supported primarily by prior work from the authors of this study. It would strengthen the manuscript to either make less strong statements or cite work from other groups to support the claims being made."*
*Authors' response: "This point is well taken. We have cited more work from other groups. We believe that our statements are justified by our analysis."*
*"I agree with the reviewer and I don't think that the authors have significantly corrected that in revision."*

We believe that we have presented strong arguments that are *supported by data* and by *thermodynamics*. An attempt to "tone down" the paper has been made as requested, but prefer not to make qualifying statements throughout the paper as it reduces the force of our arguments, which we believe are correct. How one choses to write is a question of *style*, and our preference is a direct straightforward approach for the sake of clarity.

At the end of the paper, we have added a text on limitations with this analysis: "Further assessments on possible effects of organic effects on semi-volatile partitioning of inorganic species however should be carried out, especially for regions that are chemically different from the eastern US conditions evaluated in this study."

Summary of papers reviewer 2 has suggested to cite:

1.  Allen, H. M., Draper, D. C., Ayres, B. R., Ault, A., Bondy, A., Takahama, S., Modini, R. L., Baumann, K., Edgerton, E., Knote, C., Laskin, A., Wang, B., and Fry, J. L.: Influence of crustal dust and sea spray supermicron particle concentrations and acidity on inorganic NO3- aerosol during the 2013 Southern Oxidant and Aerosol Study, Atmos. Chem. Phys., 15, 10669-10685, doi: 10.5194/acp-15-10669-2015, 2015.
2.  Bondy, A. L., Bonanno, D., Moffet, R. C., Wang, B., Laskin, A., and Ault, A. P.: Diverse Chemical Mixing States of Aerosol Particles in the Southeastern United States, Atmos. Chem. Phys. Disc., 1-37, doi: 10.5194/acp-2017-1222, 2018.
3.  Craig, R. L., Nandy, L., Axson, J. L., Dutcher, C. S., and Ault, A. P.: Spectroscopic Determination of Aerosol pH from Acid-Base Equilibria in Inorganic, Organic, and Mixed Systems, J Phys Chem A, 121, 5690-5699, doi: 10.1021/acs.jpca.7b05261, 2017.
4.  Reid, J. P., Bertram, A. K., Topping, D. O., Laskin, A., Martin, S. T., Petters, M. D., Pope, F. D., and Rovelli, G.: The viscosity of atmospherically relevant organic particles, Nat Commun, 9, 956, doi: 10.1038/s41467-018-03027-z, 2018.
5.  Rindelaub, J. D., Craig, R. L., Nandy, L., Bondy, A. L., Dutcher, C. S., Shepson, P. B., and Ault, A. P.: Direct Measurement of pH in Individual Particles via Raman Microspectroscopy and Variation in Acidity with Relative Humidity, J Phys Chem A, 120, 911-917, doi: 10.1021/acs.jpca.5b12699, 2016.
6.  Zhang, Y., Chen, Y., Lambe, A. T., Olson, N. E., Lei, Z., Craig, R. L., Zhang, Z., Gold, A., Onasch, T. B., Jayne, J. T., Worsnop, D. R., Gaston, C. J., Thornton, J. A., Vizuete, W., Ault, A. P., and Surratt, J. D.: Effect of the Aerosol-Phase State on Secondary Organic Aerosol Formation from the Reactive Uptake of Isoprene-Derived Epoxydiols (IEPOX), Environ. Sci. Technol. Lett., 5, 167-174, doi: 10.1021/acs.estlett.8b00044, 2018.

**Responses to Reviewer 2**

1.    *"The authors have added a section considering mixing state, which is a helpful addition for considering that different sources of aerosols are present in the southeast U.S. Despite this addition, there are still concerns about the conclusion NVCs are driving this process and that mixing state is not impacted by a lack of NVCs the smaller OC‑sulfate particles. The very strong arguments in the intro and remainder of the paper that organic coatings and viscous aerosols cannot have an effect on pH and partitioning are still too strong and should be weakened or acknowledged as at least potentially playing a role."*

We address the concern of mixing state below. We have attempted to tone down our statements in the paper and we claim the effects of organics are minor or negligible based on our analysis, but could be more important in other regions not considered in this study. We have also added a few lines on the limitations of this study at the end of the paper.

2.    *"The concern with this statement is that, even if including NVCs in the thermodynamic model resolves the discrepancy, if the NVCs are not in the particles this is referring to, then the correct answer is being obtained, but not for the correct reasons. Without evidence that NVCs are present in the SOA/sulfate particles that dominate in the SE US, I am still concerned about the overall finding of this manuscript. For SOAS, if ~5% of sulfate is mixed with sea salt or dust particles and ~95% of sulfate is mixed with SOA particles (simplifying here), but all of the NVCs are present in the salt/dust particles, then the NVCs likely do not play a large role in the ammonium‑sulfate molar ratio. Hence, my overall concern that the title "underappreciated role" of NVCs could mislead readers if NVCs do not in fact have much effect on ammonium‑sulfate ratios. Recently, we've seen other groups from SOAS point out in a paper under review for ACPD (Bondy et al. 2018) that NVC's are present in < 3% of SOA‑dominated particles during SOAS. If >95% of SOA particles contain nearly all the sulfate, but contain no NVCs it is still not clear how the title reflects what is occurring in the aerosols. At a minimum, the authors should discuss that the Na+ concentrations measured are not in the majority of the accumulation more and qualify their results, accordingly."*

In the added section, we show that for the data in which $Na^+$ is above LOD, on average 18% $SO_4^{2-}$ by mass is needed in the $PM_{1-2.5}$ size range to obtain the same molar ratio as a complete internal mixture. This is an extreme case. For periods when $Na^+$ concentrations are lower, the $Na^+$ mass fraction required is lower. For example, as stated in the paper, 5% by mass of the sulfate needs to be mixed with $Na^+$ when the $Na^+$ is at the study mean concentration of 0.07 µg m$^{-3}$. Bondy et al. (2018) reports that NVCs are in less than 5% of the SOA/sulfate mixed particles, by particle number (the instrument does not provide a quantitative mass

measurement). Our analysis is based on mass, so the stated percentages are not directly comparable. This is important because there is considerable uncertainty in going from a number mixing percent to a mass mixing percent (see Bondy et al. (2018)). One factor is that the average sulfate mixing is of <5% is for all sizes, and sulfate is mainly associated with $PM_1$ and $Na^+$ in $PM_{2.5}$, (as we assume in the mixing model), thus the mass fraction in the larger sizes where $Na^+$ is, will likely be larger (mass scales with $D_p^3$). Furthermore, Bondy et al. (2018) reports mixing for the complete SOAS study, we use only the first half of the study data when we measured $PM_{2.5}$ ionic species, a period when $Na^+$ was larger, thus for the whole study even less sulfate mixing % with $Na^+$ would be required as the $Na^+$ concentrations are lower. This all shows that the results of Bondy et al. are not contrary to our mixing assumptions.

In fact we feel Bondy et al. (2018) supports our findings based on their general statements. From the Abstract, Bondy et al. (2018) states "These results emphasize that neither external nor internal mixtures fully represent the mixing state of atmospheric aerosols, even in a rural, forested environment,…". From the conclusions, Bondy states "Although SOA/sulfate dominated the overall measured aerosol population, especially in the accumulation mode (0.2-1.0 μm), it was found to be present at supermicron sizes as well." We have noted these findings and cited Bondy et al. (2018) in the discussion of mixing state.

3. "*When considering mixing state, this argument is not as clear as it might appear. If dust or salts are introduced (Allen et al. 2015, Bondy et al. 2017), $Na^+$ will go up, but so will sulfate from sea salt, though it is likely a low fraction of overall sulfate (that is externally mixed from the most of the sulfate mixed with OC in submicron particles). The equilibration time scale between (as modeled now in the paper) is not well known in the atmosphere, though if the authors can define that it would be useful. Thus, the correlation here may be indicative of shifting concentrations of different populations and not particularly strong evidence that $Na^+$ is leading to changes in R (e.g. correlation equals not causation).*"

Based on a scaling analysis (see Equation (12.48) in Seinfeld and Pandis (2006)), for a 10 nm thick film a rough estimation gives the characteristic time scale of diffusion as 0.0001 sec for a typical diffusion coefficient of $10^{-12}$ $m^2/s$ in solid, and $1 \times 10^{-7}$ sec for a typical diffusion coefficient of $10^{-9}$ $m^2/s$ in liquid (the diffusivity is cited from http://webserver.dmt.upm.es/~isidoro/dat1/Mass%20diffusivity%20data.pdf). The above characteristic times are miniscule relative to the equilibration time scales of approximately 30 min for $NH_3$-$NH_4^+$ in the ambient atmosphere (Dassios and Pandis, 1999; Cruz et al., 2000; Fountoukis et al., 2009). Therefore, the quantitative data suggests this is not an issue.

4. "*The argument that mixing does not alter the conclusions of the original analysis is surprising and I am still unclear as to how this can be. Particularly as the authors now cite work showing the pH is predicted to be higher for particles > 1 micron than < 1 micron. The authors are correct to point out that not all sulfate is in SOA particles, but based on filter measurements and AMS measurements at SOA, it would be safe to say the vast majority of sulfate is present in submicron SOA ‑ sulfate particles. The concern is that if no NVCs are present in > 95% of SOA ‑ sulfate particles, it seems highly unlikely that the R is being controlled by NVCs. The authors have addressed the inverse concern (e.g. does having a small amount of sulfate with NVCs change the overall finding), but this is not the major concern. The concern is that NVCs are not present in the very acidic SOA ‑ sulfate particles that are the focus of this paper. If the NVCs in the 2nd externally mixed population are driving acidity in the other population that is a very surprising result, though perhaps I'm misunderstanding the external mixing section.*"

This has been addressed above. Again, the added plots show there is no significant difference between overall $R$ assuming complete internal mixing (i.e., as Silvern did in their paper and we did in ours) if some fraction of the sulfate is mixed with NVCs (on average 18% for periods of $Na^+$ above the LOD, only 5% by mass when $Na^+$ is at the LOD, and lower % at lower $Na^+$). We refer the reviewer again to the revised paper Fig 7.

5. "*I am still uncomfortable with the extensive use of periods of data below LOD and inferred ion balance. LOD is admittedly arbitrarily defined, but it is an analytical standard, since as data approaches background noise, measurements are less reliable. I still have concerns that a higher concentration of "inferred" $Na^+$ is needed to make the model match the measurements, perhaps that means that $Na^+$ is not driving the R here? It is appreciated that the authors clearly define the time periods.*"

This is discussed in our response to reviewer 3 above.  As noted, there is nothing special about the choice of LOD, researchers define it different ways. Fig. 4a clearly demonstrates that there is no discontinuity in the model or the data in the molar ratio discrepancy as a function or $Na^+$ when moving below the $Na^+$ LOD, and it correlates very well with MARGA Na, which has a lower LOD. If those NVCs are not driving the molar ratio, we would not see the very good agreement between predicted and observed molar ratio for WINTER and SEARCH data sets, when inferred $Na^+$ is added to the model input.

6. "*More evidence continues to come out about diffusion limitations of viscous particles (see citations within (Reid et al., 2018)) and the role of coatings inhibiting partitioning that are not simply related to organic mass fraction (Zhang et al., 2018). Zhang et al. from the Surratt Group this year for example showed that*

*even a 10 nm coating of oxidized monoterpene SOA, could decrease reactive uptake coefficients by a factor of 4 and that would decease isoprene SOA formation via IEPOX by 15‑20% in conditions representative of the SE US. Clearly, IEPOX is a much larger molecule than ammonia, but at the ionic strengths listed below it seems very plausible that a distinct organic phase that could inhibit water or ammonia uptake could be present. I think given the caveats of the author's own assumptions (e.g. inferring sodium concentrations), that the statement ruling out a role for organic coatings and glassy organics should be weakened."*

As noted, the Zhang paper is not directly applicable to this work. Generalizations from laboratory experiments involving completely different chemical systems (with much larger molecular weight, hence less diffusive and more susceptible to kinetic limitations) to this work are not directly comparable. Finally, stating that delays are "up to a factor of 4" means that the gas-to-particle mass transfer rate is still is within the same order of magnitude; persistent disequilibrium in ammonia-ammonium partitioning requires considerably longer delays than that.

7. *"By the authors' standard of a direct measurement, would they consider a pH probe a direct measurement of H+ activity/pH? If not, then it is likely the direct measurements of H+ or pH are essentially impossible based on the standard the authors establish. pH probes and other measurements of pH have a long established history, even if there are a few assumptions needed since the activity of each ion is solution is not known precisely in many systems (including aerosols). Direct in this discussion was intended to refer to measuring both the acid and conjugate base concentrations directly, of which the Rindelaub measurement is the first of its kind. Arguments over the definition of "direct" aside, the authors misunderstand the Rindelaub paper, as it does not assume an ideal solution. Activity calculations are conducted for all species in solution (since it is a model system). The followup work in Craig et al., includes an entire figure on H+ activity coefficient versus ionic strength for inorganic, organic, and mixed system. Activity coefficients are calculated for all components in both Craig et al. and Rindelaub et al., so the non‑ideality is not being ignored, as the authors claim."*

There apparently is a misunderstanding here, we were not claiming that nonideality was being ignored. In any case, Rindelaub et al. (2016) reference and associated discussion was removed and so this is a comment on a comment, not an issue with what is in the latest version of the manuscript.

8. *"The authors note that the average ionic strength for this study is 29 mol/liters, which is quite high and brings about three questions:*
*1) At that concentration are there concerns about the limited water present and the impact on the thermodynamic calculations? How are the authors handling the fact that most thermodynamic models*

*struggle with non‑dilute solutions? This may be a lack of knowledge of ISORROPIA, but Debye‑Hückel breaks down above 0.1 M ionic strength I believe, does it not? How are the authors avoiding this, if so, it would be helpful to mention this high ionic strength and include a brief discussion.*"

ISORROPIA-II, like all atmospheric aerosol thermodynamic models, uses activity coefficients to calculate nonidealities. The model uses the mean activity coefficient models of Kusik and Missner for ion pairs and the multicomponent mixing rule of Bromely. This approach applies to high ionic strengths, at least 30 M (see relevant discussion in Fountoukis and Nenes, 2007). The ability to reproduce $NH_3$, $NH_4^+$, and liquid water in SOAS (Guo et al., 2015) for the range of relative humidity considered demonstrates that it captures nonidealities reasonably well up to very high ionic strengths.

"*2) How would this change if ISORROPIA accounted for organic components, a large mass fraction of the aerosols in question? My hunch is that it would lower the ionic strength, which would impact the figure shown below of shifting Ka (moving less to left), so it would be helpful if the authors could address this.*"

This is a good point. Vasilakos et al. (2018) and Song et al. (2018) added organic-inorganic interaction calculations in the soluble phase and found that the pH to be affected very little. Pye et al. (2018) considered liquid-liquid phase separation effects, and found a slightly larger impact on pH, but it was still within 0.7 units of assuming a single aqueous organic-inorganic phase. An important impact of organics is to contribute liquid water, and through this contribution affect the inorganic equilibria, and is something we already have discussed in the manuscript.

"*At such high ionic strengths and lower water activities, it seems likely that many of the inorganics would end up salting out of the organic component, leading to core‑shell liquid liquid phase separations. Despite the authors stating that the organics will not have an effect, it would seem this likely supports that these organic phases, possibly quite viscous, could impact partitioning and thus R. If the authors could address this it would be helpful.*"

We do not claim there is absolute no effect from organics on gas-particle partitioning, but the effects are minor in the SOAS or WINTER studies, as the partitioning of inorganic semivolatile species can be carried out without considering organics. Pye et al. (2018) confirms a secondary effect from organics by including organic species in thermodynamic modeling. The effect of semi-solid phases, would be to slow down partitioning of all species, including water, ammonia/ammonium; the data suggests this is not the case. Guo

et al. (2015) provides a relevant discussion; in summary, the oxidation state and the relative humidity seen in the dataset suggests that viscous aerosol is unlikely.

9. "*Considering the challenging processes being studied by this paper and other papers focused on this topic, the authors would be well‑served to acknowledge the possibility of other factors beyond their proposed explanation, which relies on its own assumptions.*"

We are confident in our conclusions as they are well supported by observational data. Ammonium partitioning, see Fig 2a, would not be accurately predicted by an equilibrium models if an organic film inhibits $NH_3$ uptake to the point where equilibrium is never reached. We also point out that the organic effects of Silvern et al. (2017) are based on an inference that may be from a bias in the observed $NH_4^+$ concentration (i.e., use of CSN data). We feel that this, together with a lack of correlation of the *R* discrepancy with organic fraction further presents strong support for our hypothesis.

Nevertheless, at the end of the paper, we have added text on limitations with this analysis: "Further assessments on possible effects of organic effects on semi-volatile partitioning of inorganic species however should be carried out, especially for regions that are chemically different from the Eastern US conditions evaluated in this study."

10. "*The authors make a fair point, presuming organic species do not lower the ionic strength significantly. How much would this then shift the $pK_a$ of the bisulfate/sulfuric acid system, the authors noted in comment 13? Would that likely mean that it would be much lower than could be observed in ambient aerosols? Thus, sulfate and bisulfate are the only forms of sulfate that need to be addressed for aerosols? Some clarification would be helpful.*"

This is indeed a good point. From the calculations (that consider ionic strength effects, hence shifts in the $pK_a$), if pH is larger than -1 then $HSO_4^-/SO_4^{2-}$ are the only forms of sulfate that we need to consider in the thermodynamics.

[revised manuscript text omitted]

**Contents of this file**

**Table S1-S2**

**Figures S1-S12**

**1. Differences in molar ratio ($R$) observations between SOAS, SEAECH, and CSN**

**Table S1.** Comparisons of observed $PM_{2.5}$ ions and molar ratio between SOAS, SEARCH, and CSN ground sampling sites for the 11-23 June 2013 period (Fig. 1 in the main text). Since CSN (Chemical Speciation Network) doesn't have a site at CTR to be directly compared to SOAS and SEARCH, the two closest sites at Birmingham (BHM) and Montgomery (MTG) are used. The most direct comparison is between Birmingham SEARCH and CSN data. Means are shown with standard deviations.

| Network | SOAS | SOAS | SEARCH | SEARCH | CSN | CSN |
|---|---|---|---|---|---|---|
| Site location | CTR | CTR | CTR | BHM | BHM | MTG |
| Site coordinate | 32.90289, −87.24968 | 32.90289, −87.24968 | 32.90289, −87.24968 | 33.55302, −86.81485 | 33.49972, −86.92417 | 32.41281, −86.26339 |
| Method | PILS-IC | MARGA(-IC) | Teflon filter(-IC) | Teflon filter(-IC) | Nylon filter(-IC) | Nylon filter(-IC) |
| $NH_4^+$, µg m$^{-3}$ | 0.64 ± 0.22 | 0.79 ± 0.22 | 0.63 ± 0.13[*] | 0.69 ± 0.20[*] | 0.24 ± 0.14 | 0.25 ± 0.19 |
| $SO_4^{2-}$, µg m$^{-3}$ | 2.06 ± 0.68 | 2.38 ± 0.66 | 2.16 ± 0.44 | 2.23 ± 0.51 | 1.69 ± 0.40 | 1.46 ± 0.84 |
| $Na^+$, µg m$^{-3}$ | 0.07 ± 0.09 | 0.09 ± 0.10 | 0.06 ± 0.04 | 0.05 ± 0.04 | 0.13 ± 0.06 | 0.10 ± 0.04 |
| **$R$** | **1.70 ± 0.23** | **1.78 ± 0.18** | **1.57 ± 0.11** | **1.64 ± 0.14** | **0.70 ± 0.36** | **0.75 ± 0.42** |
| Data points | 229 | 229 | 13 | 4 | 5 | 3 |
| Notes on data | Hourly data | Hourly data | Daily data; every day | Daily data; every three days (6/12-6/21) | Daily data; every three days (6/9-6/24) | Daily data; every six days (6/9-6/21) |
| Reference | (Guo et al., 2015) | (Allen et al., 2015) | (Edgerton et al., 2005; Hidy et al., 2014) | | (Solomon et al., 2014) | |

[*] SEARCH $NH_4^+$ was measured by automated colorimetry.

**Table S2.** Comparisons of observed $PM_{2.5}$ ions and molar ratio between SEARCH and CSN ground sampling sites for the year of 2013.

| Network | SEARCH | SEARCH | CSN | CSN |
|---|---|---|---|---|
| Site location | CTR | BHM | BHM | MTG |
| Site coordinate | 32.90289, −87.24968 | 33.55302, −86.81485 | 33.49972, −86.92417 | 32.41281, −86.26339 |
| Method | Teflon filter(-IC) | Teflon filter(-IC) | Nylon filter(-IC) | Nylon filter(-IC) |
| $NH_4^+$, µg m$^{-3}$ | 0.55 ± 0.28[*] | 0.72 ± 0.31[*] | 0.48 ± 0.34 | 0.41 ± 0.29 |
| $SO_4^{2-}$, µg m$^{-3}$ | 1.71 ± 0.89 | 1.96 ± 0.90 | 1.91 ± 0.99 | 1.65 ± 0.89 |
| $Na^+$, µg m$^{-3}$ | 0.05 ± 0.05 | 0.05 ± 0.05 | 0.13 ± 0.30 | 0.10 ± 0.08 |
| **$R$** | **1.75 ± 0.28** | **2.05 ± 0.05** | **1.26 ± 0.59** | **1.24 ± 0.59** |
| Data points | 154 | 111 | 93 | 61 |
| Notes on data | Daily data; every three days | Daily data; every three days | Daily data; every three days | Daily data; every six days |
| Reference | (Edgerton et al., 2005; Hidy et al., 2014) | | (Solomon et al., 2014) | |

[*] SEARCH $NH_4^+$ was measured by automated colorimetry.

**1.2. Relative fractions of SO₄²⁻ and HSO₄⁻ for SOAS fine particles**

[Figure]

**Fig. S1.** Relative fractions of $SO_4^{2-}$ (red) and $HSO_4^-$ (blue) calculated based on ideal solutions (all activity coefficients equal one) and the SOAS non-ideal conditions. The average activity coefficients of

$\gamma_{SO_4^{2-}}/\gamma_{HSO_4^-} = 0.01$ $\gamma_{SO_4^{2-}}/\gamma_{HSO_4^-} = 0.01$ are predicted by ISORROPIA for the SOAS fine particles. $\gamma_{H^+} = 1$ $\gamma_{H^+} = 1$ is assumed; a smaller $\gamma_{H^+}$ $\gamma_{H^+}$ shifts the red and blue curves towards the left, increasing $SO_4^{2-}$ relative fraction at a given pH. The dissociation constant of $HSO_4^-$ is $1.015\times10^{-2}$ mol kg⁻¹ at 298.15 K (Fountoukis and Nenes, 2007)(Fountoukis and Nenes, 2007).

**3. Comparison of PILS NVCs to MARGA NVCs**

[Figure]

**Fig. S2.** (a) Comparison of $PM_{2.5}$ PILS and MARGA $Na^+$. (b) Comparison of inferred $Na^+$ (from ion charge balance; $Na^+ = 2SO_4^{2-} + NO_3^- + Cl^- - NH_4^+$, nmol m$^{-3}$) by PILS and MARGA to total measured NVCs by MARGA (represented by $Na^+$), and (c) comparison of PILS and MARGA ammonium-sulfate molar ratios (*R*). Data are from the SOAS study.

**4. Comparison of ion charge balance inferred Na⁺ including H⁺ versus excluding H⁺**

[Figure]

**Fig. S3.** Comparisons of ion charge balance inferred Na⁺ including H⁺ ($2SO_4^{2-} + NO_3^- + Cl^- - NH_4^+ - H^+$; y-axis) versus excluding H⁺ ($2SO_4^{2-} + NO_3^- + Cl^- - NH_4^+$; x-axis) for three data sets used in the paper, (a) SOAS (Fig. 1), (b) WINTER (Fig. 5), and (c) SEARCH CTR (Fig. 6). The H⁺ concentration was determined using ISORROPIA in an iterative approach. Na⁺ is predicted from the ion balance is included with all other gas/particle species in the model, resulting in a predicted H⁺. This H⁺ is included in the ion balance to predict a new Na+, which is then used in a new model iteration. The procedure is repeated until the Na⁺ concentration converges. The number of iterations for conversion are 1 for (a), 5 for (b), and 0 for (c), respectively, until inferred Na⁺ converges. Orthogonal distance regression (ODR) fits are shown and uncertainties in the fits are one SD.

**3.5. Comparison of measured and ISORROPIA-predicted ammonium-sulfate molar ratios.**

[Figure]

**Fig. S3S4.** Comparisons of PM$_{2.5}$ ammonium-sulfate molar ratios ($R$) between measurements and ISORROPIA-predictions for the base case but with differing Na$^+$ inputs. Data are from the SOAS study. Red numbers are the means and red error bars are one SD. Standard box-whisker plots are shown, with 100% and 0% data indicated by black error bars. Top and bottom of box are the interquartile ranges (75% and 25%) centered around the median value (50%). Comparisons contrast all data and periods when measured Na$^+$ > PILS LOD of 0.07 µg/m$^3$. Inferred Na$^+$ is the value calculated from an ion charge balance representing NVCs.

[Figure]

**Fig. S4S5.** Comparisons of PM$_{2.5}$ ammonium-sulfate molar ratios ($R$) between measured and ISORROPIA-predictions for differing Na$^+$ inputs for SOAS data. ISORROPIA was run with base case inputs and differing Na$^+$ inputs: (a) all Na$^+$ measured by the PILS, (b) NVCs (i.e., Na$^+$) inferred from an ion balance, and (c) Na$^+$ (NVCs) set to zero.

**4.6. The relationships between errors in molar ratio and organic aerosol mass**

[Figure]

**Fig. S5S6.** Effect of nonvolatile cations (NVC) on the PM$_{2.5}$ ammonium-sulfate molar ratio ($R$) as a function of measured organic aerosol (OA) concentrations based on AMS data (SOAS). The orange circular points denote Δ$R$ calculated from ISORROPIA predicted $R$ with measured Na$^+$ included in the model input minus ISORROPIA predicted $R$ without Na$^+$ in the model input. Grey diamonds are for Δ$R$ equal to measured $R$ minus 2. Note that Δ$R$ should be negative since including Na$^+$ in the thermodynamic model results in $R$ lower than 2, whereas not including Na$^+$ results in an $R$ close to 2 (on average $R$ predicted without Na$^+$ is 1.97 ± 0.02).

7. **Comparison of predicted and measured $R_{SO4}$ by adding NVC**

[Figure]

**Fig. S7.** Comparison between PM$_1$ AMS-measured $R_{SO4}$ and ISORROPIA-predicted $R_{SO4}$, which is based on (a) inferred Na$^+$ excluding H$^+$ (see Fig. S3 x-axis) and (b) inferred Na$^+$ including H$^+$ (see Fig. S3 y-axis). The color wave indicates an ion charge balance, (2SO$_4$$^{2-}$ + NO$_3$$^-$ − NH$_4$$^+$) in units of nmol m$^{-3}$. The figure (a) is the same as Fig. 5b in the main text. The two results, (a) and (b) are very close, and the inferred Na$^+$ including H$^+$ produces slightly better result for a closer slope to one. ODR fits are shown and uncertainties in the fits are one SD.

[Figure]

**Fig. S8.** Comparison between ISORROPIA-predicted and AMS-measured PM$_1$ $R_{SO4}$ ($R_{SO4}$ = (NH$_4$$^+$ − NO$_3$$^-$)/SO$_4$$^{2-}$, mol mol$^{-1}$), where the model predictions are based on NVC-NH$_4$$^+$-SO$_4$$^{2-}$-NO$_3$$^-$(-Cl$^-$) system for the WINTER study. NVC (nonvolatile cation) was determined by an ion charge balance (color wave), that is, (2SO$_4$$^{2-}$ + NO$_3$$^-$ − NH$_4$$^+$) in units of nmol m$^{-3}$. This results in 200% mole-equivalent concentrations of Na$^+$ and K$^+$ compared to Ca$^{2+}$ and Mg$^{2+}$ due to +1 versus +2 charges. NH$_4$$^+$, SO$_4$$^{2-}$, NO$_3$$^-$ are observed AMS mass concentrations. For each graph, NVC is set to be a single species, including (a) Na$^+$, (b) K$^+$, (c) Ca$^{2+}$, (d) Mg$^{2+}$. For K$^+$, Ca$^{2+}$, and Mg$^{2+}$, a trace amount of total chloride (0.01 µg m$^{-3}$; a negligible level of Cl$^-$, 0.0012 µg m$^{-3}$, is predicted) is assumed to eliminate potential model errors but doesn't perturb the calculation of pH or HNO$_3$-NO$_3$$^-$ partitioning. (Note that, Cl$^-$ is only utilized as an input

to ISORROPIA and is not included in the charge balance calculation.) Adding $Na^+$ and $K^+$ results in predicted $R_{SO4}$ agreeing with measured $R_{SO4}$. $Mg^{2+}$ also results in closer agreement, although some points deviate. $Ca^{2+}$ doesn't work at all as it precipitates out from the aqueous phase as $CaSO_4$. The solubility of $CaSO_4$ is only 0.2 g per 100 mL water at 20 °C. An approximate calculation on $CaSO_4$ solubility shows that the average predicted particle liquid water $\cancel{W_i}W_i$ (2.0 µg m$^{-3}$) could only dissolve 0.004 µg m$^{-3}$ $Ca^{2+}$, a tenfold lower amount than the inferred $Ca^{2+}$ of 0.13 µg m$^{-3}$ from an ion charge calculation. ODR fits are shown and uncertainties in the fits are one SD.

**6.8. The nonlinear response of NH₃-NH₄⁺ or HNO₃-NO₃⁻ partitioning to pH (S curve)**

[Figure]

**Fig. S7S9.** S curves illustrate the nonlinear response in particle phase fraction, ε(NH₄⁺) or ε(NO₃⁻), to variation in pH: (a) ε(NH₄⁺) and (b) ε(NO₃⁻) plotted vs. pH. The two S curves are calculated based on T = 20 °C, particle liquid water level = 5 µg m⁻³, and ideal solution (i.e. γ = 1). The S curve equations can be found at Guo et al. (2017)γ = 1). The S curve equations can be found at Guo et al. (2017). Non-ideality only shifts the S curves but does not change the shapes. The 0.3 unit pH (SOAS) and 0.5 unit pH (WINTER) variations (biases) are the upper limit values based on the difference between zero and inferred Na⁺ inputs and indicated by paired red and blue sticks, respectively. The response of ε(NH₄⁺) or ε(NO₃⁻) to pH reaches maximum at 50% ε(NH₄⁺) or ε(NO₃⁻) (i.e., position (2), 0.3 unit pH change causes ~20% or 0.5 unit pH change causes ~30% shift in the particle phase fraction), but down to nearly zero when 100% or 0% ε(NH₄⁺) or ε(NO₃⁻) (e.g. position (1) or (3)).

**7.9. Thermodynamic predictions of the 15 years' summertime molar ratio at CTR site (Centreville, AL)**

[Figure]

**Fig. S8S10.** Comparison of the measured and predicted $R_{SO4}$ (with inferred Na$^+$ as input), summer means at CTR, as shown in the Figure 6 in the main text. The upper limit of $R_{SO4}$ is 2 for a composition of $(NH_4)_2SO_4$ in ambient aerosols. A few observed points above 2 are results of measurement uncertainties.

**8.10.** **Internal vs. external mixture: the effect on molar ratio prediction**

[Figure]

**Fig. S9S11.** Comparison of predicted $R$ between internally and externally mixed aerosols, as shown in Figure 7 in the main text. NVCs (here represented by Na$^+$) are all assumed in PM$_{1-2.5}$ for the external mixing case, while (a) 0%, (b) 10%, (c) 20%, and (d) 30% (mass) fractions of PM$_{2.5}$ sulfate is assumed to be mixed with Na$^+$, as shown above each figure. The two points with the highest Na$^+$ concentrations require 40% sulfate in PM$_{1-2.5}$ to reach agreement (i.e. on 1:1 line).

[Figure]

**Fig.** S12. Minimum sulfate fraction in $PM_{1-2.5}$ to obtain agreeable ammonium-sulfate molar ratio ($R$) between the internally and externally mixed aerosols (as shown in Fig. S9) versus measured $Na^+$ concentrations.